# Growth-dependent concentration gradient of the oscillating Min system in *Escherichia coli*

Claudia Morais Parada[1]*, Ching-Cher Sanders Yan[2]*, Cheng-Yu Hung[3]*, I-Ping Tu[3], Chao-Ping Hsu[2,4,5], and Yu-Ling Shih[1,6,7]

**Cell division in *Escherichia coli* is intricately regulated by the MinD and MinE proteins, which form oscillatory waves between cell poles. These waves manifest as concentration gradients that reduce MinC inhibition at the cell center, thereby influencing division site placement. This study explores the plasticity of the MinD gradients resulting from the interdependent interplay between molecular interactions and diffusion in the system. Through live cell imaging, we observed that as cells elongate, the gradient steepens, the midcell concentration decreases, and the oscillation period stabilizes. A one-dimensional model investigates kinetic rate constants representing various molecular interactions, effectively recapitulating our experimental findings. The model reveals the nonlinear dynamics of the system and a dynamic equilibrium among these constants, which underlie variable concentration gradients in growing cells. This study enhances quantitative understanding of MinD oscillations within the cellular environment. Furthermore, it emphasizes the fundamental role of concentration gradients in cellular processes.**

## Introduction

Self-organization provides spatiotemporal information related to the cellular organization of biological processes. This self-organization phenomenon of proteins is exemplified by the MinD and MinE proteins of *Escherichia coli*, which undergo spontaneous oscillation that mediates the placement of the division septum at the mid cell. Although a wealth of biochemical and biophysical characterization methods are available to comprehend the complex molecular interactions that underlie the oscillatory pattern formation of the Min system (Denk et al., 2018; Lutkenhaus, 2007; Shih and Zheng, 2013; Vecchiarelli et al., 2016), mapping the *in vitro* characteristics in the cellular context remains a great challenge. The differences in the spatial and temporal scales and the reduced complexity of the in vitro reconstitution experiments may partially explain the differences. Our investigation into Min protein oscillation in the cellular context was prompted by interest in the *in vivo* versus *in vitro* characteristics of the Min system and the relation of this system to global bacterial physiology.

MinD is a deviant form of Walker-type ATPases, and MinE drives MinD oscillation by stimulating the ATPase activity of MinD (Hu and Lutkenhaus, 2001; Zhou et al., 2005). In the cellular context, an oscillation cycle starts with a fully assembled MinD polar zone covering half of a cell, with the MinD polar zone then disassembling in response to stimulation by MinE. The MinD molecules that dissociate during polar zone disassembly diffuse to the opposite pole of the cell and reassemble into a polar zone until half of the cell is covered. The new polar zone then disassembles and the molecules diffuse back to the original pole for reassembly into a polar zone to complete an oscillation cycle. Mechanistically, the dynamic MinD concentration gradient emerges from the intricate balance of the MinD and MinE concentration ratio, resulting in pole-to-pole oscillation (Mizuuchi and Vecchiarelli, 2018; Vecchiarelli et al., 2016). In a time-averaged form, the MinD concentration gradient along a cell's long axis is high at the poles but low at the midcell. MinC interacts and oscillates along with MinD to destabilize the FtsZ polymers at the poles (Hu and Lutkenhaus, 1999; Hu et al., 1999; Raskin and de Boer, 1999a, 1999b), thus facilitating the formation of the FtsZ ring to direct the assembly of the division septum at the midcell (de Boer et al., 1988; Fu et al., 2001; Raskin and de Boer, 1999a, 1999b).

The concentration gradient of the Min proteins over the cell cycle is one missing piece of information that is needed to bridge what is known about the underlying biochemical and molecular interactions and the oscillation in the cellular context. By quantitative imaging analyses of the MinD concentration

[1]Institute of Biological Chemistry, Academia Sinica, Taipei, Taiwan; [2]Institute of Chemistry, Academia Sinica, Taipei, Taiwan; [3]Institute of Statistical Science, Academia Sinica, Taipei, Taiwan; [4]Division of Physics, National Center for Theoretical Sciences, Taipei, Taiwan; [5]Genome and Systems Biology Degree Program, Academia Sinica and National Taiwan University, Taipei, Taiwan; [6]Institute of Biochemical Sciences, National Taiwan University, Taipei, Taiwan; [7]Department of Microbiology, College of Medicine, National Taiwan University, Taipei, Taiwan.

*C.M. Parada, C.-C.S. Yan, and C.-Y. Hung contributed equally to this paper. Correspondence to Yu-Ling Shih: ylshih10@gate.sinica.edu.tw; Chao-Ping Hsu: cherri@sinica.edu.tw.

gradient during oscillation, we discovered the morphological plasticity of the MinD concentration gradient when cells elongate concomitantly with a gradual reduction in the MinD concentration at the midcell. Presumably, the distribution of the division inhibitor MinC is synchronous with the spatiotemporal difference in the MinD concentration, mediating stable placement of the FtsZ ring at the midcell after MinD reaches a low concentration (Cameron and Margolin, 2024). We found that, contrary to what was expected, the oscillation period does not change significantly with cell elongation, suggesting that the intrinsic properties of the Min system are yet to be elucidated. These observations prompted us to hypothesize that the length-dependent variable concentration gradient is generated by the coordinated molecular interactions underlying the Min oscillation. Additionally, we used a numerical model of the reaction-diffusion system to support the experimental measurements and examined the molecular interactions represented by the kinetic rate constants of each reaction step. This numerical model can generate a similar coupling of the variable concentration gradient and the diminishing MinD midcell concentration as the length increases, providing insights into the intricate molecular interactions underlying the length-dependent variable concentration gradient.

Taken together, the findings of this study reveal the inherent plasticity and adaptability of the MinD concentration gradient, a critical factor in the Min system that orchestrates division site placement. By exploring the parameter space of kinetic rate constants that represent different molecular interactions within the Min system using a numerical model, we reveal how these interactions lead to variable concentration gradients during cell growth. This concept of variable cellular concentration gradients not only enhances our understanding of bacterial cell division but also offers broader insights into essential cellular processes such as transport, signaling, and homeostasis, which rely on concentration gradients.

## Results

### Endogenous expression of sfGFP-MinD ensures the characteristic features of oscillation

In this study, *E. coli* strain FW1541, which expresses the endogenous level of *sfGFP-minD* from the native chromosomal locus (Wu et al., 2015a), at the exponential growth stage was used (Table S1). The cellular abundances of sfGFP-MinD and MinE in FW1541 cells were determined to be 2,205 ± 178 molecules per cell (1.95 ± 0.16 µM) and 1,580 ± 148 molecules per cell (1.40 ± 0.13 µM), respectively. In parallel, the cellular abundances of MinD and MinE in the parental strain W3110 were 3,532 ± 61 molecules/cell (3.12 ± 0.05 µM) and 2,150 ± 228 molecules/cell (1.90 ± 0.20 µM), respectively (Fig. S1). Although the expression levels of both sfGFP-MinD and MinE were lower in FW1541, the MinD and MinE concentration ratios were close (sfGFP-MinD/MinD to MinE: FW1541, 1.40; W3110, 1.64). Thus, the protein abundances of sfGFP-MinD and MinE in this strain were comparable with those in the parental strain W3110 and to those in previous reports (Hale et al., 2001; Li et al., 2014; Schmidt et al., 2016; Shih et al., 2002). Moreover, they fall in a range within

which variation of the absolute number of MinD (and MinE) molecules can be tolerated if the ratio of MinD to MinE is maintained.

The classical features of MinD oscillation were identified alongside new information by tracing the fluorescence intensity profiles measured along the medial axes of cells over time (Figs. 1 and 2). First, the standing wave pattern of the pole-to-pole oscillation was demonstrated by the registration of the sfGFP-MinD intensity plots located at various positions in the opposite cell halves (Fig. 1, A and B). The waves in opposite cell halves were registered in the anti-phase and the waves at various positions in the same cell half were registered nearly in phase (Fig. 1 B). Second, the uniformity of the oscillation intervals appeared to increase with length, as demonstrated in the kymographs that depicted changes in the fluorescence intensity over time in one dimension (Fig. 1 C). The periodicity gradually stabilized as the cells approached division, which was supported by the decreasing standard deviations of the periods as the cell length increased (Fig. 1 D). These observations echoed the changes in the oscillation patterns before and after cell division, as previously reported (Juarez and Margolin, 2010). Third, the oscillation period, which is defined as the time taken by the MinD to travel back and forth between two cell poles, was determined to be 46 s (median, *n* = 130; Table 1). The oscillation velocity was then calculated to be 0.127 µm/s by dividing the period by two cell lengths, which is the distance traveled by MinD during one oscillation cycle. These results were not only comparable with those of previous studies (Fu et al., 2001; Hale et al., 2001; Shih et al., 2002) but also more closely related to physiological conditions than those of studies that either expressed from plasmids (Di Ventura and Sourjik, 2011; Meacci and Kruse, 2005; Tostevin and Howard, 2005; Touhami et al., 2006) or under the control of the *lac* promoter on the chromosome (Juarez and Margolin, 2010).

### The oscillation period is quite stable within a growing cell exhibiting different lengths at different time points

Alongside the characteristic features, the oscillation period was relatively stable, showing only a slight change across different cell lengths (r = 0.226; Fig. 1 E). If the periodicity is maintained regardless of the length increase, the velocity of the Min oscillation must increase as the cell cycle progresses. Here, we considered only cell length because the shape descriptors of length, width, and aspect ratio showed little or no difference and were highly correlated (Fig. S2, A and B). Undivided cells showing constriction were included in the measurements. The cell length and time within a cell cycle were positively correlated in an actively growing population of cells (Fig. S2 C), allowing examination of the hypothesis that the oscillation period remained relatively stable throughout the cell cycle (Fig. S2 D) and that the velocity increased with time (Fig. 1 F).

In addition, the faster velocity in longer cells and slower velocity in shorter cells suggest that the velocity was reset to a slower speed after division. By tracking sfGFP-MinD oscillations in individual cells with a clear mother–daughter lineage, we confirmed that the velocity in newborn cells decreased significantly in all examined cases (Fig. 3), even when the period was

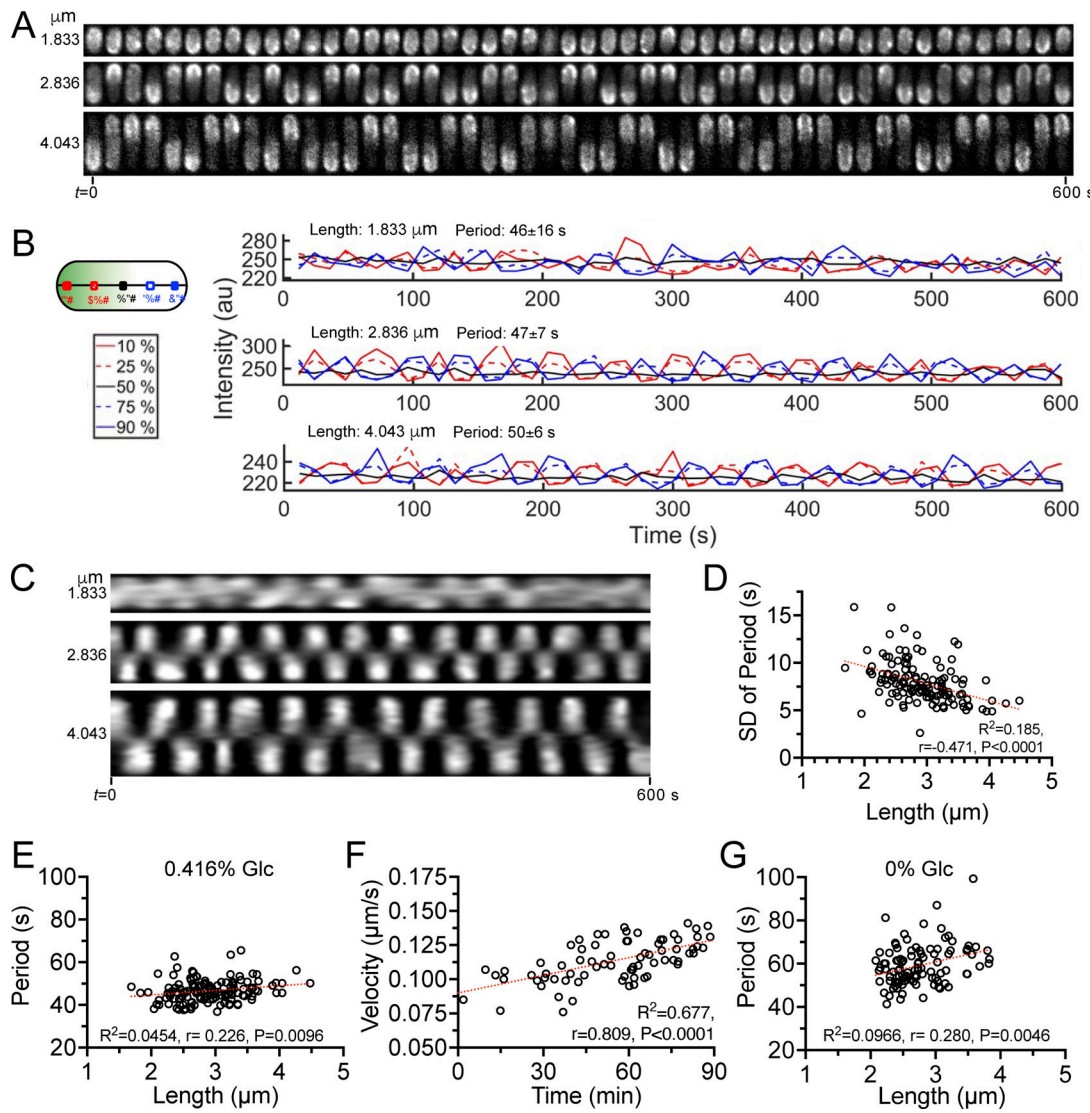

Figure 1. **MinD oscillation in cells of different lengths. (A)** Time-lapse micrographs illustrating the pole-to-pole oscillation of sfGFP-MinD in cells of different lengths. The exponentially growing *E. coli* strain FW1541 was imaged at 12-s intervals for 10 min. Scale bar: 2 μm. **(B)** Standing wave features are demonstrated at 10% and 25% of the cell length, which was equivalent to the positions at 90% and 75%, respectively, along the medial axes of cells. **(C)** Kymographs demonstrating changes in the intensity profiles with length. **(D)** The standard deviation (SD) of the oscillation period is reduced in longer cells. **(E and G)** The correlation plots between the oscillation period and cell length when cultured with 0.416% (E) and 0% glucose (G). The oscillation periods range from 37 to 66 s for 0.416% glucose and from 41 to 99 s for 0% glucose. **(F)** Correlation between the oscillation velocity and the cell cycle duration. In A–C, MinD oscillations in three representative cells with varying lengths are presented. In D–G, analyses based on pooled data from multiple image fields are presented to demonstrate the oscillatory dynamics across cells of different lengths. Correlations were calculated using the nonparametric Spearman correlation method with 95% confidence intervals. The red dashed lines represent the lines fitted by simple linear regression. $R^2$, goodness of fit; r, Spearman coefficient; P: two-tailed probability; *n*, population size. D–F, *n* = 130; G, *n* = 101.

prolonged due to unfavorable growth conditions during imaging. Nonetheless, the position of the division site marked by FtsA at the midcell was found to be unrelated to cell length (Fig. 4). Thus, the quite stable oscillation period and variable velocity did not change the precision of the septum placement.

**The concentration of MinD changed little throughout the cell cycle**

We wondered how the cellular MinD and MinE concentrations contribute to the maintenance of the oscillation period. Thus, the bulk fluorescence of sfGFP-MinD in individual cells was measured from each time frame in the time-lapse sequence. As shown in Fig. 5 A, while the fluorescence of sfGFP-MinD gradually increased with time, the intensity per unit area (μm²) showed relatively small changes. All measurements obtained from the complete cell cycle were then identified, aligned, and fitted by simple linear regression. The fluorescence intensity was converted to the number of molecules by applying 2,205 molecules per cell at the midpoint of the experimental doubling time (75 min, *n* = 26) on the fitted curve (Fig. 5 B). This resulted in 1,844–2,537 molecules, equivalent to 0.92–1.26 μM in concentration, per cell over a cell cycle of 90 min. Interestingly, the

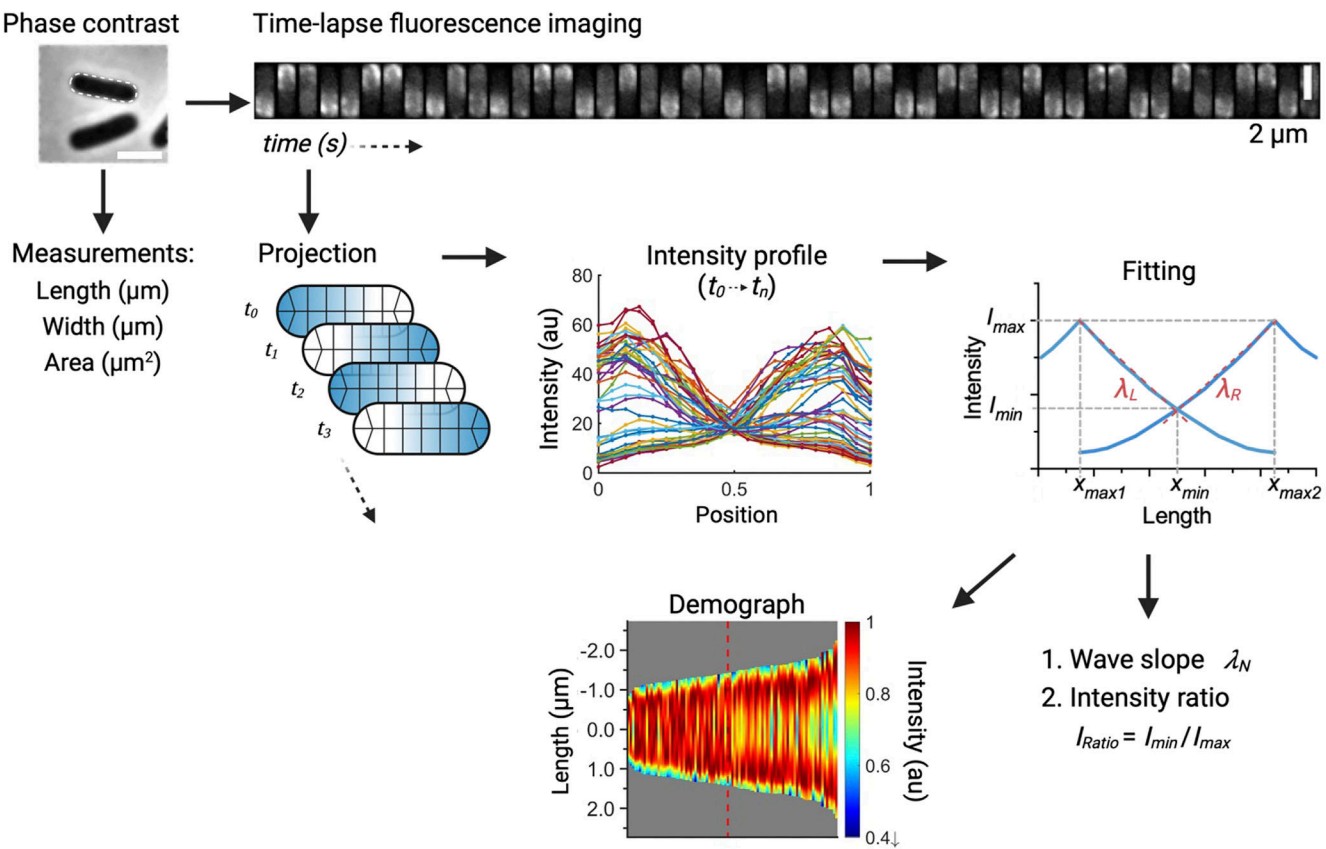

**Figure 2. Schematic diagram of image processing.** Top panel: A phase-contrast image was taken for measurements of cell dimensions before acquisition of the time-lapse sequence of sfGFP-MinD oscillation. Middle panel: The fluorescence image of a cell was processed by projection of the intensity values onto the medial axis. The procedure was applied to all cell images in the entire time-lapse sequence (left). The intensity at different axial positions was plotted to visualize the fluctuations in fluorescence intensity, which indicated changes in protein concentration (middle). Fitting curves were generated using the intensity profiles of the same cell in the time-lapse sequence (right). Lower panel: The intensity profiles of different cells were projected into one dimension, sorted by cell length, and compiled into a kymograph (left). The fitted curves of individual cells were presented in XY plots followed by calculation of the wave slope ($\lambda_N$) and the intensity ratio ($I_{Ratio}$) (right).

values just before division were not doubled compared with daughter cells, suggesting a balance between de novo synthesis and degradation or a burst of MinD synthesis at cell division followed by a steady rate of synthesis. Nonetheless, the number of molecules per μm² fluctuated within a very narrow range (Fig. 5 B, blue-hued line). Thus, the MinD concentration may be of limited importance in maintaining the oscillation period throughout the cell cycle. However, when focusing within the midcell zone, defined as within 200 nm of the midpoint of a cell, the protein molecules per μm² were reduced to ∼20 at division, i.e., ∼4 molecules within the midcell zone (Fig. 5 C). This was calculated from the fraction of intensity in the MinD gradient profile. This analysis suggested that the spatiotemporal distribution of MinD, i.e., the concentration gradient, maybe the key to answering how the cellular MinD and MinE maintain the oscillation period.

**The wave slope toward the cell center becomes steeper as cells grow longer**

The time-averaged MinD concentration gradient was analyzed in detail (Fig. 2) to understand how it contributes to the abovementioned features. The fluorescence intensity profiles collected at different time points in individual cells in the time-lapse image sequences were separated by their locations at the left or right poles and fitted by the exponential decay equation. Then, the fitted curves for the entire population of cells were compressed into one dimension, sorted by cell length, and compiled into a kymograph, which provided a comprehensive view of the changes in the cellular MinD concentration gradient at different growth stages (Fig. 6 A). The kymograph results showed that the MinD concentration, proportional to the intensity, near the cell center was greater in shorter cells than in longer cells.

Next, the decay constant of the concentration gradient, $\lambda_N$, was calculated to numerically describe the wave slope of the MinD concentration gradient (Fig. 2). The $\lambda_N$ value was estimated over the normalized cell length, which was dimensionless for comparison across different lengths of cells. As demonstrated in Fig. 6 B, a smaller $\lambda_N$ (with a gentler slope) was found for shorter cells and a larger $\lambda_N$ (with a steeper slope) was found for longer cells, suggesting that the slope of the wave toward the cell center becomes steeper as the cells grow longer.

These observations led us to further hypothesize that the variable MinD concentration gradient could be coupled with a

Table 1. **Characteristics of sfGFP-MinD oscillation in the wild-type strain**

| Glucose | Statistics | Period (s) | Length (μm) | Velocity (μm/s) | Width (μm) | Aspect ratio | Area (μm²) | [a]n/n* |
|---|---|---|---|---|---|---|---|---|
| 0.416% | Median | 46 | 2.843 | 0.127 | 0.968 | 2.98 | 2.516 | 130/90 |
|  | Mean ± SD | 47 ± 5 | 2.911 ± 0.518 | 0.126 ± 0.023 | 0.966 ± 0.042 | 3.03 ± 0.59 | 2.559 ± 0.478 |  |
| 0% | Median | 57 | 2.676 | 0.094 | 0.895 | 2.95 | 2.165 | 101/82 |
|  | Mean ± SD | 59 ± 9 | 2.748 ± 0.436 | 0.094 ± 0.017 | 0.891 ± 0.037 | 3.09 ± 0.53 | 2.238 ± 0.385 |  |

[a]n is the sample size that indicates the number of cells used in the measurement of periodicity. n* is the sample size that indicates the number of cells used in extrapolating the $\lambda_N$ values.

changing MinD concentration at the midcell to facilitate septum formation. Therefore, the second numerical descriptor, the intensity ratio ($I_{Ratio}$), was defined as the ratio between the minimum and maximum fluorescence values identified in the intensity profiles. The minimal intensity, $I_{min}$, was identified at the intersection of the paired concentration gradients (Fig. 2), which was in the proximity of the midcell with rare exceptions. The maximum intensity, $I_{max}$, was identified at ~14.9 ± 4.5% of the cell length from either pole. The decrease in fluorescence near the pole may be caused by the smaller volume of the hemisphere (Fig. S2 E). The $I_{Ratio}$ values calculated from $I_{min}$ and $I_{max}$ could reflect the amount of MinD accumulation near the midcell and were used to predict the probability of septum formation. As shown in Fig. 6 C, an inverse proportionality between $I_{Ratio}$ and cell length was found, suggesting that the difference in the MinD concentration at the poles and the division site gradually increased in growing cells.

## Sharper concentration gradient in shorter cells under glucose starvation

Similarly, a variable MinD concentration gradient during cell growth was also observed under glucose starvation conditions (Fig. 6, D–F), even though the oscillation period increased (Fig. 1 G). A greater reduction in the MinD concentration at the midcell occurred in shorter cells cultured in the absence of glucose (Fig. 6 D) than in those cultured with 0.416% glucose (Fig. 6 A). Furthermore, the $\lambda_N$ values were greater in cells cultured in the absence of glucose (Fig. 6 E), suggesting that a sharper concentration gradient under glucose starvation may facilitate division at shorter cell lengths. However, the $I_{Ratio}$ values at the median cell length showed no clear difference between nutrient shifts (Fig. 6, C and F), suggesting that the $I_{Ratio}$ values at the time of division were similar regardless of glucose supply.

The analyses provided a better understanding of the coupling between the varying concentration gradient and the division site placement. First, both $\lambda_N$ and $I_{Ratio}$ were length-dependent, as demonstrated in the $\lambda_N$–length and $I_{Ratio}$–length plots. Second, the increase in $\lambda_N$ was coupled with the decrease in $I_{Ratio}$, a condition that could favor the placement of the FtsZ ring to allow cells to prepare for division. Third, the slopes of both the $\lambda_N$–length and $I_{Ratio}$–length correlation plots were sharper in carbon-starved cells, suggesting that more drastic changes occur spanning shorter length ranges.

While fluctuations in intracellular ATP concentration caused by glucose downshifts are not expected to significantly impact the ATP needed for MinD oscillation, we cannot entirely rule out this possibility. When the cellular concentration of ATP is reported to be 1–5 mM under various conditions (Lasko and Wang, 1996; Mathis and Brown, 1976; Schneider and Gourse, 2004; Soini et al., 2005), the rate of ATP consumption by 6 μM MinD with an equal concentration of MinE was reported to be ~30 nmol Pi/mg MinD/min (~0.9 μmol Pi/μmol MinD/s) (Shih et al., 2011). Therefore, even during glucose starvation, the cellular ATP levels appear to be substantially higher than the amount consumed by MinD oscillations.

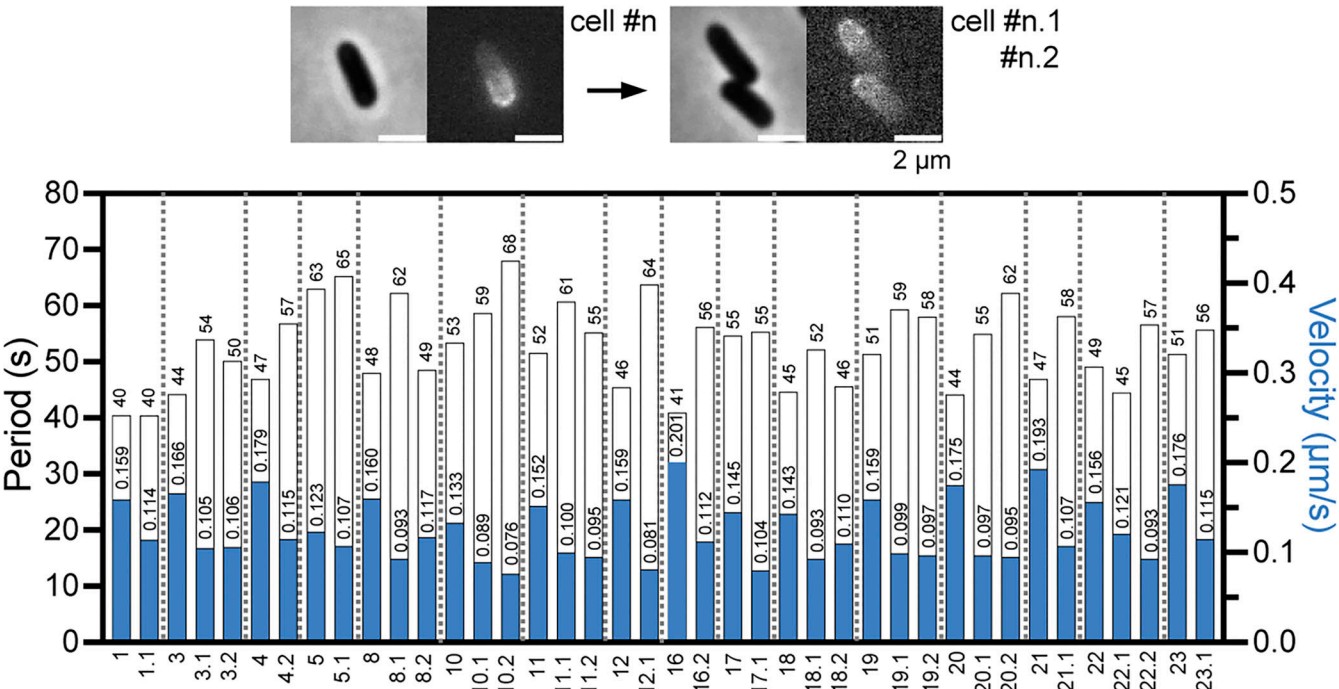

Figure 3. **The oscillation period and velocity before and after division demonstrated that the oscillation velocity decreased in newborn cells.** In some cases, the fluorescence in one of the two newborn cells was too weak to be studied. The numbers on the x-axis indicate the cell lineage. To image MinD oscillations before and after cell division, cells mounted on an agarose pad were initially imaged for 10 min, then incubated on the heating stage of the microscope at 30°C for 2 h before being imaged again for 10 min. Divided cells were selected for analysis.

### Simulating the dynamic MinD concentration gradient in growing cells

A one-dimensional (1D) mathematical model based on Fange and Elf (2006); and Wu et al. (2015a) was employed for revealing the physical insights of the observed dynamic concentration gradient as cells elongate. The model describes the reaction steps depicted in Fig. 7 A, including the attachment of cytosolic MinD-ATP to the membrane, MinD-ATP recruitment by the membrane-bound MinD-ATP, recruitment of MinE to the membrane by the membrane-bound MinD-ATP, detachment of

the MinDE complex from the membrane due to ATP hydrolysis in MinD by MinE stimulation, and recharging of MinD-ADP with ATP by nucleotide exchange. Each step corresponds to a rate constant, i.e., $k_D$, $k_{dD}$, $k_{dE}$, $k_{de}$, or $k_{ADP \to ATP}$ (Table S4). The reaction conditions were also determined by protein concentrations and diffusion coefficients. Our model also assumes that the concentrations of total MinD and MinE in the cell are fixed. We applied the experimentally estimated protein concentrations in this study as the total concentration of MinD in the simulation, which were contributed by cytosolic MinD-ADP ($c_{DD}$), cytosolic

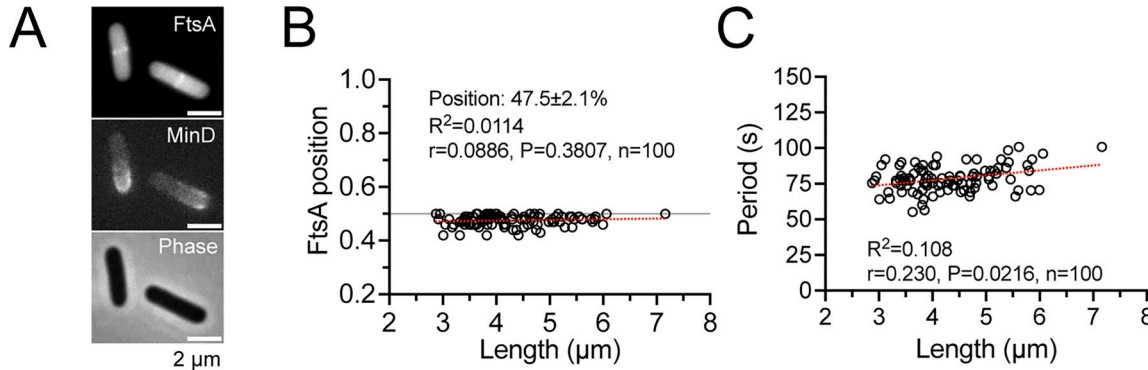

Figure 4. **Examination of FtsA localization, which marks the division site, in relation to cell length and MinD oscillation period in *E. coli* strain FW1541. (A)** Representative images demonstrating simultaneous observation of FtsA-mScarlet-I, sfGFP-MinD, and cell morphology. **(B and C)** The ring-like structure of FtsA-mScarlet-I had a relative position of 47.5 ± 2.1% from either pole (B), regardless of the cell length and the period of the MinD oscillations (C). The cell center is set at position 0.5 for normalization of the FtsA position. Pooled data from several image fields were used for the analysis. The Spearman nonparametric correlation method was used to calculate the correlation with 95% confidence intervals. The red dashed line represents the best-fit linear regression line. *n*, population size; $R^2$, goodness of fit; r, Spearman coefficient; P: two-tailed probability.

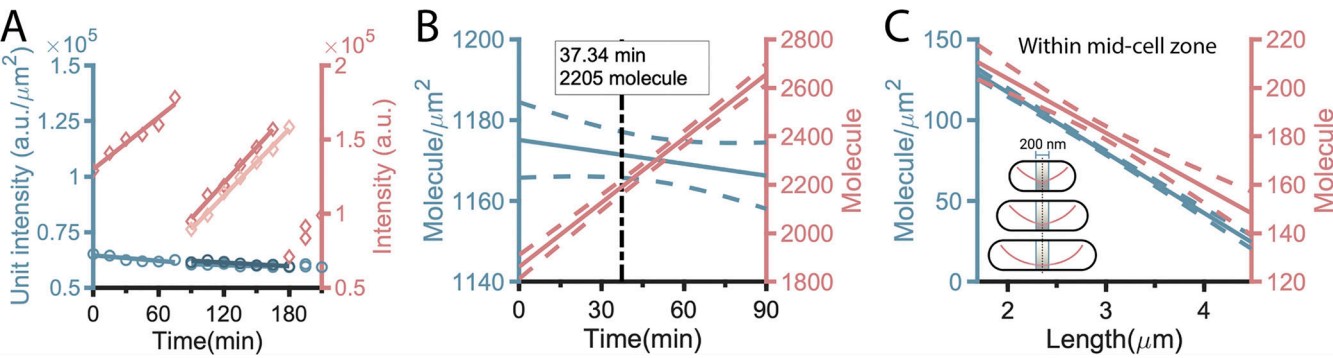

**Figure 5. Number of MinD molecules throughout the cell cycle. (A)** The bulk fluorescence intensity of sfGFP-MinD (diamond, red-hued lines) increased over time, while the intensity per unit area (circle, blue-hued lines) exhibited only moderate fluctuations. Data from paired daughter cells are distinguished by different colors. This plot includes partial data of a dividing cell from time-lapse image sequences acquired over a 5-h period at 15-min intervals. **(B)** The number of molecules in a single cell (red-hued line) and per unit area (μm²; blue-hued line) were plotted against time and fitted by simple linear regression (n = 100). Pooled data from several image fields were used for the analysis. The solid line represents the best-fit regression line, while the dashed lines represent the errors. Fast-growing cells that divided within five frames were excluded. **(C)** The number of molecules (red-hued line) and the number per unit area (μm²; blue-hued line) within 200 nm of the cell center were estimated based on the fraction from the MinD gradient profile (inset) before being plotted against time and fitted using simple linear regression (n = 100). The solid line represents the best-fit regression line, while the dashed lines represent the errors.

MinD-ATP ($c_{DT}$), membrane-bound MinD-ATP ($c_d$), and the membrane-bound MinD-MinE complex ($c_{de}$); and the total concentration of MinE, which was contributed by cytosolic MinE ($c_E$) and the membrane-bound MinD-MinE complex ($c_{de}$). The concentration is defined as the number of protein molecules per micron in this 1D model. Furthermore, each protein species is associated with a diffusion coefficient, including $D_D$ for the diffusion of cytosolic MinD-ADP and MinD-ATP, $D_E$ for the diffusion of cytosolic MinE, and $D_d$ and $D_{de}$ for the diffusion of the membrane-bound MinD-ATP and MinD-MinE complex (Tables 2 and S4). To randomly search for combinations of the parameter sets $k_{dD}$, $k_{dE}$, $k_D$, and $k_{ADP \to ATP}$, the following parameters

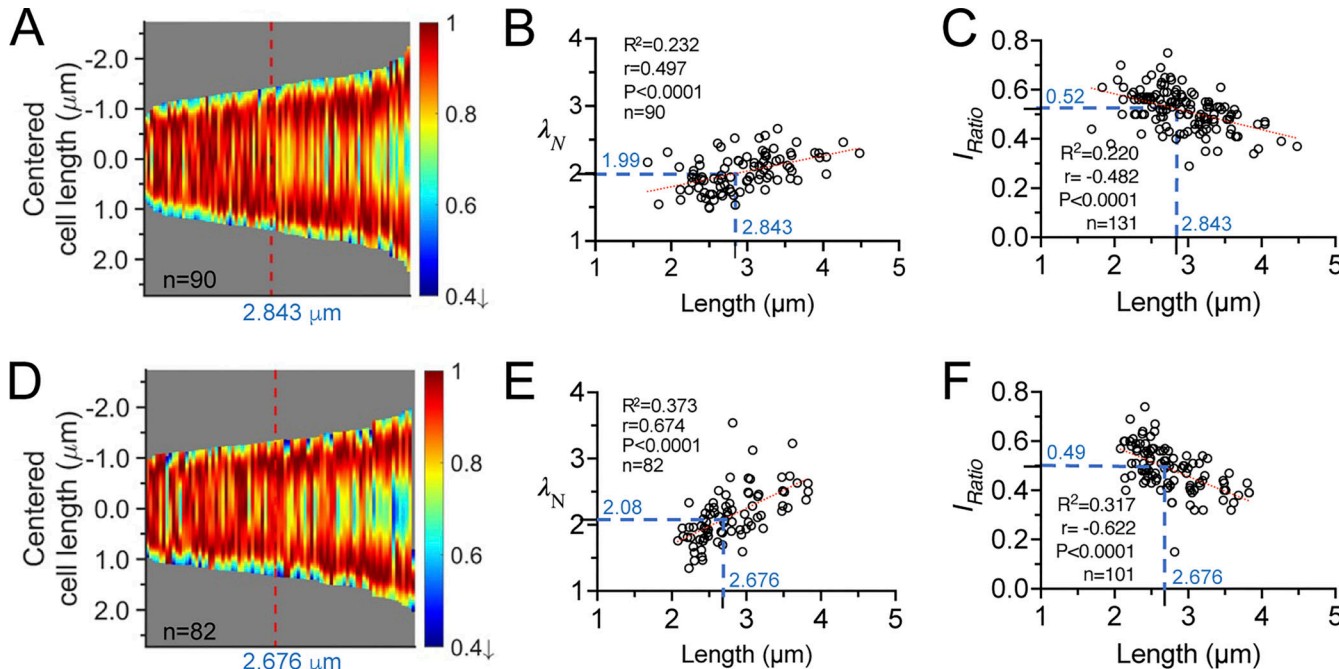

**Figure 6. Characteristics of the MinD concentration gradient. (A–F)** The cells were grown in 0.4% (A–C) or 0% glucose (D–F). **(A and D)** Kymographs of the fitted intensity profiles. The red line in each graph indicates the median cell length in the population. **(B and E)** The correlation plots of the wave slope ($\lambda_N$) against cell length. **(C and F)** The correlation plots of the wave slope ($I_{Ratio}$) against the cell length. In B, C, E, and F, the blue dashed line indicates the $\lambda_N$ or $I_{Ratio}$ value at the median cell length. The red dashed lines are obtained by simple linear regression. Correlations were calculated using the nonparametric Spearman correlation method with 95% confidence intervals. R², goodness of fit; r, Spearman coefficient; P: two-tailed probability; n, population size. This analysis was performed using the same image datasets as in Fig. 1, D–G, with an additional intensity threshold applied to filter the datasets for the estimation of $\lambda_N$.

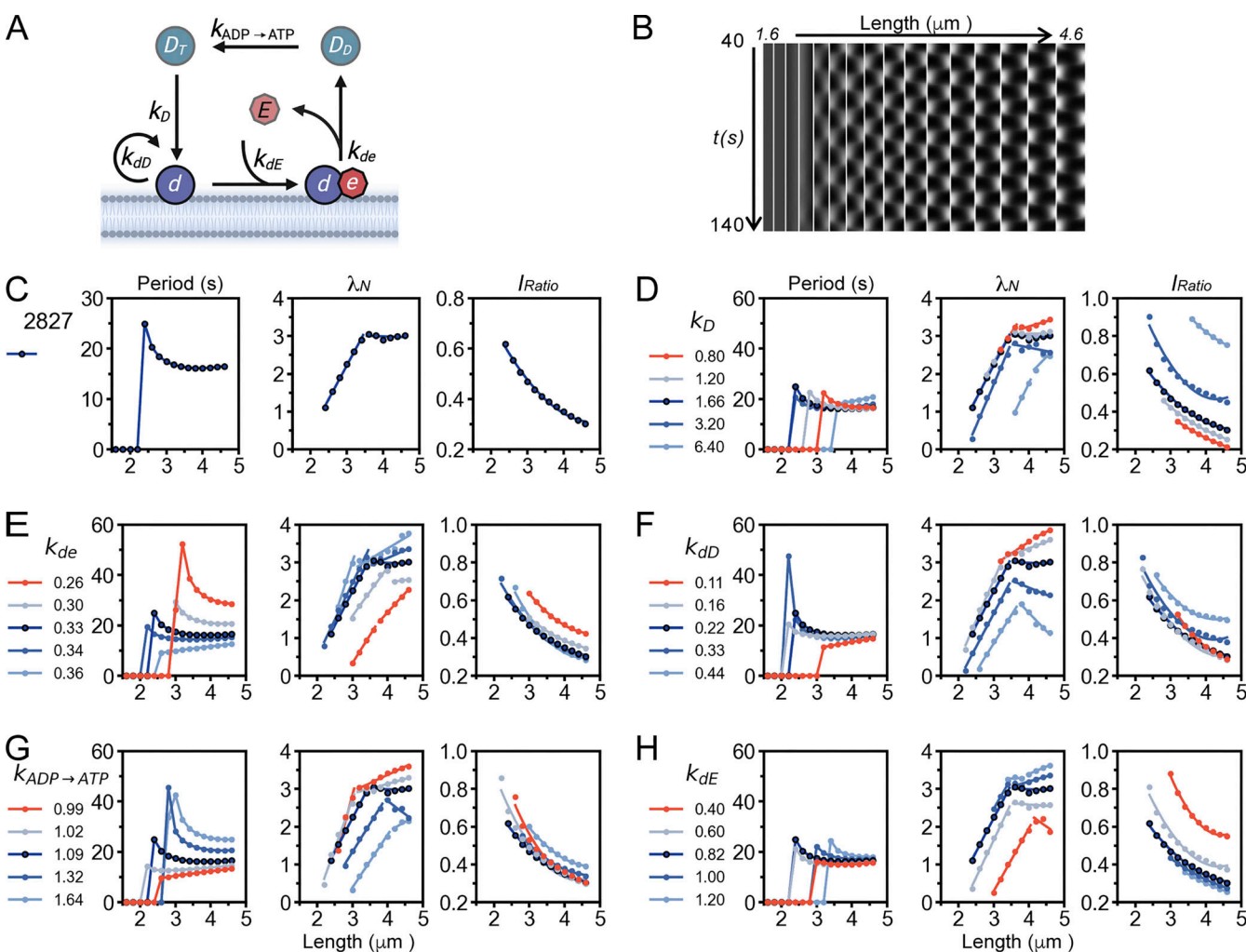

Figure 7. **Length-dependent characteristics of MinD concentration gradients in simulated cellular environments and the effects of different kinetic rate constants. (A)** Illustration of the reaction steps of Min oscillation and the corresponding reaction rate constants in the mathematical model. **(B)** Kymograph of the simulated MinD oscillation as demonstrated by parameter set #2827. **(C)** Oscillation period (left), $\lambda_N$ (middle) and $I_{Ratio}$ (right) of the simulated MinD oscillation, as demonstrated by parameter set #2827. **(D–H)** Influence of different rate constants on the oscillation period (left), $\lambda_N$ (middle) and $I_{Ratio}$ (right). D, $k_D$; E, $k_{de}$; F, $k_{dD}$; G, $k_{ADP \rightarrow ATP}$; H, $k_{dE}$.

were fixed in the simulation: the diffusion coefficients $D_d$ and $D_{de}$ were assumed values based on those of the bacterial membrane proteins (Schavemaker et al., 2018), the diffusion coefficients $D_D$ and $D_E$ were from Meacci et al. (2006), and the dissociation rate constant $k_{de}$ were from a previous simulation (Wu et al., 2015a). This operation allowed us to probe for the general behaviors of the system.

At the start of the simulation, MinD was concentrated in half of the cell, with half of the MinD on the cell membrane and the other half in the cytosol. MinE was evenly distributed throughout the cytosol and the no-flux boundary condition was applied. We collected 5,000 parameter sets, and these results were subjected to linear stability analysis, with the necessary condition that the most divergent linearized solution near a steady state was oscillatory in both time and space (Fig. 8 A). We performed simulations for 140 s at fixed concentrations at 16 different lengths from 1.6 to 4.6 μm (intervals of 0.2 μm) and investigated the kinetic properties for each length. Among them,

1,443 sets passed the criteria and showed spatiotemporal patterns. We applied additional filters to screen this result, including an oscillation period ≥15 s, a pole-to-pole oscillation pattern, and a $\lambda_N$ range of 1.2 ≤ $\lambda_N$ ≤ 3, which was approximated from the experiments. Since the experimental median cell length was 2.843 μm, we focused on the simulation data at lengths of 2.8 and 3 μm. This screening process reduced the parameter sets to 22 (Figs. S3, S4, and S5), including set #2827, which showed features similar to those of the experimental data as described in the following sections.

**Analysis of the simulation results: period, $\lambda_N$, and $I_{Ratio}$**
To ensure stable oscillation, we ignored the first 40 s of each selected case in the subsequent analysis. The intensity, $I(x,t)$, was defined as the concentration of membrane-bound MinD ($C_d$), and the MinD and MinE complex ($C_{de}$) normalized to the maximal value in both time and space. The oscillation period was identified as the average time difference between two

**Table 2. Values of the simulated parameter set #2827**

| Parameter | Unit | Value | Source |
|---|---|---|---|
| $k_D$ | 1/s | 1.66 | This study: Simulation |
| $k_{dD}$ | μm/s | 0.22 | This study: Simulation |
| $k_{dE}$ | μm/s | 0.82 | This study: Simulation |
| $k_{ADP \to ATP}$ | 1/s | 1.09 | This study: Simulation |
| $k_{de}$ | 1/s | 0.33 | Wu et al. (2015a) |
| $D_d$ | μm²/s | 0.2 | This study: Hypothetical |
| $D_{de}$ | μm²/s | 0.2 | This study: Hypothetical |
| $D_D$ | μm²/s | 16 | Meacci et al. (2006) |
| $D_E$ | μm²/s | 10 | Meacci et al. (2006) |
| $c_{DD} + c_{DT} + c_d + c_{de}$ | μM | 1.95 | This study: Experimental |
| $c_E + c_{de}$ | μM | 1.4 | This study: Experimental |

consecutive maximal $I$ value in the same grid from 40 to 140 s. Within a grid, only oscillation peak changes <5% were accepted. Otherwise, the oscillation appeared to be dampening. The intensity $I(x)$ identified from all time points was fitted with an exponential decay function to determine the spatial profile and calculation of $\lambda_N$:

$$I(x) = a\, e^{-\lambda_N x} + c \qquad (1)$$

The equation was fitted from the maximal value to the minimal value, with $a$ and $c$ being additional fitting constants. The relative position x on the normalized length scale may not always range from 0 to 1 since it depends on the locations of the maximal and the minimal values (Fig. 8 B). In some short cells, oscillation was observed only in time but not in space, where the same amount of membrane-bound MinD was obtained across the cell length. In these cases, $\lambda_N$ was set to zero. Only when the maximal $I$ value appears alternatively in either the left or right poles of a cell, we then identified $I_{max} = I(x = 0.0)$ and $I_{min} = I(x = 0.5)$ based on the merged profiles, allowing calculation of $I_{Ratio} = I_{min}/I_{max}$.

### *In silico* oscillation resembles oscillation in a cellular context

The simulation data reveal a continuous view of how concentration gradients evolve in a length-extending setting, which complements our experimental measurements of a population of cells associated with different lengths. We found from the simulation results of set #2827 that the initial oscillation period is slow but quickly accelerates to a relatively stable period, with only small changes as the length increases (Fig. 7, B and C, left). The initial phase of slow interpolar oscillations may be due to a transition from irregular to stable interpolar oscillations in vivo. Moreover, we found that $\lambda_N$ exhibited a biphasic trend, showing a gradual increase in the first phase and reaching an approximate value in the second phase (Fig. 7 C, middle), indicating that the concentration gradient gradually steepened and then stabilized at longer lengths. This scaling behavior, which was observed in the early stages of cell growth in simulation, was not seen in our experimental measurements using heterogeneous cell populations. Moreover, $I_{Ratio}$ showed a decreasing trend

(Fig. 7 C, right), indicating a gradual decrease in the relative MinD concentration at the midcell during elongation. We found that with a stable $\lambda_N$, $I_{Ratio}$ still drops with cell length since the peak value, a in Eq. 1, increases with a stable basal value, c in simulating longer cells. Mathematically, while the increase in $\lambda_N$ in the first phase is the nonlinear dynamic behavior of the system, the second phase shows a linear behavior. The former mimics a Turing model, in which the oscillation wavelength is determined by interactions between a local activator (MinD), a long-range repressor (MinE), and their diffusion coefficients. On the other hand, the oscillation wavelength doubles the cell length in the second phase.

The behavior of the increasing phase of $\lambda_N$ and the trend of $I_{Ratio}$ coincide with our in vivo measurements in a subset of cells under normal conditions (Fig. 6 B) that was exacerbated in carbon-starved cells (Fig. 6 E). Therefore, the simulation results not only support our experimental measurements but also reveal additional insights that will be discussed below.

### Effect of the kinetic rate constant on the MinD concentration gradient

Inspired by the analyses of these simulation results, we further investigated the impact of individual parameters on the oscillation period, $\lambda_N$, and $I_{Ratio}$ by varying the parameter values. First, we obtained a rough range for each kinetic constant from the 22 parameter sets that met the screening criteria (Fig. 8 A) and investigated the flexibility of the system by fine-tuning the rate constants with values approximate to those of set #2827. The range of each parameter was set under conditions that maintained typical pole-to-pole oscillation (Fig. 8, D a).

A general feature of the oscillation period is the appearance of an initial slow oscillation followed by switching to a relatively stable period, which shows a slight increase with increasing length. We found that $k_{de}$ and $k_{ADP \to ATP}$ more significantly affected the oscillation period after stabilization; $k_{de}$, $k_{ADP \to ATP}$, and $k_{dD}$ affected the period at the initial oscillation; and all rate constants affected the length at which the oscillation began. Interestingly, in contrast to the period that increases with decreasing $k_{de}$, the period that increases with increasing $k_{ADP \to ATP}$. This observation suggested that these two kinetic parameters have opposite effects on the system. We also found that all rate constants affected the length at which $\lambda_N$ showed a biphasic mode transition, i.e., from the increasing phase to the stable phase or mathematically, from the non-linear behavior to linear behavior. In the stable phase, $k_{dD}$ had the greatest impact on $\lambda_N$, which contrasted with other rate constants showing a mild increase with length in most tests. Moreover, $k_{dD}$, $k_{dE}$, and $k_D$ had greater impacts on $I_{Ratio}$ than did $k_{de}$ and $k_{ADP \to ATP}$. Furthermore, $I_{Ratio}$ increases with decreasing $k_{dE}$ and increasing $k_D$.

In summary, the three characteristics of MinD oscillations can be adjusted by varying the rate constants. The rate constants $k_{de}$ and $k_{ADP \to ATP}$ are the most critical for tuning the oscillation period; $k_{dD}$ is most critical for maintaining the stable phase of $\lambda_N$; and $k_D$, $k_{dE}$, and $k_{dD}$ are more influential for $I_{Ratio}$. Therefore, a balanced strength between all reaction rate constants is necessary to achieve normal interpolar oscillations.

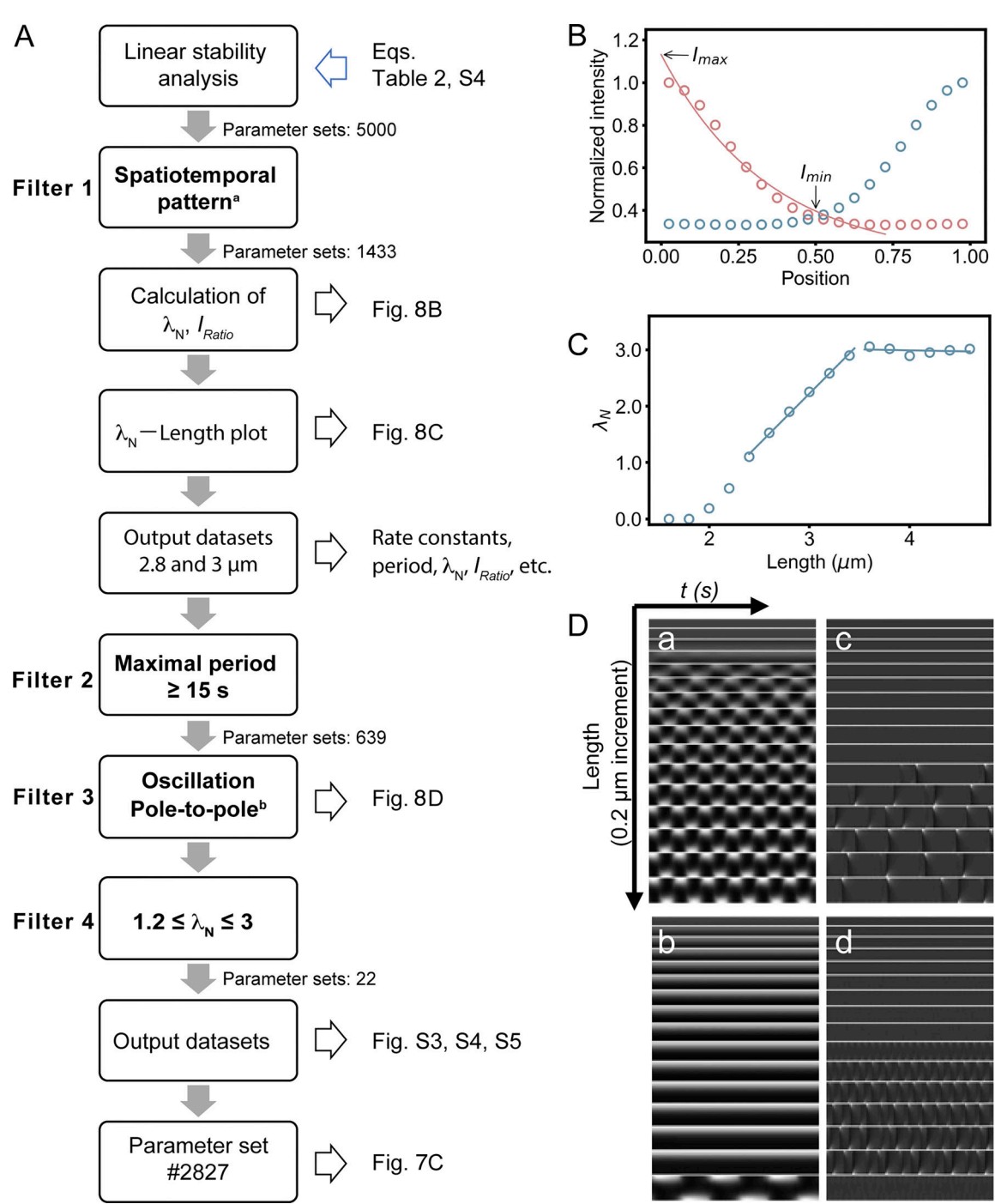

Figure 8. **Screening procedure of the parameter sets and analyses of the simulation results. (A)** Flowchart for screening oscillation features in the simulation results. There are four filters in this flowchart. Corresponding figures to illustrate or demonstrate the results are also labeled. [a] In practice, at least one length with a period greater than zero. [b] The set with only one length with a period greater than zero was removed. **(B)** Demonstration of the concentration profile of membrane-bound MinD ($C_d + C_{de}$) and identification of $I_{Ratio}$. The left intensity profile (red-hued hollow markers) was collected at 48.88 s when the maximal intensity occurred at the left pole. Each data point is normalized against the maximum intensity on this profile. Similarly, the right profile (blue-hued hollow markers) was collected at 73.9 s and processed. The resulting left and right profiles are symmetrical, with less than a 0.02% difference between them. The $I_{max}$ and $I_{min}$ values were identified from the fitted curve for calculation of the theoretical $I_{Ratio}$. Here, the profile is fitted from the maximum to the minimum intensity using an exponential equation $y = 0.96exp(-2.89x)+0.17$, with $\lambda_N = 2.89$. **(C)** Demonstration of the length-dependent changes in $\lambda_N$ and the biphasic mode. The incremental $\lambda_N$ values are fitted with a linear function $\lambda_N = aL + b$ or $\lambda_N = c$, where $L$ is the length. **(D)** Kymographs of oscillating membrane-bound MinD ($C_d + C_{de}$) between 40 and 140 s. (a) Typical pole-to-pole oscillation. (b) Oscillation starts late during simulation. (c and d) Irregular oscillation.

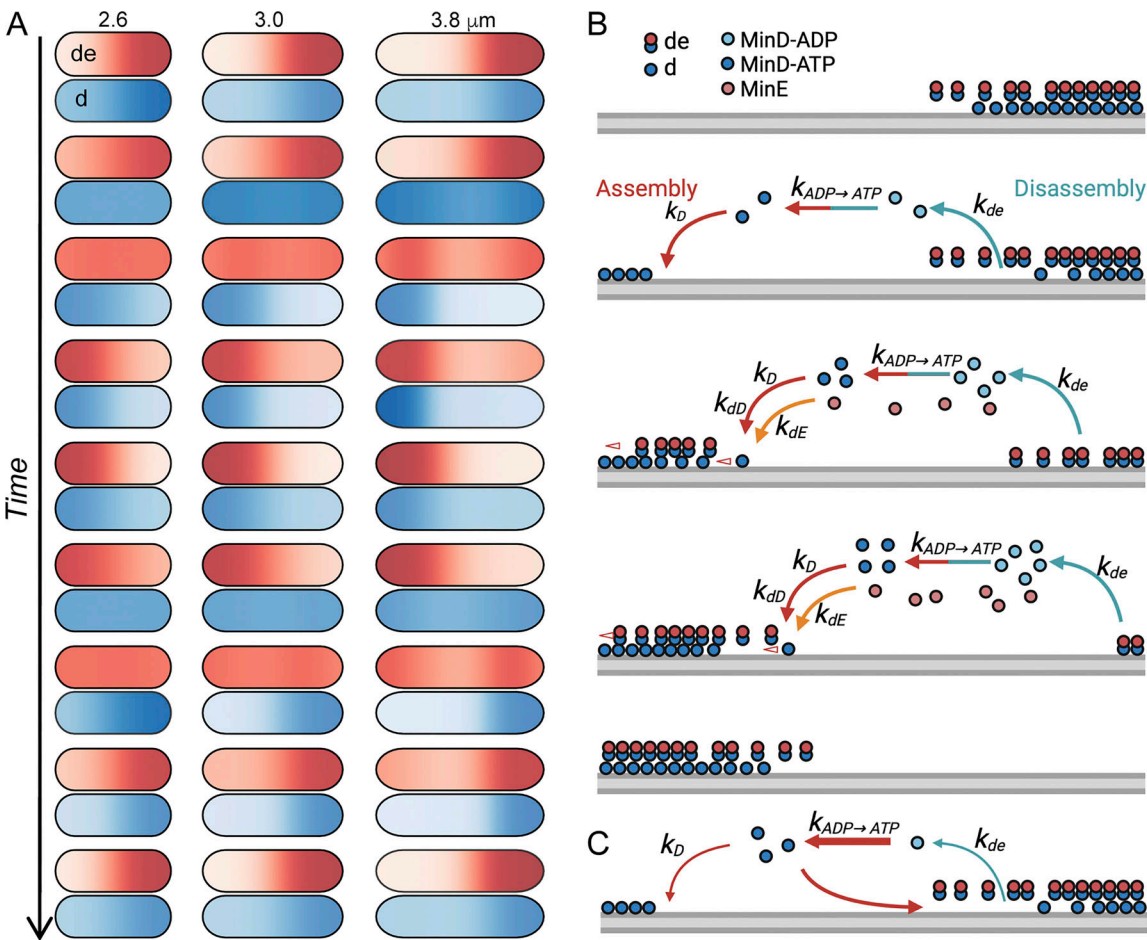

Figure 9. **The tunable MinD concentration gradient throughout the cell cycle. (A)** Spatial distribution of the membrane-bound MinD and MinE (de, top cell) and MinD (d, lower cell) in a complete oscillation cycle. The corresponding intensity plots generated in the simulation are shown in Figs. S6 and S7; and Video 1. **(B)** Kinetic steps driving pole-to-pole oscillations in a one-dimensional virtue cell. **(C)** Slow dissociation of the membrane-bound MinD and MinE or fast nucleotide exchange in cytosolic MinD can cause an inhibitory effect on pole-to-pole oscillation.

**Spatiotemporal distribution of the concentration gradient**

We conducted a detailed examination of the distributions of membrane-bound MinD (d) and the MinD-MinE complex (de) using set #2827 at different lengths to unravel the underlying mechanism of the length-dependent interpolar oscillations (Figs. 9, S6, and S7; and Video 1). Focusing on a half oscillation cycle, the assembly phase commenced with both species (d, de) predominantly located at the right pole and exhibiting the lowest concentration at the left pole (position 0). Initially, de significantly surpasses d at the right pole. While the concentration of de remains relatively constant for some time at the right pole, the concentration of d gradually decreases and starts to accumulate at the left pole. Subsequently, de decreases following the decrease in d at the right pole and reappears at the left pole, accumulating at the trailing end of d. Despite the increase in concentration, this pattern persists until d reaches its maximum value at the left pole (position 0), while de continues to accumulate more at the trailing end. As d starts to decrease at the left pole and spread in the opposite direction, de continues to migrate toward the pole (position 0), surpassing the concentration of d and reaching a peak before

undergoing reduction. This marks the completion of a half oscillation cycle.

The characteristics of MinD oscillations are summarized in Figs. S6 and S7, and they are less clear for shorter lengths that correspond to smaller $\lambda_N$ and larger $I_{Ratio}$. Our model also revealed the characteristic behavior of MinE, positioning itself at the trailing edge of the shrinking polar zone and advancing with MinD disassembly (Figs. 9 A and S6; and Video 1). Simultaneously, while cytosolic MinD-ADP (DD) is distributed at the shrinking end, the cytosolic variants MinD-ATP (DT) and MinE (E) have diffused to the opposite end and are likely primed for the reinitiation of MinD assembly (Fig. S7 and Video 1).

**Comparison with previous studies**

In this section, we compare the experimental works and mathematical models of Meacci and Kruse (2005), Fischer-Friedrich et al. (2010), Wu et al. (2015a), and our study.

*Comparison of live cell experiments*

Differences in bacterial strains and culture conditions, which can significantly affect protein expression levels, likely

contributed to the varied observations across studies. Our study used the *E. coli* strain FW1541, designed by Wu et al. (2015a), to allow endogenous expression levels, simplifying the study of MinD oscillations in living cells. In contrast, Meacci's studies (Meacci and Kruse, 2005; Meacci et al., 2006) used GFP-MinD expressed in the strain JS964 (MC1061 *malP::lacI_q Δmin::kan*) (Pichoff et al., 1995) to investigate MinD oscillations, though further details on GFP-MinD expression are unavailable for direct comparison. Additionally, their focus on protein mobility in Meacci et al. (2006) differs from ours.

Fischer-Friedrich et al. (2010) examined the transition of MinD oscillations from stochastic switching to regular pole-to-pole oscillation using the strain JS964 with plasmid pAM238 carrying *gfp-minD* and *mnE* under the *lac* promoter (Hu et al., 1999). In summary, while some observations are similar, differences in experimental design and research goals make direct comparisons difficult. Instead, we highlight the experimental rigor and unique insights each study offers into the Min system.

### Model comparison with Meacci and Kruse (2005)
The key differences between the Meacci model (Meacci and Kruse, 2005) and ours are in chemical kinetics, despite sharing similar assumptions regarding reaction steps and one-dimensional spatial reduction. The Meacci model assumes a homogeneous cytosolic distribution due to rapid diffusion, facilitating oscillations by integrating the dynamic equations of MinD and MinE on the membrane. In contrast, our model introduces specific diffusion coefficients for cytoplasmic MinD ($D_D$ = 16 μm²/s) and MinE ($D_E$ = 10 μm²/s), which are larger than the membrane-bound MinD diffusion ($D_d$ = 0.2 μm²/s), allowing us to capture cytoplasmic inhomogeneities.

Additionally, the Meacci model limits MinD membrane density to 1,000 molecules/μm, while our model imposes no such restriction, avoiding saturation effects. Our model also features a two-step MinD attachment process (spontaneous binding, $k_D$, and membrane-bound MinD-induced binding, $k_{dD}$), whereas the Meacci model employs a single-step attachment mechanism ($k_{dD}$). Another difference is in the rate of nucleotide exchange in MinD, which we set at 1.091/s, while the Meacci model assumes a much faster rate, omitting this reaction. Moreover, key rate constants in the Meacci model, such as $k_{dD}$, $k_{dE}$, and the detachment rate $k_{de}$, are smaller than those in our model, as shown in Table S4.

Lastly, while Meacci's modeling result shows an increase in oscillation period with cell length (Meacci and Kruse, 2005, Fig. 5 a), our result shows only a mild increase (Fig. 7 C), likely due to the nonuniform cytoplasmic distribution introduced by distinct diffusion coefficients.

### Model comparison with Fischer-Friedrich et al. (2010)
Fischer-Friedrich et al. (2010) observed that MinD oscillations in cells shorter than 2.7 μm exhibited stochastic interpolar switching, while in longer cells, oscillations became regular with less variation. They used "resident time at the pole" as a key metric in their experiments.

Using a 1D stochastic model based on Meacci and Kruse (2005), they found that stochastic switching in shorter cells

was due to the interplay of protein kinetics and the number of proteins. In this state, ATP consumption per MinD molecule was half that of the oscillatory state. The model, detailed in Table S4, excluded $k_{dD}$, assuming membrane-bound MinD aggregates via random diffusion.

### Model comparison with Wu et al. (2015a)
Although our model shares the same reaction steps and kinetic parameters as Wu et al. (2015a), there are key differences in construction. Our model explored the MinD concentration gradient during cell elongation but did not assess the effects of cell shape, confinement, or MinD nucleation. We simulated the Min system in a 1D lattice, while Wu used a 3D lattice, leading to different units for concentration and rate constants (Table S4). Additionally, the recruitment rate constant $k_D$ in our model is a first-order reaction, while Wu's model treats it as recruitment involving only cytosolic MinD-ATP near the membrane. Our model sums cytosolic concentrations across each 0.2-μm segment, whereas Wu's only considers the concentration near the membrane. Rate constants $k_{dD}$ and $k_{dE}$ also differ due to dimensional considerations, but first-order constants like $k_{ADP \to ATP}$ and $k_{de}$ remain comparable. Overall, discrepancies between the models make direct comparisons difficult.

## Discussion
This study investigated how the spatiotemporal distribution of MinD, in the form of a concentration gradient, influences Min protein oscillations during cell elongation (Figs. 9 A and S6). We found that the MinD concentration gradient became steeper as the cells elongated (Fig. 6 B), implying that the MinD concentration at the midcell decreased with length (Fig. 6 C). This length-dependent regulation of the MinD concentration may affect FtsZ ring formation via MinC. Although the mean cellular concentration of MinD is independent of cell length (Fig. 5 B), the MinD concentration at the midcell is inversely proportional to cell length (Fig. 5 C). This result indicates the importance of the shape of the concentration gradient rather than the concentration itself. In addition, we found that the oscillation period of MinD remained relatively consistent in cells of different lengths (Fig. 1 E). Similar oscillation periods and concentration gradients were observed for cells grown under carbon stress conditions (Figs. 1 G; and 6, E and F), although the average cell length was shorter. The observed phenomenon of relatively stable oscillation periods, accompanied by increasing velocity as cells elongate, may be partly attributed to the physical properties of the Min system, as well as potential unknown regulatory mechanisms that could adjust the kinetic rate constants of the system.

In this study, the MinD concentration gradient was characterized by the gradient slopes ($\lambda_N$) and the ratio ($I_{Ratio}$) between the minimum and maximum values in the gradient (Fig. 2). Since the minimum and maximum intensities were measured from the concentration gradient characterized by $\lambda_N$, a direct correlation between them exists. That is, a larger $\lambda_N$ will result in a smaller $I_{Ratio}$ and vice versa. Furthermore, while $\lambda_N$ showed an increasing trend with increasing length in a population of

cells (Fig. 6 B), $I_{Ratio}$ showed a gradual reduction with increasing length (Fig. 6 C). The same trend was identified for cells cultured with rich and poor carbon sources. However, cells grown in the absence of glucose showed more drastic changes in $\lambda_N$ within a shorter length range, which is also coupled with more drastic changes in $I_{Ratio}$. This property may necessitate a faster reduction in the concentration gradient to reach the effective $I_{Ratio}$ for cells dividing at shorter lengths. We hypothesize that there may be an effective $I_{Ratio}$ that is low enough for stable FtsZ ring formation. Based on our results, this effective $I_{Ratio}$ can occur at any cell length since with a $I_{Ratio}$ value of ~0.5, the corresponding cell lengths under both carbon-rich and starvation conditions differed greatly. As a result, by adjusting $\lambda_N$ as a function of length, the steepness of the $I_{Ratio}$ reduction can be altered. As further supported by numerical simulations (discussed below), this property appears to be intrinsic to the Min system and is the result of the complex interplay between the dynamic molecular interactions and diffusion (Figs. 7 and 9).

We used a 1D model to numerically replicate experimental findings and explored the kinetic rate constants and molecular interaction parameters that influence the stable period, steeper concentration gradient, and reduced MinD concentration at the midcell in elongated cells. While the results are promising, the current simulation used a minimal model with key steps to generate oscillations but showed shorter oscillation periods (Fig. S3). For future work, incorporating more complex models with additional steps, such as unknown modulators of $k_{de}$ and $k_{ADP \to ATP}$, and introducing stochasticity in reaction and diffusion, may improve the model's alignment with actual measurements.

In the simulation, we fixed the protein concentrations, diffusion coefficients, and $k_{de}$ to randomly search for combinations of four kinetic rate constants $k_D$, $k_{dD}$, $k_{dE}$, and $k_{ADP \to ATP}$, which—that generated results resembling the oscillation features in vivo by screening through filters based on experimental measurements (Fig. 8). This process led to the identification of parameter set #2827, which produced features similar to experimental measurements, including the oscillation period, concentration gradient slope $\lambda_N$, and relative midcell concentration $I_{Ratio}$ (Figs. 7 C, S3, S4, and S5). The simulated $\lambda_N$ shows a biphasic nature during elongation, revealing a scaling behavior in the shorter length. This may be advantageous, particularly in newborn cells or cells under stress conditions, where a faster reduction of $\lambda_N$ with length may facilitate the Min system to switch from irregular to regular oscillations. Our simulation results provide insight into the kinetic aspects of molecular interactions that mediate the length dependence of the MinD concentration gradient and the slight change in the oscillation period. The spatial distribution of the MinD–MinE complex and MinD alone on the membrane, along with the kinetic rate constants at different oscillation stages are detailed in Fig. 9 B.

We found that differences between the five rate constants ($k_D$, $k_{dD}$, $k_{dE}$, $k_{de}$ and $k_{ADP \to ATP}$) of the parameter set #2827 are within one order of magnitude (Table S4), making them kinetically compatible in the reaction. This parameter set allows for maximum utilization of protein molecules to drive membrane-associated oscillations. Using this parameter set, we further

tested the impact of different parameters on Min oscillations by fine-tuning the individual constants. This investigation revealed the delicately balanced strengths between them (Fig. 7, D–H). Notably, the rate constants, $k_{de}$ and $k_{ADP \to ATP}$, which govern MinD and MinE detachment from the membrane and nucleotide exchange, respectively, significantly influenced the oscillation period and the gradient slope $\lambda_N$. As shown in Fig. 9 C, a longer oscillation period may occur with a small $k_{de}$ when dissociation of the membrane-bound MinD and MinE slows. Additionally, maintaining a balanced recharging rate of cytosolic MinD ($k_{ADP \to ATP}$) is crucial for spatial distribution in oscillation dynamics. When recharging occurs too rapidly (greater $k_{ADP \to ATP}$), reattachment of the recharged MinD to the nearby membrane where it was detached could occur, thus slowing reassembly at the opposite pole. On the other hand, while membrane-bound MinD is greatly influenced by the recruitment rate constants of MinD and MinE to the membrane, $k_{dD}$ and $k_{dE}$, fast cytosolic diffusion allows MinD and MinE to be well buffered and to slowly vary in the cytosol. We observed that a larger $k_{dD}$ enhances the accumulation of MinD in the membrane, reduces $\lambda_N$, and increases $I_{Ratio}$ of the concentration gradient. A larger $k_{dE}$ decreases the accumulation of MinD, increases $\lambda_N$, and decreases $I_{Ratio}$.

Mechanistically, the rate constants $k_{de}$ and $k_{ADP \to ATP}$ are controlled at the protein structural level. We learnt from the literature that MinE shows different structural features and conformations that contribute to its interaction with MinD. The cytosolic latent form of MinE sequesters the MinD-interacting domain in a β-stranded conformation in the core region of the MinE dimer (Cai et al., 2019; Ghasriani et al., 2010; King et al., 2000). Upon contact with MinD or membranes, correlating with the rate constant $k_{dE}$, the MinD-interacting domain is exposed and refolded into an α-helix that interacts with the MinD dimer, forming the MinD–MinE complex on the membrane surface (Park et al., 2011; Wu et al., 2011). This process also involves a critical step where the N-terminus of MinE that is unstructured in the cytosol folds into an amphipathic helix when in contact with the membrane (Hsieh et al., 2010; Shih et al., 2011). In this step, MinD is in its ATP-bound form in the complex. Subsequently, this interaction stimulates the ATPase activity in MinD, causing ATP hydrolysis and conformational switches on the membrane surface. This eventually leads to the detachment of MinD and MinE, correlating with $k_{de}$, and then recycled back into the cytosol (Hu et al., 2002). At this step, MinD is in its ADP-bound form, which can be recharged with ATP for the next interaction cycle with MinE and the membranes (Fig. 7 A). Exactly how the nucleotide exchange takes place in the cytosol is unknown.

We speculate that under low glucose conditions, stress responses associated with restricted glucose conditions likely signal to the cell division process, including the Min system, through uncharacterized regulatory pathways. Such regulation may involve stabilizing and destabilizing factors of the MinD–MinE complex on the membrane, which changes the kinetic rate constant $k_{de}$ to alter the oscillation period. In other words, our simulation result, demonstrating the effects of varying $k_{de}$ on the oscillation period (Fig. 7 E), may reflect cellular conditions

influenced by stabilizing and destabilizing factors of the MinD–MinE complex on the membrane. Regulators of the Min system and the mechanism of nucleotide exchange in cytosolic MinD are currently underexplored.

Furthermore, our simulation results show that the interaction between MinD and MinE shifts the equilibrium of kinetic rate constants. A similar equilibrium shift has been reported in the literature by adjusting the MinD-to-MinE concentration ratio in both in vitro and in silico studies (Denk et al., 2018; Halatek et al., 2018; Mizuuchi and Vecchiarelli, 2018). In Vecchiarelli's in vitro studies (Mizuuchi and Vecchiarelli, 2018; Vecchiarelli et al., 2016), period maintenance was observed and linked to an increase in MinE concentrations, facilitating MinD release. In the cellular environment, uncharacterized regulators of the Min system that modulate the equilibrium between kinetic rate constants in response to environmental cues remain to be explored further. Candidates include MinC, which stabilizes the binding of MinD to the membrane at the poles (Ghosal et al., 2014; Zheng et al., 2014), and ClpXP, which degrades MinCD copolymers in vitro and impacts the mobility of MinD in the stationary-phase cells (LaBreck et al., 2021), and interactors of the Min proteins identified by Lee et al. (2016). This missing information on the regulators of the Min system that alter the dynamic equilibrium during oscillation cycles may also explain the differences between in vivo and in vitro studies.

In conclusion, this study reveals the inherent plasticity and adaptability of the Min system in orchestrating division site placement and the quantitative understanding of Min oscillations within the cellular environment. In addition, concentration gradients play a key role in various cellular processes that enable cells to regulate their internal conditions and perform functions critical to their survival in response to their environment; thus, the findings of this study are of broader relevance in cell biology.

## Materials and methods

### Strains and plasmids
Genotypic descriptions of the strains and plasmids used are provided in Table S1. The oligonucleotide sequences are provided in Table S2.

The *E. coli* strains MC1000 (*araD139 Δ(araABC-leu)7679 galU galK Δ(lac)X74 rpsL thi*; RRID: Addgene_71852) (Casadaban and Cohen, 1980) and DH5α (*ΔlacZ ΔM15 Δ(lacZYA-argF) U169 recA1 endA1 hsdR17(rK-mK+) supE44 thi-1 gyrA96 relA1*; RRDI: SCR_006368) (Taylor et al., 1993) were used for general cloning purposes. The strain BL21(DE3)/pLysS (*str. B F⁻ ompT gal dcm lon hsdSB(rB⁻ mB⁻) λ(DE3 [lacI lacUV5-T7p07 ind1 sam7 nin5]) [malB⁺]K-12(λS) pLysS [T7p20 orip15A] cat*; RRID: SCR_012821) (Studier and Moffatt, 1986) was used for protein production. Strain FW1541 (*ΔminD minE::sfgfp-minD minE::frt aph frt*) (Wu et al., 2015a) was derived from the laboratory strain W3110 (*F-λ-IN(rrnD-rrnE) 1 rph-1*; RRID: SCR_007682) (Bachmann, 1972). The SOT88 strain (W3110 *ΔminC minD minE*) was constructed following the standard procedures of the λ Red recombination method (Datsenko and Wanner, 2000). The plasmid pKD3 (RRID: Addgene_45605) was used as the template to generate the recombinant DNA

fragment using primers that carried homologous sequences 50 bp upstream of the ATG start codon of *minC* and the terminal 50 bp of *minE* as well as the primer sequences for pKD3. The PCR products were transformed into W3110/pKD46 competent cells, which were subsequently screened for recombinants. The procedure generated strain SOT87 (W3110 *ΔminC minD minE, cat*), which was cured for pKD46 (RRID: Addgene_45606), followed by removal of the *cat* cassette to generate the strain SOT88. The flanking DNA fragments of *minCDE* were PCR-amplified for sequencing to confirm the deletion of the three genes.

The plasmid pSOT370 ($P_{LtetO-1}$::*ftsA-mScarlet-I*) was generated using the Gibson assembly method (Gibson Assembly Master Mix, New England Biolabs) to ligate the DNA fragments of *ftsA* that were PCR-amplified from the genomic DNA, *mScarlet-I*, which was amplified from pEB2-mScarlet-I (Balleza et al., 2018), and pMLB1113-$P_{LtetO-1}$, which was the large fragment of pSOT329 digested with SmaI and HindIII ($P_{LtetO-1}$::*ftsZ^{G55}-mKO2-ftsZ^{Q56}*). The plasmid pSOT329 ($P_{LtetO-1}$::*ftsZ^{G55}-mKO2-ftsZ^{Q56}*) was modified from pSOT295 ($P_{lac}$::*ftsZ^{G55}-mKO2-ftsZ^{Q56}*) by replacing the *lac* promoter with the *LtetO-1* promoter from pdCas9 bacteria (Qi et al., 2013) (PRID: Addgene_44249). pSOT295 was created by replacing *mCerulean* in pSOT294 ($P_{lac}$::*ftsZ^{G55}-mCerulean-ftsZ^{Q56}*) with *mKO2*, which was amplified from pSOT291 ($P_{lac}$::*ftsZ-mKO2*). pSOT291 originated from pSOT157 ($P_{lac}$::*ftsZ-yfp*) after *yfp* was replaced with *mKO2* amplified from FW2454 (*hupA-mKO2*) (Wu et al., 2015b). pSOT157 was constructed by three-fragment ligation between the XbaI and HindIII fragments from pMLB1113 (de Boer et al., 1989), the XbaI-*ftsZ*-BamHI fragment that was amplified from the *E. coli* genome by PCR and subjected to restriction digestion, and the BamHI-*yfp*-HindIII fragment from pYLS67 (Shih et al., 2002). The *mCerulean* gene was amplified by PCR from pJSB-FtsZ^{G55}-mCerulean-FtsZ^{Q56} (Moore et al., 2016).

For the construction of pSOT279 ($P_{lac}$::*his_{6x}-sfgfp-minD*), *sfgfp-minD* was amplified from pBVS4 ($P_{lac}$::*sfgfp-minD minE*) (Wu et al., 2015a), and the *his_{6x}* tag and restriction sites NheI and BamHI were introduced using primers. The resulting PCR product was restriction digested and ligated into pET21a (Novagen; RRID: Addgene_69745).

### Growth conditions
Each bacterial culture was grown from a single colony in a minimal medium containing M9 salts (Difco), 0.25% casamino acids, 2 mM MgSO₄, and 0.1 mM CaCl₂ supplemented with 0.416% glucose at 30°C. The following concentrations of antibiotics were used for plasmid selection: 50 µg/ml ampicillin, 50 µg/ml kanamycin, and 34 µg/ml chloramphenicol. An overnight culture was used to inoculate fresh medium of the same kind to an $OD_{600 nm}$ value of ~0.05, which was subsequently allowed to grow at 30°C until the $OD_{600 nm}$ reached between 0.3 and 0.4. The cells were washed twice with M9 salt solution and resuspended in 30 µl of minimal medium supplemented with the desired concentration of glucose. 3 µl of cell suspension was then spotted on a 2% agarose pad containing 0.416% or 0% glucose on a glass slide to investigate the effect of glucose concentration on sfGFP-MinD oscillation.

To study FtsA localization, FW1541/pSOT370 cells were grown from a single colony in an M9 minimal medium

supplemented with 0.416% glucose and ampicillin at 30°C. FtsA was studied in place of FtsZ because of the abnormal morphology caused by the overexpression of *ftsZ* or by engineering *ftsZ* on the chromosome. The overnight culture was used to spike the fresh medium and allowed to grow until the mid-log phase. The cells were spun down, washed, and diluted to an $OD_{600\,nm}$ of ~0.2 in the same medium. Then, 0.1 µM anhydrotetracycline (aTc) was added to induce the expression of *ftsA-mScarlet-I* from pSOT370. The culture was incubated for 2.5 h at 30°C before being spun down and resuspended in 30 µl of medium. 3 µl of cell suspension was spotted on an agarose pad containing M9 medium supplemented with 0.416% glucose and 0.1 µM aTc for imaging. After the agarose pad was sealed under a coverslip, the slide was placed on a preheated stage at 30°C installed on the microscope and allowed to stabilize for 10 min before image acquisition.

### Microscopy

Our microscopy system included an Olympus IX81 inverted microscope (Olympus) equipped with a CCD camera (ORCA-R2), an objective lens (UPlanFLN 100×, NA 1.30; Olympus), and filter sets for sfGFP (ET-Narrow Band EGFP filter set, cat. 49020; Chroma Technology Corporation), mScarlet-I (Semrock LF561-A-OMF; IDEX Health & Science; LLC Center of Excellence), and simultaneous imaging of both sfGFP and Alexa Fluor 647 in the immunofluorescence experiment (a quad band filter set [ET-391-32/479-33/554-24/638-31 Multi LED set, cat. 89402; Chroma Technology Corporation]). Images were acquired using Xcellence Pro software (SCR_000079; Olympus Corp) with an X-Cite 120 Metal Halide lamp (Excelitas Technologies Corp) or cellSens Dimension software (RRID: SCR_014551; Olympus Corp) with an X-Cite TURBO multiwavelength LED illumination system (Excelitas Technologies Corp).

Time-lapse images of sfGFP-MinD were acquired at 12-s intervals for 10 min or before the fluorescence diminished. The images were analyzed using the MicrobeJ plugin v5.11j (Ducret et al., 2016) (RRID: SCR_017116) installed in NIH ImageJ (Rasband, W.S., ImageJ, U.S. National Institutes of Health, Bethesda, MD, USA, RRID: SCR_003070) and Fiji (RRID: SCR_002285). As shown in Fig. 2, the time-lapse images were converted into kymographs compiled from the one-dimensional fluorescence intensity profiles generated by the projection of fluorescence onto the medial axes of the same cell. This measurement revealed the ensemble fluorescence of sfGFP-MinD in the membrane and the cytosol at the axial position. The oscillation period was measured from the kymograph. Only oscillation cycles with clear starting and ending points were used in the subsequent statistical analyses.

### Statistical analyses

Statistical analyses of the experimental data were performed using GraphPad Prism (version 8.3; GraphPad Software, Inc.; RRID: SCR_002798). Data collection is specified in the figure legends. The nonparametric Spearman correlation method was employed to calculate the correlation between paired data with a 95% confidence interval, yielding the correlation coefficient (r) and the two-tailed probability value (P). This analysis assumes that the data is at least ordinal in scale, that the variables represent paired observations, and that there is a monotonic relationship between the two variables without significant outliers. Where a simple linear regression equation was applied to generate a fitting line, the goodness of fit ($R^2$) is reported. To compare the shape characteristics of FW1541 and W3110, unpaired nonparametric Mann–Whitney tests were conducted to assess the differences in their distribution profiles. Cumulative samples collected from a random sampling of different imaging fields were analyzed by an experimenter blinded to the various treatments. The number of data points (n) and any data exclusions, when applicable, and the use of the median ± interquartile range, mean ± error, or mean ± standard deviation (SD) are specified in the figures, figure legends, or tables.

### Characterization and purification of the anti-MinD antiserum

The $Trx-His_{6x}$-MinD protein was overexpressed from the BL21(DE3)/pLysS/pSOT6 ($P_{lac}::trx-his_{6x}-minD$) strains and purified following the procedure described in Hsieh et al. (2010), Shih et al. (2019). The fusion protein was used to raise rabbit polyclonal antiserum by LTK Biolaboratories. A total of 50 ml of the final bleed was collected from the immunized rabbit to obtain the crude antiserum (RRID: AB_3662849) that was then purified against purified $Tx-His_{6x}$-MinD using a blotting method as follows. 30 µg of $Trx'-His_{6x}$-MinD were run on a 10% SDS–PAGE gel and transferred onto a polyvinylidene fluoride (PVDF) membrane (0.45 µm; Amersham GE Healthcare Europe GmbH) using a TE 22 Mighty Small Transphor Tank Transfer Unit (Amersham GE Healthcare Biosciences, Inc.). The membrane was stained with 0.5% Ponceau S to visualize the protein for excision of the membrane band containing $Trx-His_{6x}$-MinD. The membrane was washed three times with Tris-buffered saline containing Tween-20 (TBST) (20 mM Tris-Cl, pH 7.4, 150 mM NaCl, 0.05% Tween 20) and blocked in 10 ml of 5% bovine serum albumin (BSA) prepared in TBST. The membrane band was incubated with 0.5 ml of crude antiserum diluted in 9.5 ml of TBST at 4°C with gentle shaking overnight. The band was washed three times with 10 ml of TBST before the antibodies were removed from the blot using 1 ml of 0.2 M glycine/HCl (pH 2.5) and immediately neutralized with 1 ml of 1 M Tris-Cl, pH 9.0. The purified antiserum was validated against total cell lysates of FW1541 and W3110 as well as against $His_{6x}$-sfGFP-MinD and $Trx-His_{6x}$-MinD by western blotting, as shown in Fig. S1 A. The antiserum was divided into aliquots and stored in 10% (vol/vol) glycerol at −20°C.

### Characterization and purification of the anti-MinE antiserum

Overexpression and purification of the fusion protein $MinE-His_{6x}$ from the strain BL21(DE3)/pLysS/pSOT13 ($P_{lac}::minE-his_{6x}$) were performed following previously described procedures (Shih et al., 2019). The purified $MinE-His_{6x}$ was used to raise rabbit polyclonal antiserum by LTK Biolaboratories. A total of 50 ml of the final bleed was collected from the immunized rabbit to obtain the crude antiserum (RRID: AB_3662848) that was purified against the cell lysate of strain SOT88 to remove nonspecific contaminants, followed by purification against $MinE-His_{6x}$. In brief, 100 µg of

SOT88 cell lysate and 30 µg of MinE-His$_{6x}$ were separated on 15% Tris-Tricine gels and transferred independently onto PVDF membranes. The membrane was stained with 0.5% Ponceau S for visualization and excision of the membrane band containing the SOT88 cell lysate or MinE-His$_{6x}$. Both membrane bands were treated as described in the previous section.

Crude antiserum (0.5 ml) diluted in 9.5 ml of TBST was first incubated with the membrane band of SOT88 cell lysate with gentle shaking at 4°C for 8 h. After the removal of the first membrane band that absorbed nonspecific contaminants in the antiserum, the solution was incubated with the second membrane band containing MinE-His$_{6x}$ with gentle shaking at 4°C overnight. This membrane band was washed before stripping off the antibodies using 1 ml of 0.2 M glycine/HCl (pH 2.5) and immediately neutralized with 1 ml of 1 M Tris-Cl, pH 9.0. The purified MinEantiserum was validated against the W3110 and SOT88 total cell lysates as well as MinE-His$_{6x}$ by western blotting, as shown in Fig. S1 B. The antiserum was divided into aliquots and stored in 10% (vol/vol) glycerol at −20°C.

### Determination of the cellular protein concentrations by western blotting

One liter of cells at the exponential growth phase was spun down at 6,000 × $g$ for 20 min at 4°C and washed three times with 30 ml of 1× phosphate-buffered saline (PBS; 137 mM NaCl, 2.7 mM KCl, 10 mM Na$_2$HPO$_4$, 1.8 mM KH$_2$PO$_4$, pH 7.4). The cell pellet was resuspended in 20 ml of lysis buffer (50 mM Tris-Cl pH 7.5, 500 mM NaCl, 100 µg/ml lysozyme, 200 µg/ml DNase I, and 1 protease inhibitor tablet [cOmplete, EDTA-free Protease Inhibitor Cocktail; Roche]). This cell suspension was incubated on ice for 30 min before disruption using a NanoLyzer N2 (Gogene Corporation) with two passages at 15,000 psi. The clear lysate was recovered after centrifugation at 15,000 × $g$ for 10 min at 4°C and quantified using the Bio-Rad Protein Assay Kit (Bio-Rad Laboratories, Inc.).

The purified Trx-His$_{6x}$-MinD and His$_{6x}$-sfGFP-MinD fusion proteins were used to generate concentration standards for quantifying the MinD and sfGFP-MinD cellular concentrations in strains W3110 and FW1541. At least two batches of cell lysates of each strain and three independent repeats of each batch were analyzed. 40 µg of clear lysate along with a serial dilution of the purified His$_{6x}$-sfGFP-MinD or His$_{6x}$-MinD were separated on a 10% SDS–PAGE gel, followed by blotting onto a PVDF membrane. The membrane was blocked with 5% BSA in TBST (50 mM Tris-Cl, pH 7.4, 150 mM NaCl, 0.1% Tween 20) for 1 h at room temperature before incubation with the purified anti-MinD antiserum at a 1:100 dilution at 4°C overnight. The blot was washed three times with TBST and incubated with 10,000-fold diluted horseradish peroxidase-conjugated anti-rabbit IgG antibody (Amersham ECL rabbit IgG, HRP-linked whole Ab [from donkey]; Cytiva; RRID: AB_3662847) at room temperature for 1 h. After washing five times with TBST, the blots were treated with enhanced chemiluminescence (ECL) reagents (Amersham Biosciences ECL Select western blotting Detection Reagent; GE Healthcare). The bioluminescence signal was

detected using an ImageQuant LAS-4000 system (GE Healthcare Life Sciences).

The band intensity of the blots was quantified by NIH ImageJ. The measurements from serial dilutions of His$_{6x}$-sfGFP-MinD and Trx-His$_{6x}$-MinD were used to generate a calibration curve, allowing for the determination of sfGFP-MinD and MinD amounts in the lysate through simple linear regression equations in GraphPad Prism (RRID: SCR_002798). The cellular concentrations (M) of sfGFP-MinD and MinD were calculated by dividing the amount of protein in grams by the molecular weight (sfGFP-MinD: 56,817 Da; MinD: 29,614 Da), the number of cells in 40 µg of cell lysate, and the single-cell volume (Fig. S2 E). The number of protein molecules per cell was determined by multiplying the result by Avogadro's number.

The single-cell weight was determined from three independent exponentially growing cultures of FW1541 and W3110 (Table S3). Cells were collected from these cultures, freeze-dried overnight at 10 mTorr (using an UNISS Freeze Dryer FDM-20; Taiwan Green Version Technology Ltd.), and then weighed. The dry weights were 5.6 ± 0.7 mg for FW1541 and 5.2 ± 0.6 mg for W3110, equivalent to 0.112 and 0.104 mg/ml of culture, respectively. Assuming water constitutes ∼75% of cell weight (Bionumbers ID 105482; https://bionumbers.hms.harvard.edu/bionumber.aspx?id=105482), the wet weight per ml of culture was estimated to be 0.149 mg for FW1541 and 0.139 mg for W3110. Cell counts under the same conditions were 7.96 × 10$^8$ CFU/ml for FW1541 and 7.71 × 10$^8$ CFU/ml for W3110. Consequently, the single-cell weights were calculated to be 5.63 × 10$^{-13}$ g for FW1541 and 5.40 × 10$^{-13}$ g for W3110.

For determination of the MinE concentration, 60 µg of clear lysate along with a serial dilution of the purified MinE-His$_{6x}$ were separated on a 15% Tris-Tricine gel, followed by the procedures described above for MinD. The molecular weight of MinE (10,235 Da) was used in the calculation.

### Image processing

Image processing was performed using MATLAB R2018b (The MathWorks, Inc.; RRID: SCR_001622). The computer codes are available from the following repository site: https://zenodo.org/records/13927188, Ccsyan (2024).

### *Photobleaching normalization*

The intensity $I$ was collected as $\{I_j(x)\}_{j=1}^{n}$ with a time stamp $\{T_j\}_{j=1}^{n}$ associated with the individual intensity datum and the position $x \in L_j$, where $L_j$ is a set to collect the observation positions of $x$ and is normalized by setting $(L_j) = 0$ and $(L_j) = 1$ for each $j$. Further data processing shifts all $I_j(x)$ to a minimum value of 0 over $x$.

As the intensity has been observed to decay over time due to photobleaching, we scale the intensity function by a factor $exp(-\widehat{b}T_j)$, where $\widehat{b}$ is the fitted value of the decay rate $b$ in the model to describe the decay behavior due to photobleaching, resulting in $f(t) = a \cdot exp(-bt)$. We obtained $\widehat{b}$ by taking the regression of $f(t)$ values at time stamps $\{T_j\}_{j=1}^{n}$ on the total intensity $S_{T_j} = \sum_{x \in j} I_j(x)$ with $j = 1, ..., n$. Thus, we have $\tilde{I}_j(x) = \frac{I_j(x)}{exp(-\widehat{b}T_j)}$ for each $j = 1, \cdots, n$ after data processing.

### Algorithm for calculating center fraction

Let $\{\tilde{I}_{ij}(x)\}_{j=1}^{n}$ be the intensity calculated above after correcting for photobleaching, associated with the individual intensity data and position $x \in L_j = \{x_1, \cdots, x_k\}$, where $L_j$ is the set of observation positions for $x$ and is normalized as before with $0 = x_1 < x_2 < \ldots < x_k = 1$.

We interpolated the intensities $\{\tilde{I}_{ij}(x_a)\}_{j=1}^{n}$ and $\{\tilde{I}_{ij}(x_b)\}_{j=1}^{n}$ at $x_a = 0.5 - \frac{100\,nm}{particle\ length}$ and $x_b = 0.5 + \frac{100\,nm}{particle\ length}$ linearly with $x_1 < x_2 < \ldots < x_a < \ldots < x_b < \ldots < x_{k+2}$.

For each $j$, we calculated the center intensity $I_{j,center}$ by summing the trapezoidal areas between each interval $[x_l, x_{l+1}]$, where $x_a \leq x_l \leq x_{l+1} \leq x_b$, representing the intensity within the center plus or minus 100 nm. The total intensity $I_{j,total}$ is calculated by summing the trapezoidal areas between each interval $[x_l, x_{l+1}]$ for $l = 1, \ldots, k+1$. We then calculated the averages $\overline{I_{center}} = \frac{1}{20}\sum_{j=1}^{20} I_{j,center}$ and $\overline{I_{total}} = \frac{1}{20}\sum_{j=1}^{20} I_{j,total}$ and report the fraction as $\frac{\overline{I_{center}}}{\overline{I_{total}}}$ for each particle.

### Algorithm for $I_{Ratio}$

To quantify changes in $\tilde{I}_j(x)$ over $j$ (the time stamp index), which is expressed as $\{\tilde{I}_j(x)\}_{j=1}^{n}$, we introduced an index $I_{Ratio} = \frac{I_{min}}{I_{max}}$, where $I_{min}$ and $I_{max}$ are the minimum and maximum values of the summarized curve for $\{\tilde{I}_j(x)\}_{j=1}^{n}$. We split the data into two groups according to their left or right position by applying k-means analysis to the slope estimations at positions 0.2 and 0.8 and excluding those with small slopes within the first quartile range. We then applied the Gaussian kernel smoothing technique to these two position groups and output the corresponding summary curves $\widehat{f}_{Left}(x)$ and $\widehat{f}_{Right}(x)$, as well as $I_{max} = max_{i \in \{left, right\}} max_{x \in [0,1]} \widehat{f}_i(x)$.

To obtain $I_{min}$, defined as the intensity at the intersection of the left and right curves, we employed the exponential decay model for robust estimation. We applied the equations $c_{i,\lambda_i}(x) = d_i exp(-\lambda_i|x - m_i|)$ and $i \in \{left, right\}$ to fit these two groups of data by taking $\widehat{m}_i = max_{x \in [0,1]} \widehat{f}_i(x)$. We measured the overall decay rate by the weighted average $\lambda'$ of $\lambda_{left}$ and $\lambda_{right}$ for $\widehat{c}_{Left,\lambda'}$ and $\widehat{c}_{Right,\lambda'}$, in which the weights are the sizes for these two groups. That is, $\lambda' = \frac{\lambda_{left} \times n_{left} + \lambda_{right} \times n_{right}}{n_{left} + n_{right}}$. Thus, $I_{min}$ is the intensity of the intersection of $\widehat{c}_{Left,\lambda'}$ and $\widehat{c}_{Right,\lambda'}$. Then, $I_{Ratio} = \frac{I_{min}}{I_{max}}$ is calculated.

### Determination of the cellular concentration of sfGFP-MinD by fluorescence imaging

The cellular concentration of sfGFP-MinD was studied by culturing FW1541 in M9 minimal medium supplemented with 0.416% glucose, and the image acquisition and processing methods were performed as described earlier. Snapshots of both phase-contrast and fluorescence signals were acquired every 15 min for 5 h. The sum intensity in individual cells and the intensity per unit area ($\mu$m²) were corrected as described below and plotted against time to understand the changes in fluorescence intensity during the cell cycle (Fig. 2, A and B).

The number of protein molecules in the mid-cell zone was calculated based on the fluorescent intensity distribution in a gradient shape using the algorithm for calculating center fraction as described above rather than assuming a uniform distribution in the mid-cell zone. In brief, we identified the portion of the profile that falls within the mid-cell zone ($\pm100$ nm from the cell center) from fluorescent intensity profiles (in gradient form) of individual cells, as illustrated in Fig. 2. The fraction of intensity in the mid-cell zone was calculated by dividing the intensity within the mid-cell zone by the total intensity of the entire profile. Then, the number of MinD molecules in the mid-cell zone and their concentration (molecules/ $\mu$m²), as shown in Fig. 5 C, were obtained by multiplying the numbers presented in Fig. 5 B by this fraction.

The time-lapse fluorescence sequences were corrected for photobleaching. The intensity $I_j^{tot}$ measured from a cell is defined as a function of $k$, $I_j^{tot}(k)$, where $k$ refers to the number of rounds of photobleaching ($k \geq 0$) caused by light exposure applied to the $j^{th}$ cell ($j = 1, \cdots, n$). $I_j(k)$ is a decreasing function of $k$ since exposure to light can cause irreversible damage to the fluorophores, resulting in a gradual decrease in the fluorescence intensity. The normalized intensity, $\tilde{I}_j^{tot}(k) = \frac{I_j^{tot}(k)}{I_j^{tot}(0)}$, is obtained by dividing $I_j^{tot}(k)$ by its maximum $I_j^{tot}(0)$. Given $\{I_j^{tot}(k)\}_{j=1}^{n}$, we employed a biexponential curve model $f(k) = a_1 e^{-b_1 k} + a_2 e^{-b_2 k}$ to obtain the parameters $\widehat{a}_1, \widehat{b}_1, \widehat{a}_2, \widehat{b}_2$ to construct the reference curve $\widehat{f}(k) = \widehat{a}_1 e^{-\widehat{b}_1 k} + \widehat{a}_2 e^{-\widehat{b}_2 k}$. The image data obtained from the time-lapse sequence, $C_j(k)$, were corrected for photobleaching. The intensity of the $j^{th}$ cell after photobleaching correction was $\widehat{C}_j(k)$, where $\widehat{C}_j(k) = \frac{C_j(k)}{\widehat{f}(k)}$. A linear regression model was applied to the data obtained from individual cells using the following formula: $\widehat{C}_j(k) = \alpha_j(t_{kj} - t_{c_j^{(1)}j}) + \beta_j + \epsilon_k$, where $k = c_j^{(1)}, \cdots, c_j^{(2)}$. Here, $c_j^{(1)}$ is the first image data point of a cell acquired after the first cell division and $c_j^{(2)}$ is the last image data point of the same cell acquired before the next cell division. Therefore, $[c_j^{(1)}, c_j^{(2)}]$ is the number of image frames or data points between two rounds of cell division. $t_{kj}$ is the time when the $j^{th}$ cell undergoes the $k^{th}$ round of light exposure. Then, the overall trend for a population of cells ($n$) was estimated using $\widehat{\alpha} = \frac{1}{n}\sum_{j=1}^{n}\widehat{\alpha}_j$ and $\widehat{\beta} = \frac{1}{n}\sum_{j=1}^{n}\widehat{\beta}_j$.

### Modeling

A mathematical model was developed based on the rate constants of the following reactions to examine the MinD concentration gradient at different cell lengths (Figs. 8 A, S6, and S7). The computer codes are available from the following repository site: https://zenodo.org/records/13927188, Ccsyan (2024).

(1) The cytosolic MinD-ATP (MinD.ATP$_c$) is attached to the membrane and becomes a membrane-bound form (MinD.ATP$_m$):

$$\text{MinD.ATP}_c \rightarrow \text{MinD.ATP}_m.$$

(2) Recruitment of MinD.ATP$_c$ by MinD.ATP$_m$:

$$\text{MinD.ATP}_m + \text{MinD.ATP}_c \rightarrow \text{MinD.ATP}_m + \text{MinD.ATP}_m.$$

(3) Recruitment of cytosolic MinE (MinE$_c$) by MinD.ATP$_m$ and the formation of a membrane-bound MinDE complex (MinDE$_m$):

$$\text{MinD.ATP}_m + \text{MinE}_c \rightarrow \text{MinDE}_m.$$

(4) Dissociation of MinDE$_m$ from the membrane through ATP hydrolysis in MinD, resulting in MinD.ADP$_c$ and MinE$_c$:

$$\text{MinDE}_m \rightarrow \text{MinD.ADP}_c + \text{MinE}_c.$$

(5) Nucleotide exchange in MinD.ADP$_c$ and becoming MinD.ATP$_c$:

$$\text{MinD.ADP}_c \rightarrow \text{MinD.ATP}_c.$$

The chemical kinetics for the concentrations $c_x$, with $x = DD$, $DT$, $E$, $d$ and $de$, denoted MinD.ADP$_c$, MinD.ATP$_c$, MinE$_c$, MinD.ATP$_m$ and MinDE$_m$, respectively, and their diffusion are:

$$\frac{\partial}{\partial t}c_{DD} = D_{DD}\nabla^2 c_{DD} - k_{ADP\rightarrow ATP}c_{DD} + k_{de}c_{de} \tag{2}$$

$$\frac{\partial}{\partial t}c_{DT} = D_{DT}\nabla^2 c_{DT} + k_{ADP\rightarrow ATP}c_{DD} - (k_D c_{DT} + k_{dD}c_d c_{DT}) \tag{3}$$

$$\frac{\partial}{\partial t}c_E = D_E\nabla^2 c_E + (k_{de}c_{de} - k_{dE}c_d c_E) \tag{4}$$

$$\frac{\partial}{\partial t}c_d = D_d\nabla_m^2 c_d + (k_D c_{DT} + k_{dD}c_d c_{DT} - k_{dE}c_d c_E) \tag{5}$$

$$\frac{\partial}{\partial t}c_{de} = D_{de}\nabla_m^2 c_{de} + (k_{dE}c_d c_E - k_{de}c_{de}). \tag{6}$$

**Screening for the rate constants that generate MinD oscillations**

The kinetic simulation was conducted using a one-dimensional model, and Eqs. 2, 3, 4, 5, and 6 were numerically solved employing a simple finite difference scheme with a grid size of 0.2 μm. Parameter sets were generated by screening for a divergent oscillating solution in time and space via linear stability analysis. In linear dynamics, the eigenvalue with the largest real part prevails for a long time, and thus, an oscillation in time is predicted. Such an eigenvalue, which is sometimes called Hopf instability, has a nonzero imaginary part and a positive real part. When an oscillatory trial solution occurs in space, $e^{iqx}$, a Turing instability is predicted. To facilitate the search for parameters showing features similar to those of the Min oscillation, linear stability analysis along with the constraints of both Hopf unstable (in time) and Turing unstable (in space) conditions were employed. Namely, we calculated the steady state using its corresponding Jacobian matrix (Eq. 11), with q = π/L. Judging from the eigenvalues, the parameter sets from those with the most divergent component, namely, with a positive real part and a nonzero imaginary part, were kept for full numerical simulation using Eqs. 2, 3, 4, 5, and 6. Here, the most divergent solution in the linear approximation describes MinD oscillation between two cell ends with a wavelength of 2 L, where $L$ is the cell length under a least-oscillating condition that satisfies the no-flux boundary condition. Under these conditions, the wavevector q is set to π/L for these tests.

To search for parameters, the simulation was initially performed under a length of 3 μm and subsequently scanned through a length range from 1.6 to 4.6 μm. The concentrations of the MinD and MinE molecules were set to 2,205 (1.95 μM) and 1,580 (1.4 μM), respectively, at an experimental median cell length of 2.84 μm (Table 1). The concentration conversion between cytosolic and membrane locations became unnecessary, reducing the reaction-diffusion model to a 1D counterpart. Thus, the concentration in one dimension, $c_x^*$, is equivalent to the number of molecules divided by the length (molecule/μm). When simulating under different cell-length regimes, the number of MinD and MinE molecules were scaled up proportionally with a fixed concentration.

The rate constant $k_{de}$ was set at 0.33 (1/s) (Wu et al., 2015a), given the sensitivity of the dynamics to this parameter in time. Other reaction rates needed in the model were determined through random sampling. The diffusion coefficients for MinD and MinE in the cytosol were set at 16 and 10 μm$^2$/s, respectively, and in the membrane, they were 0.2 μm$^2$/s (Meacci et al., 2006). All fixed parameters are listed in Table S4. Initial tests were conducted using linear stability analysis, and parameter sets exhibiting both turning and Hopf instability were selected for further tests of the actual dynamics. A no-flux boundary condition was employed, and the simulation results were examined after 40 s. Eqs. 2, 3, 4, 5, and 6 were propagated using a simple finite difference scheme, with a time step set at 3.125 × 10$^{-5}$ s. To achieve numerical precision, the concentration change was kept under 5% with a spatial grid size of 0.2 μm. The four reaction rate constants, including $k_D$, $k_{dD}$, $k_{dE}$, and $k_{ADP\rightarrow ATP}$, were randomly searched as 10$^N$, with N being a random number drawn from a normal distribution with a mean of zero and a standard deviation of 3.0.

**Analysis of the simulation results**

To ensure stable oscillation, the first 40 s of each selected case were disregarded in the subsequent analysis. The intensity, denoted as $I(x,t)$, representing the concentration of membrane-bound MinD $C_d$ and $C_{de}$, was normalized to the maximum value in both time and space. The oscillation period was determined by averaging the time difference between two consecutive maximal values of $I$ within the same grid. This analysis was performed in the time window from 40 to 140 s.

*Linear stability analysis of the numerical model*

We performed linear stability analysis (Segel and Jackson, 1972; Turing, 1990) on the five-chemical reaction-diffusion model to determine the parameters. This analysis is based on Hopf bifurcation theory, which predicts a fixed point near oscillation. With diffusion, the uniform solution becomes unstable, and an oscillation pattern appears with a specific finite wavelength, a condition we aimed to investigate by finding suitable parameters.

For simplicity, we reduced the diffusion-reaction problem to one dimension, in which x denotes the position along the cell's long axis. Thus, Eqs. 1, 2, 3, 4, and 5, as described in the main text, can be rewritten in vector form:

$$\frac{\partial}{\partial t}\mathbf{u} = \mathbf{D}\frac{\partial^2}{\partial x^2}\mathbf{u} + \mathbf{f}(\mathbf{u}) \tag{7}$$

with the diagonal diffusion matrix $\mathbf{D}$ defined as:

$$\mathbf{D} = \begin{pmatrix} D_{DD} & 0 & 0 & 0 & 0 \\ 0 & D_{DT} & 0 & 0 & 0 \\ 0 & 0 & D_E & 0 & 0 \\ 0 & 0 & 0 & D_d & 0 \\ 0 & 0 & 0 & 0 & D_{de} \end{pmatrix}$$

and a vector function $\mathbf{f}(\mathbf{u})$ denoting the reaction expressions shown in Eqs. 1, 2, 3, 4, and 5.

For oscillation dynamics to occur, it is generally assumed that there is a uniform fixed point near the steady state, which is denoted as $\mathbf{u}^*$, with $\mathbf{f}(\mathbf{u}^*) = 0$. After Taylor expansion, we retained only the linear terms near the steady state $\mathbf{u}^*$ and rewrote Eq. 6 as a set of linear equations with a constant, A:

$$\frac{\partial}{\partial t}\delta\mathbf{u} = \mathbf{D}\frac{\partial^2}{\partial x^2}\delta\mathbf{u} + \mathbf{A}\,\delta\mathbf{u}, \tag{8}$$

where $\mathbf{A} = \partial\mathbf{f}/\partial\mathbf{u}$ is a 5 by 5 Jacobian matrix evaluated at $\mathbf{u}^*$. With Eq. 7 being a linear equation for $\delta\mathbf{u}$, a standard procedure is used in the following trial solution:

$$\delta\mathbf{u} = \delta\mathbf{u}_q e^{\sigma_q t}e^{iqx}, \tag{9}$$

where $\delta\mathbf{u}_q$ is an arbitrary initial condition expressed as a vector. Substituting the expression in Eq. 8 with $\delta\mathbf{u}$ in Eq. 7, we obtained the following eigenvalue problem:

$$\mathbf{A}_q\delta\mathbf{u}_q = \sigma_q\delta\mathbf{u}_q, \tag{10}$$

with the Jacobian $\mathbf{A}_q$ defined as:

$$\mathbf{A}_q = \mathbf{A} - \mathbf{D}q^2 = \begin{pmatrix} -k_{\text{ADP}\rightarrow\text{ATP}}-D_{DD}q^2 & 0 & 0 & 0 & k_{de} \\ k_{\text{ADP}\rightarrow\text{ATP}} & -k_D - k_{dD}c_d^* - D_{DT}q^2 & 0 & -k_{dD}c_{DT}^* & 0 \\ 0 & 0 & -k_{dE}c_d^* - D_E q^2 & -k_{dE}c_E^* & k_{de} \\ 0 & k_D + k_{dD}c_d^* & -k_{dE}c_d^* & k_{dD}c_{DT}^* - k_{dE}c_E^* - D_d q^2 & 0 \\ 0 & 0 & k_{dE}c_d^* & k_{dE}c_E^* & -k_{de} - D_{de}q^2 \end{pmatrix}. \tag{11}$$

Namely, the eigenvalue ($\sigma_q$) of $\mathbf{A}_q$ is solved as a function of $q$, which describes the stability of such a solution. When projected to its eigenvector, a positive real part of $\sigma_q$ indicates an increased deviation of $\delta\mathbf{u}_q$ with time, and a negative real part of $\sigma_q$ indicates a decreased deviation of $\delta\mathbf{u}_q$. Moreover, a complex value of $\delta\mathbf{u}_q$ is a solution for oscillation in time.

## Online supplemental material

Fig. S1 shows the quantification of sfGFP-MinD, MinD, and MinE. Fig. S2 shows the characteristic features of *E. coli* strains FW1541 and W3110. Fig. S3 shows period analyses of the simulated MinD dynamics. Fig. S4 shows concentration profile analyses, $\lambda_N$, of the simulated MinD dynamics. Fig. S5 shows concentration profile analyses, $I_{Ratio}$, of the simulated MinD dynamics. Fig. S6 shows concentration profiles of the membrane-bound species in an oscillation cycle for four different cell lengths. Fig. S7 shows concentration profiles of the cytosolic species in an oscillation cycle at four different cell lengths. Table S1 lists the strains and plasmids. Table S2 shows the list of oligonucleotides. Table S3 shows estimation of single-cell weight. Table S4 shows comparison of parameters among different studies. Video 1 shows concentration profiles of the membrane-bound species (top panel) and the cytosolic species (lower panel) in an oscillation cycle for four different cell lengths. Data S1 shows the measurement of time-lapse images showing MinD oscillations. Data S2 shows oscillation period and velocity before and after division. Data S3 shows cell lengths, oscillation periods, and positions of division sites labeled by FtsA. Data S4 shows the bulk fluorescence intensity of sfGFP-MinD, cell areas measured in individual growing cells, and estimation of number of MinD molecules through out the cell cycle. Data S5 shows measurements of $\lambda_N$ and $I_{Ratio}$ using the experimental datasets. Data S6 shows simulated reaction rate constants of MinD oscillation through screening procedures as illustrated in Fig. 7. Data S7 shows data of spatiotemporal distribution of different protein species in a complete oscillation cycle, including membrane-bound MinD (d), MinD-MinE complex (de) and their corresponding intensities. Data S8 shows data used to determine the number of MinD and MinE molecules in W3110 and sfGFP-MinD and MinE molecules in FW1541.

## Data availability

The plasmids, genetically engineered *E. coli* strains, crude antisera, and reagents generated in this study are available from the lead contact with a completed materials transfer agreement. The computer codes for image processing and modeling are available from the following code repository: https://zenodo.org/records/13927188, Ccsyan (2024).

## Acknowledgments

This article is dedicated to Dr. Lawrence I. Rothfield (1927–2022), a pioneer in the study of the Min system.

We thank Cees Dekker and Harold Erickson for providing plasmids and strains, Ester Malau for assistance in protein purification, and Jian Liu, Min Wu, Chien-Jung Lo, and Todd Lowary for comments and discussion. We acknowledge the technical supports from the NGS core at Academia Sinica.

This work was funded by the National Science and Technology Council, Taipei, Taiwan through grants to Y.-L. Shih (NSTC 113-2311-B-001 -013 -MY3, 111-2311-B-001-016, 110-2311-

B001-011, 108-2311-B001-012, and 106-2311-B001-009), I.-P. Tu (NSTC 106-2118-M-001-MY2), and C.-P. Hsu (NSTC 113-2123-M-001-001), and by Academia Sinica grants to I-P. Tu (AS-IA-110-M05) and C.-Y. Hung (AS-GCS-108-08). Open Access funding provided by the National University of Taiwan.

Author contributions: C.M. Parada: Data curation, Formal analysis, Investigation, Methodology, Writing - original draft, C.-C.S. Yan: Data curation, Formal analysis, Software, Validation, Visualization, Writing - original draft, C.-Y. Hung: Formal analysis, Methodology, Software, Validation, Visualization, Writing - original draft, I-P. Tu: Formal analysis, Funding acquisition, Methodology, Resources, Supervision, Writing - original draft, C.-P. Hsu: Conceptualization, Formal analysis, Funding acquisition, Methodology, Project administration, Resources, Writing - original draft, Y.-L. Shih: Conceptualization, Data curation, Formal analysis, Funding acquisition, Methodology, Project administration, Resources, Supervision, Validation, Visualization, Writing - original draft, Writing - review & editing.

Disclosures: The authors declare no competing interests exist.

Submitted: 18 June 2024

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

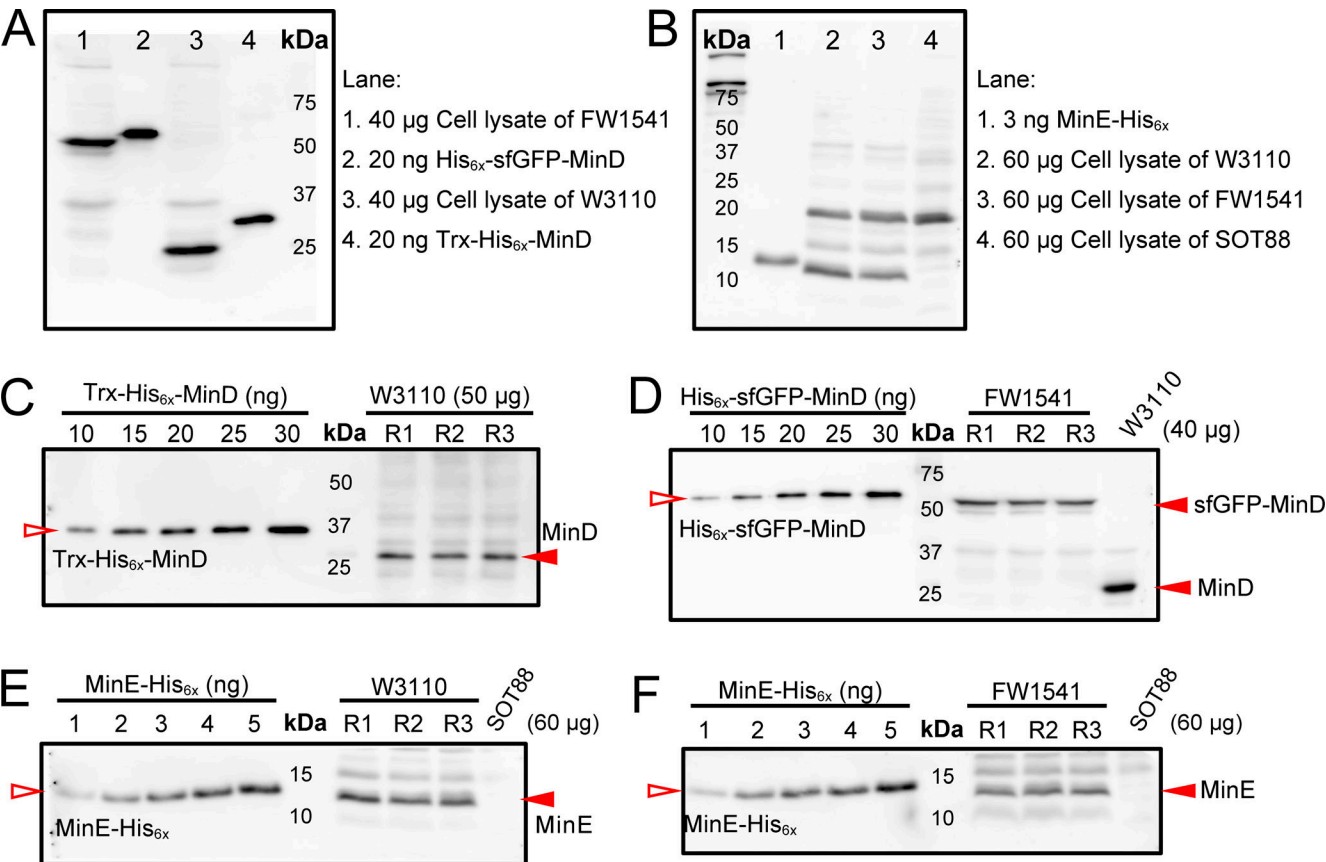

Figure S1.   **Quantification of sfGFP-MinD, MinD, and MinE. (A)** Validation of the ability of the anti-MinD antiserum to detect sfGFP-MinD and MinD in cell lysates alongside purified His$_{6x}$-sfGFP-MinD and Trx-His$_{6x}$-MinD. The blot-purified antiserum was used at a 1:100 dilution. **(B)** Validation of the anti-MinE antiserum ability to detect MinE in cell lysates alongside MinE-His$_{6x}$. The blot-purified antiserum was used at a 1:100 dilution. **(C and D)** An example of the western blots used to determine the concentrations of MinD in the cell lysates of the W3110 and FW1541 strains, respectively. Three independent cultures (sample repeats $n$ = 3) were collected for each western blot, and the experiment was repeated at least twice for each strain. **(E and F)** An example of the western blots used to determine the concentrations of MinE in the cell lysates of the W3110 and FW1541 strains, respectively. Three independent cultures (sample repeats $n$ = 3) were collected for each western blot, and the experiment was repeated five times for each strain. In C–F, serial dilutions of purified Trx-His$_{6x}$-MinD, His$_{6x}$-sfGFP-MinD, or MinE-His$_{6x}$ were applied to generate a calibration curve for interpolating the amount of MinD, sfGFP-MinD, or MinE in the sample by linear regression. Source data are available for this figure: SourceData FS1.

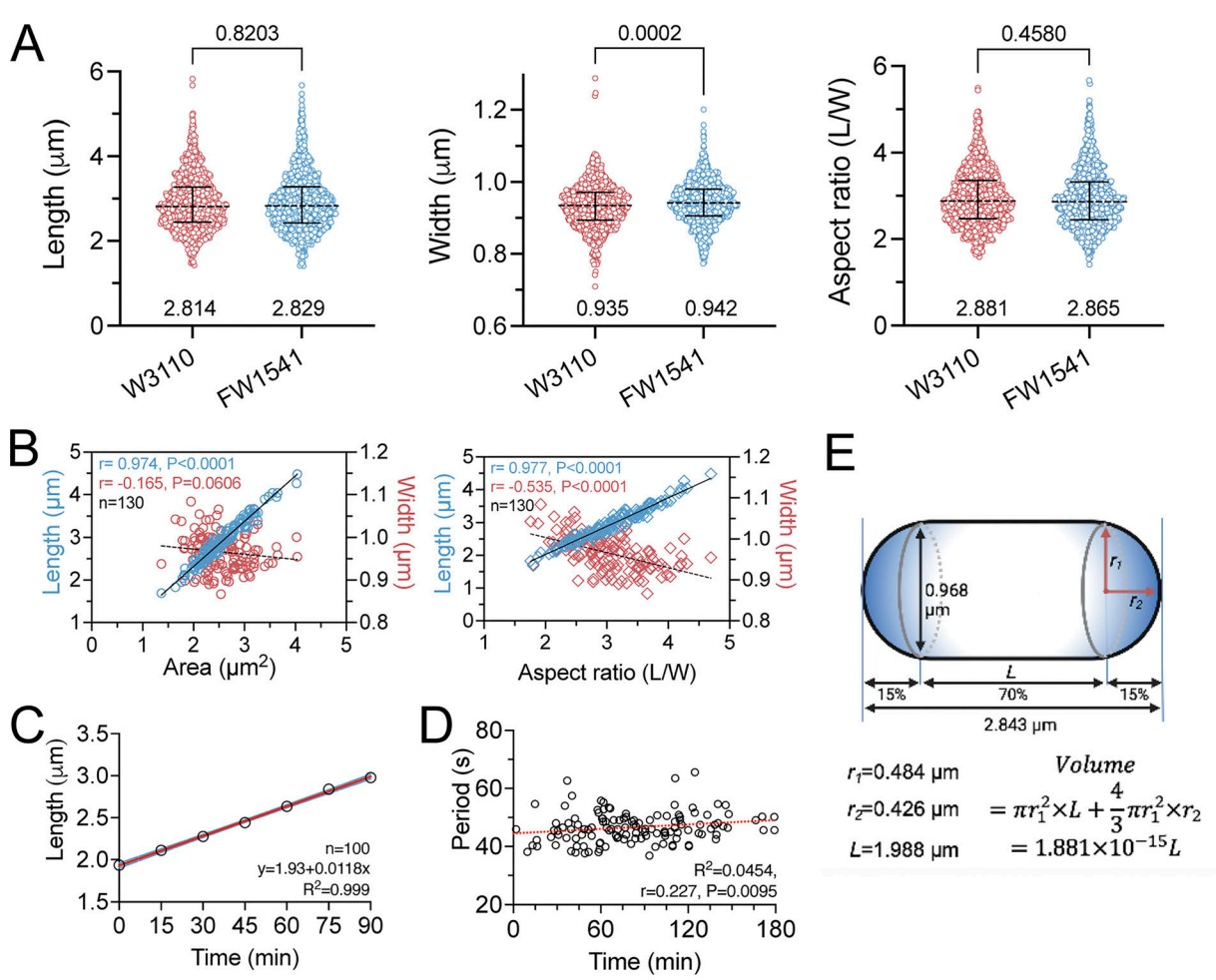

**Figure S2. Characteristic features of *E. coli* strains FW1541 and W3110 cultured in M9 medium supplemented with 0.416% glucose at 30°C.**
**(A)** Comparison of cell length, width, and aspect ratio. Data are presented as median values with interquartile ranges. Pooled data from several image fields were used for the analyses. Statistical comparisons were made using unpaired nonparametric Mann–Whitney tests. The median values are labeled and shown as the dashed lines, while the solid lines indicate the interquartile ranges. The P values of each pairwise comparison are labeled on top. W3110, $n$ = 943; FW1541, $n$ = 941. **(B)** Cell area (left) and aspect ratio (right), respectively, plotted against length and width. Correlations were calculated using the non-parametric Spearman correlation method with 95% confidence intervals. The same image dataset as in Fig. 1, D and E was used for the analysis. r, Spearman coefficient; P: two-tailed probability; $n$, population size. **(C)** A positive correlation between cell length and time was observed based on experimental data collected at 15-min intervals. The same image dataset as in Fig. 5, covering a complete cell cycle, was used for the analysis. Simple linear regression was applied to determine the length increase over time. The red line represents the best-fit regression line, while the blue dashed lines represent the errors. **(D)** A correlation plot of the period versus time was generated based on data from Figs. 1 E and S2 C. The Spearman nonparametric correlation method was used to calculate the correlation with 95% confidence intervals. The red dashed line represents the best-fit regression line. **(B–D)** $R^2$, goodness of fit; r, Spearman coefficient; P: two-tailed; probability. $n$, population size. **(E)** Illustration of the cell volume calculation of FW1541.

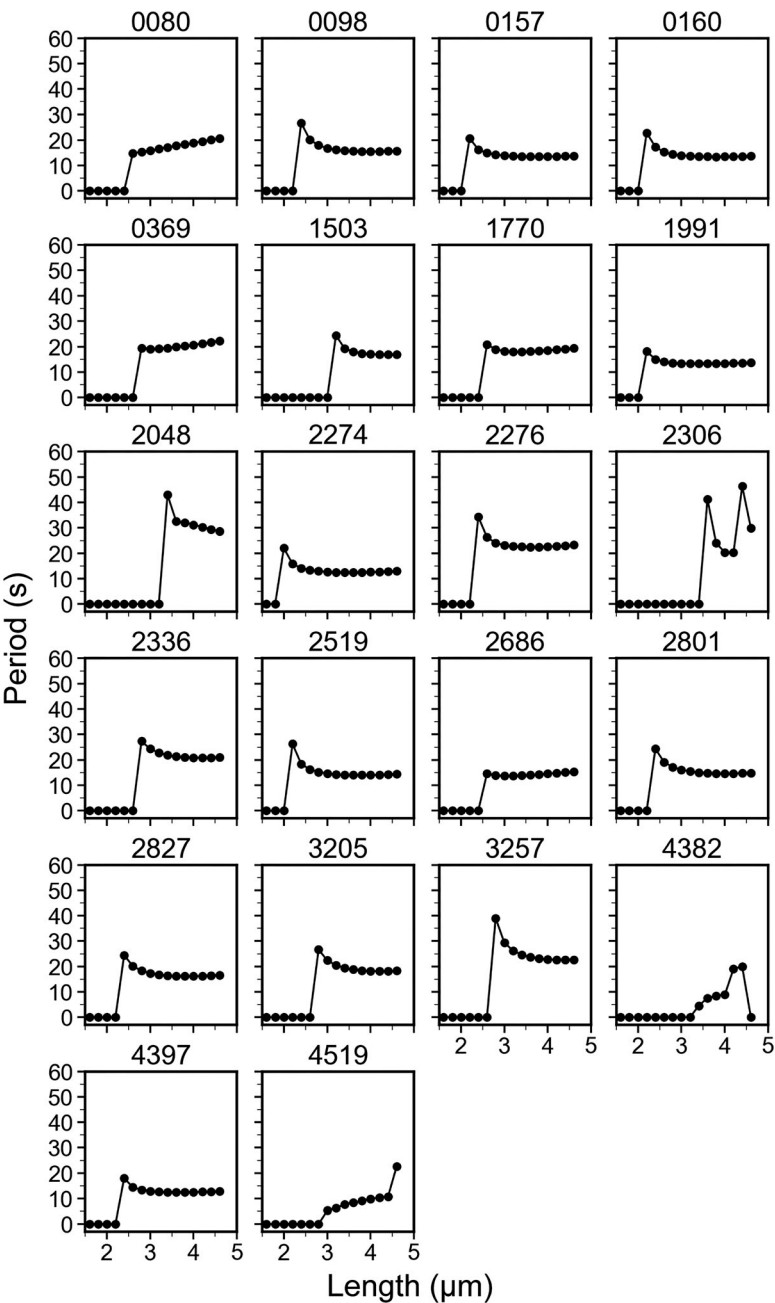

Figure S3.  **Period analyses of the simulated MinD dynamics.** This figure is supplemental to Fig. 7 C.

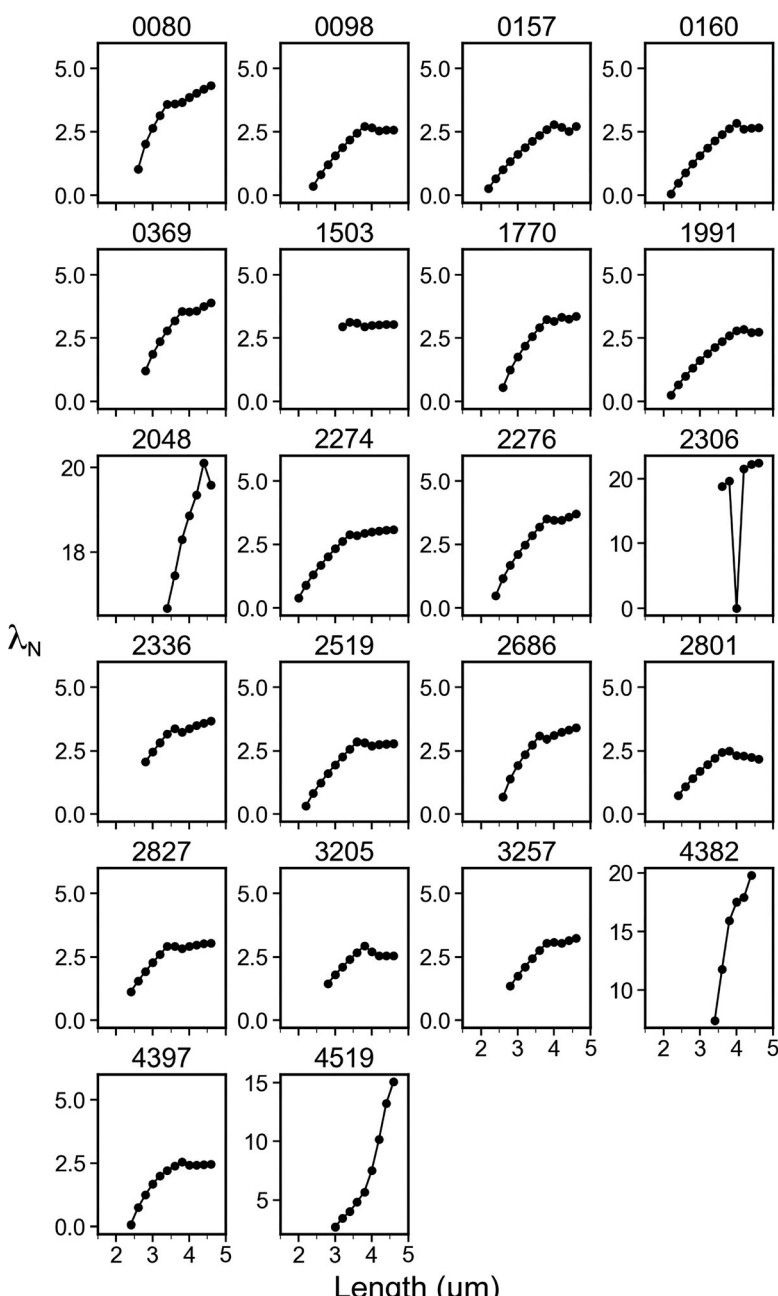

Figure S4. **Concentration profile analyses, λ_N, of the simulated MinD dynamics.** This figure is supplemental to Fig. 7 C. Data points for non-oscillatory cases (Period = 0) were omitted for both $\lambda_N$ (Fig. S4) and $I_{Ratio}$ (Fig. S5). Non-oscillatory cases are typically homogeneous, with trivial values of $\lambda_N$ and $I_{Ratio}$. Occasional non-oscillatory inhomogeneous cases occurred between two poles, resulting in non-trivial $\lambda_N$ and $I_{Ratio}$. However, these were unrelated to oscillation and have been removed.

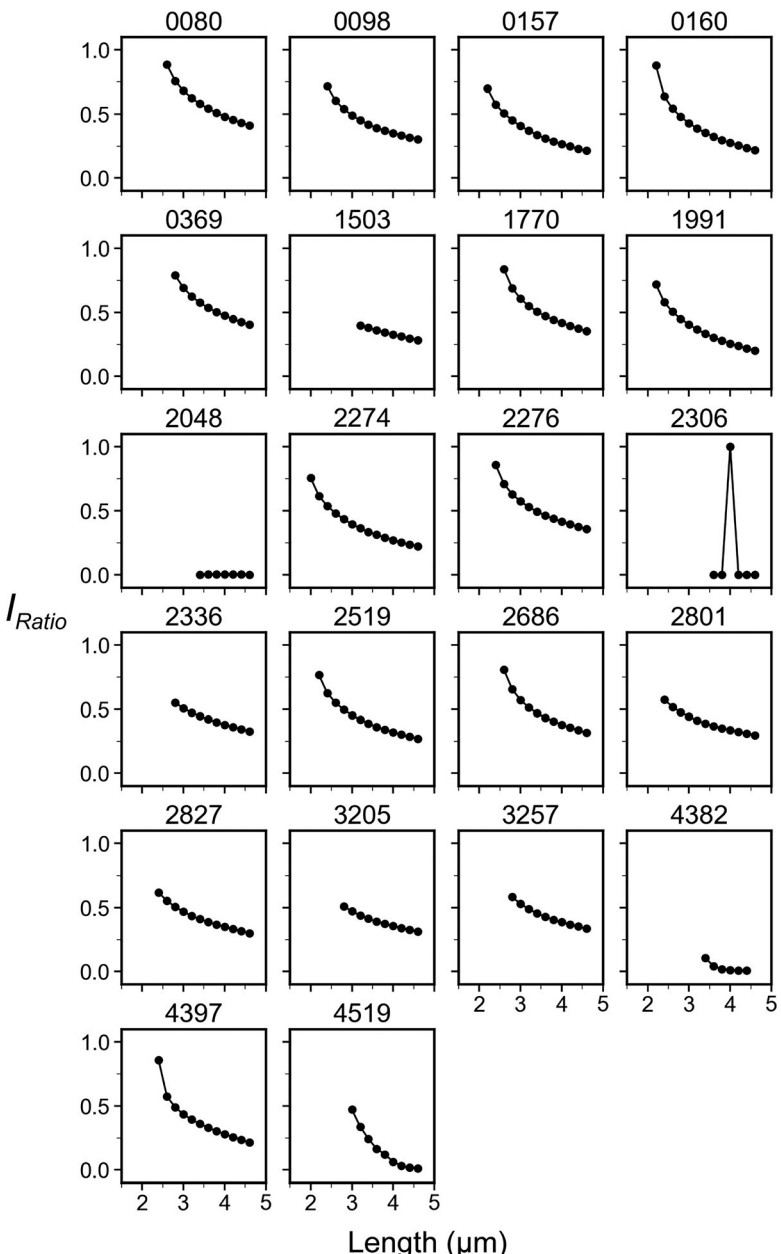

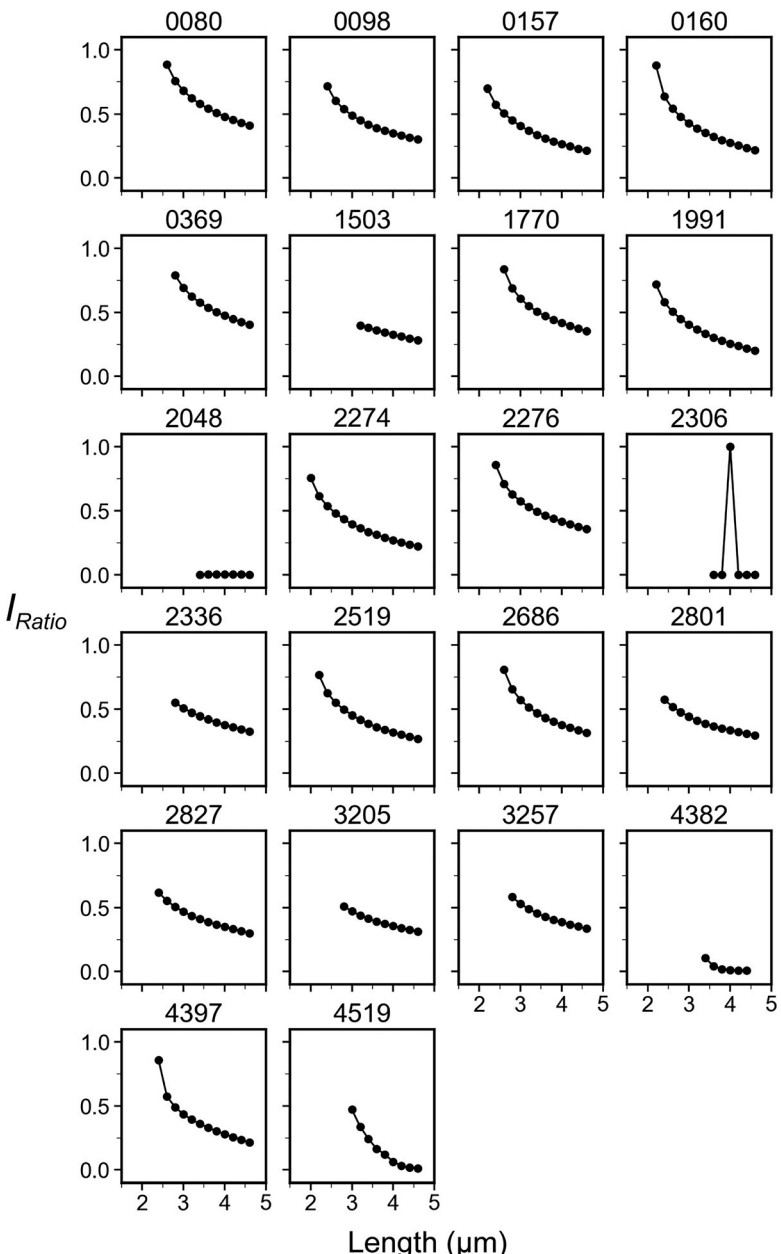

Figure S5. **Concentration profile analyses, $I_{Ratio}$, of the simulated MinD dynamics.** This figure is supplemental to Fig. 7 C.

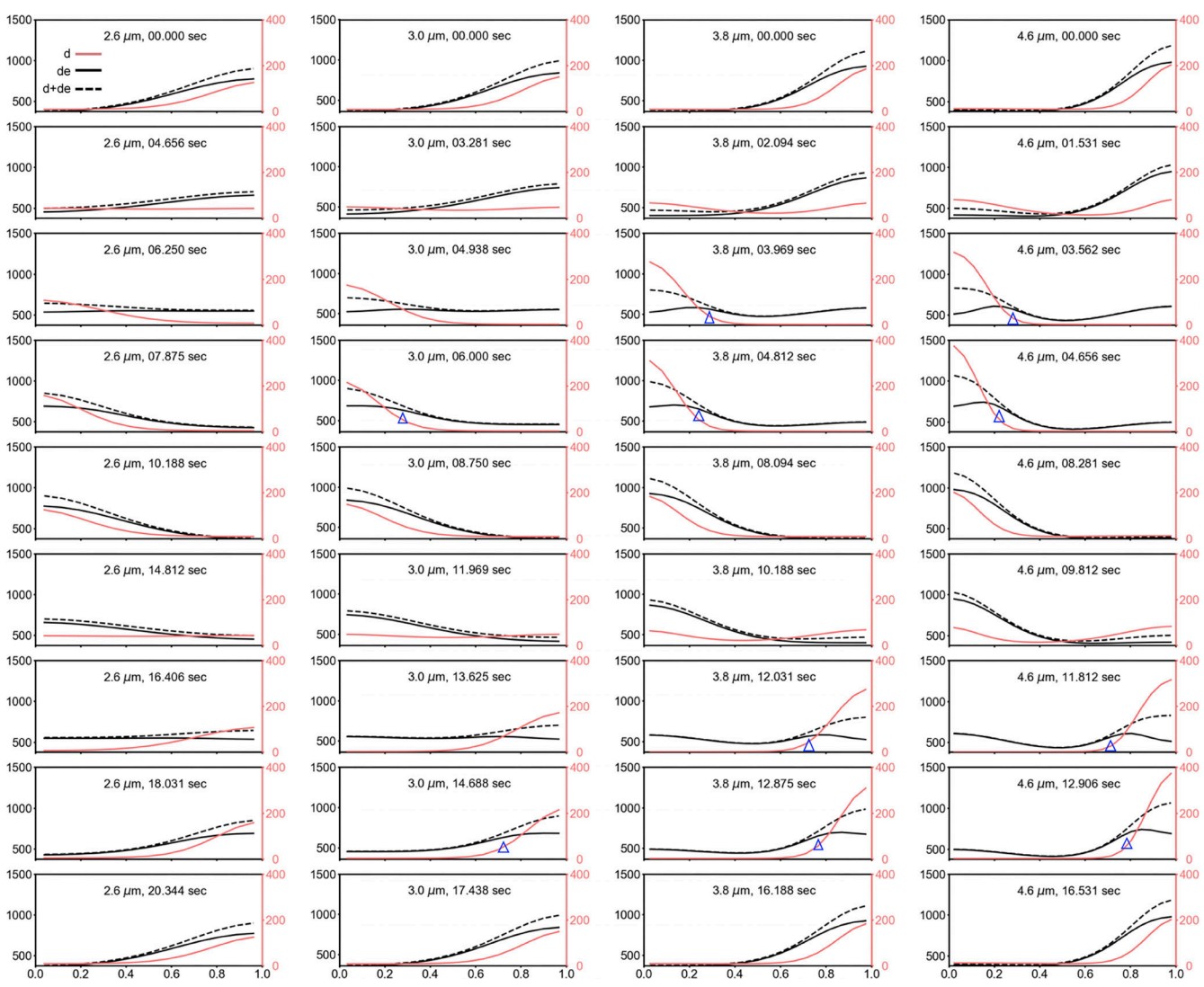

Figure S6. **Concentration profiles of the membrane-bound species in an oscillation cycle for four different cell lengths.** This figure is supplemental to Fig. 9 and contains snapshots captured from the top panel of Video 1. Black solid line: $C_{de}$ (de); black dashed line: $C_d + C_{de}$ (d + de); red-hued line: $C_d$ (d). The blue empty arrows indicate the locations where MinE accumulated at the trailing edge of the MinD polar zone.

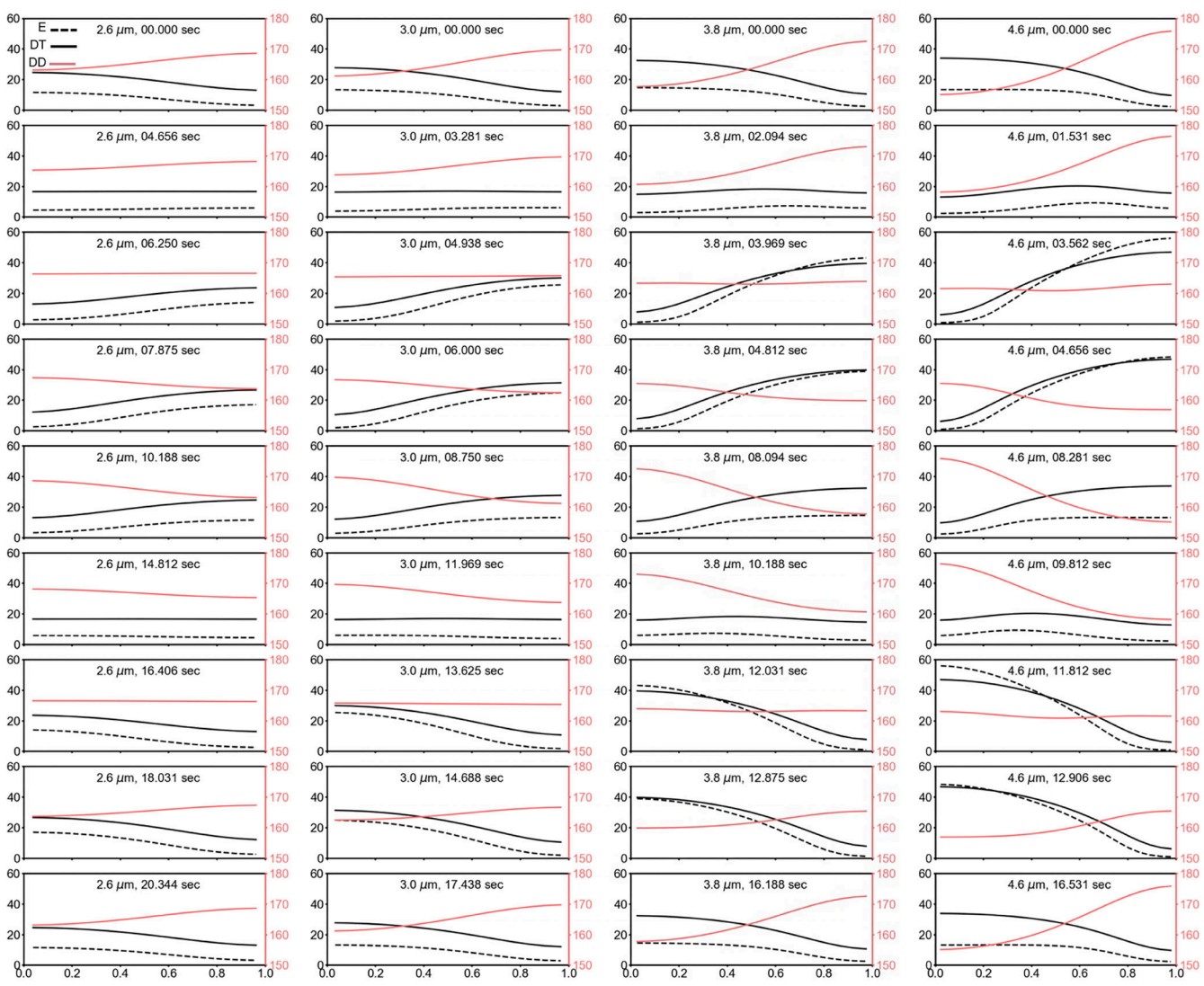

**Figure S7. Concentration profiles of the cytosolic species in an oscillation cycle at four different cell lengths.** This figure is supplemental to Fig. 9 and contains snapshots captured from the lower panel of Video 1. Black solid line: $C_{MinD\text{-}ATP}$ (DT); black dashed line: $C_{MinE}$ (E); red-hued line: $C_{MinD\text{-}ADP}$ (DD).

**Video 1. Concentration profiles of the membrane-bound species (top panel) and the cytosolic species (lower panel) in an oscillation cycle for four different cell lengths.** These include $C_{MinD\text{-}MinE}$ (de) (upper panel, black solid line), $C_{MinD} + C_{MinD\text{-}MinE}$ (d + de) (upper panel, black dashed line), $C_{MinD}$ (d) (upper panel, red-hued line), and the cytosolic species $C_{MinD\text{-}MTP}$ (DT) (lower panel, black solid line), $C_{MinE}$ (E) (lower panel, black dashed line), and $C_{MinD\text{-}ADP}$ (DD) (lower panel, red-hued line). This composite video contains a total of 131, 113, 105, and 107 frames for cell lengths of 2.6, 3.0, 3.8, and 4.6 µm, respectively. The time step between consecutive frames is 0.156 s. The oscillation periods are 20.312, 17.500, 16.250, and 16.563 s, for cell lengths of 2.6, 3.0, 3.8, and 4.6 µm, respectively.

**Provided online are Table S1, Table S2, Table S3, Table S4, Data S1, Data S2, Data S3, Data S4, Data S5, Data S6, Data S7, and Data S8. Table S1 lists strains and plasmids. Table S2 lists oligonucleotides. Table S3 shows estimation of single-cell weight. Table S4 shows comparison of parameters among different studies. Data S1 shows measurement of time-lapse images showing MinD oscillations. Data S2 shows oscillation period and velocity before and after division. Data S3 shows cell lengths, oscillation periods, and positions of division sites labeled by FtsA. Data S4 shows bulk fluorescence intensity of sfGFP-MinD, cell areas measured in individual growing cells, and estimation of number of MinD molecules through out the cell cycle. Data S5 shows measurements of $\lambda_N$ and $I_{Ratio}$ using the experimental datasets. Data S6 shows simulated reaction rate constants of MinD oscillation through screening procedures as illustrated Fig. 7. Data S7 shows data of spatiotemporal distribution of different protein species in a complete**

oscillation cycle, including membrane-bound MinD (d), MinD-MinE complex (de) and their corresponding intensities. Data S8 shows data used to determine the number of MinD and MinE molecules in W3110, and sfGFP-MinD and MinE molecules in FW1541.

