## [Peer Review File · The Journal of Cell Biology]

Growth-dependent concentration gradient of the oscillating Min system in *Escherichia coli*

Claudia Parada, Ching-Cher Yan, Cheng-Yu Hung, I-Ping Tu, Chao-Ping Hsu, and Yu-Ling Shih

Corresponding Author(s): Yu-Ling Shih, Institute of Biological Chemistry, Academia Sinica and Chao-Ping Hsu, Institute of Chemistry, Academia Sinica

Review Timeline:

Submission Date:	2024-06-18
Editorial Decision:	2024-08-21
Revision Received:	2024-09-17
Editorial Decision:	2024-10-02
Revision Received:	2024-10-22

Monitoring Editor: Min Wu

Scientific Editor: Dan Simon

Transaction Report:

DOI: <https://doi.org/10.1083/jcb.202406107>

Revision 0

Review #1

1. Evidence, reproducibility and clarity:

Evidence, reproducibility and clarity (Required)

****Summary:****

Parada et al. studied both experimentally and theoretically the MinD concentration distribution of Min waves during cell growth. The main finding was that (i) the gradient of MinD is steeper for longer cells and accordingly the MinD concentration at the middle of cell is lower, (ii) period of the oscillation is independent to the cell length, and (iii) those features are shared even under glucose starvation except the MinD gradient is steeper. (iv) Those results are supplemented by the analyses of the reaction-diffusion equations in which parameters that can reproduce the MinD concentration distribution are identified.

I think the results are interesting; basically, as the cell grows, the contrast of the wave becomes clearer, such the MinD concentration at the cell centre decreases. The results may clarify the mechanism of FtsZ accumulation at the cell centre more quantitatively. The experiments were performed by measuring the fluorescent intensity of MinD during cell growth and analysing the intensity distribution along the long axis of the cell. The theoretical results were based on the analyses of the reaction-diffusion model. Both approaches are already well established and the results sound. Nevertheless, I do not think the novelty of this work is not well highlighted in the current manuscript; I think most of the results, except (iii) and (iv), have already been shown explicitly or implicitly in the previous studies. Min oscillations in a growing cell have been analysed both theoretically and experimentally in (Meacci 2005) and [1]. The concentration distribution and period of the oscillation were measured. The complete results were presented in [2], and I am not aware of those results in scientific journals (the thesis is available online). Nevertheless, I think it is fair to cite those studies and compare the current results with them. In fact, in [2], it was shown that the concentration of MinD near the cell centre decreases as the cell grows, the total MinD concentration is approximately constant during the growth (therefore, the number of the molecules increases), and that the variance of the period becomes smaller as the cell grows. I do not think those previous studies spoil this work, and this work deserves publication somewhere. Still, the authors should highlight the novelty of this study more clearly.

****Major comments:****

(i) In (Meacci 2005) and [1,2], it was claimed that the standard deviation of the period is comparable with the mean period, particularly for the shorter cell. Therefore, they did not claim the period is independent to the cell length. As far as I understood, the variance arises from the variance of the total protein concentration in the assemble of cells. I am wondering how the authors are able to conclude the constant period in different cell length. I also point out that in the theoretical part of (Meacci 2005), the period is, in fact, increasing as the cell grows and suddenly decreases at the length in which cell division occurs.

(ii) I do not think the explanations of the reaction-diffusion model were well described. The authors mentioned that they studied a one-dimensional model and used the delta function to describe the membrane reaction. Did the authors study 1D cytosol and 0D membrane? Then, why the surface diffusion term exists in (4) and (5)? I believe the authors simply assumed that both the membrane and the cytosol are 1D (with larger diffusion constants for cytosolic Min concentrations). Then, the delta functions in (1)-(5) are not necessary. In (Wu 2015), the delta function was used in order to treat a 2D membrane embedded in 3D space.

Besides that, there is no description of the initial conditions for the concentration fields to solve the reaction-diffusion equations. I think the description of the no-flux boundary condition is better put in the Methods rather than supplementary materials.

(iii) As in the previous comment, the current model did not take into account the geometry of the system; namely, cytosol is in 3D and membrane is on 2D. Recent theoretical studies can handle the effect, and also the effect of confinement. I would appreciate it if the authors would make a comment on whether those issues are relevant or not for the conclusion of this work.

(iv) I would appreciate it if the authors would describe the screening process more clearly. I did understand the first screening is a finite imaginary part and a positive real part at the first mode of spatial inhomogeneity in the eigenvalues. However, I did not understand the other processes clearly. The second screening is based on λ_N and I_{Ratio} , but its criteria is not clear. I think both quantities fluctuated in experimental results and I am not sure what to define numerical results match them.

The third process is based on a fitting error using the fitting function of linear increase plus a constant. I am not sure why we need to exclude, for example, the bottom right example in Fig.S6 because it shows no oscillation until the cell length of 3 μm but then the gradient linearly increases. Please clarify how to justify the criteria. The same argument applies to the fourth screening process. It is not clear why the slope should be smaller than 2.

(v) The authors claimed that the steeper gradient of MinD under glucose starvation results in cell division for shorter cells. I do not think the claim is convincing. It is necessary to measure the correlation between the length at the cell division and the gradient. It would also be nicer to show the correlation under other parameters. I think those studies truly support the authors' claim and the novelty of this work.

(vi) The conclusion at Line 346 "This plasticity arises from spatial differences in molecular interactions between MinD and MinE, as demonstrated..." looks unclear to me. My understanding is that (i) by screening the randomly sampled parameters in the reaction-diffusion model, the authors found the parameters that "match" experimental results, and (ii) the parameters after screening show the correlation between them (k_{dD} - k_{dE} and k_{D} - $k_{\text{ATP}\rightarrow\text{ADP}}$). The logic heavily relies on the reaction-diffusion model is quantitatively correct. First, I think it is better to explain the logic more explicitly, that is, the claim of the molecular interaction is not based on the experimental

facts. Second, I personally think the reaction-diffusion model used in this work does not reproduce quantitatively the experimental results, as discussed in (iii) and also (iv). Please make some discussions on how to justify the comparison between the model and experiments.

(vii) I did not capture the point why the authors can claim "... further distinguishing in vivo and in vitro observations. " at Line 350. I did not find the results comparing with vitro studies. I would appreciate a demonstration of vitro results and/or references.

****Minor comments:****

1. Line 214:

It should be "Fange and Elf".

2. I think it is better to show sampled points in Fig.4C and 4D to show how dense the authors sampled in the parameter space.

REFERENCES:

[1] Fischer-Friedrich, Elisabeth / Meacci, Giovanni / Lutkenhaus, Joe / Chaté, Hugues / Kruse, Karsten, "Intra- and intercellular fluctuations in Min-protein dynamics decrease with cell length", Proceedings of the National Academy of Sciences, 107, 6134-6139 (2010).

[2] Meacci, Giovanni, "Physical Aspects of Min Oscillations in Escherichia Coli", PhD thesis (2006) available at https://www.pks.mpg.de/fileadmin/user_upload/MPIPKS/group_pages/BiologicalPhysics/dissertations/GiovanniMeacci2006.pdf

2. Significance:

Significance (Required)

General assessment:

I think the strength of this study is that it potentially shows the quantitative correlation between the MinD concentration gradient during the oscillation and the cell length when it divides. However, the current data of glucose starvation is not convincing enough. The model parts are interesting but their connection to the experiments is not clear in the current manuscript.

Advance:

The advance of this study is to measure the MinD concentration gradient under glucose starvation, and to compare the experimental results with the (simplified) model under a wide range of parameters. I do not think the advance in the current manuscript looks conceptual level because the conceptual conclusions are not really convincing from the results. In this respect, the advance of this work may be technical.

Audience:

As a theoretician working on biophysics, including the model of the Min system, I think a specialised audience would be interested in this study. People who are studying the mechanism of the Min oscillation and resulting cell division, particularly those who are interested in both experiments and models, would be interested in this work. For the broad audience, I do not think the novelty of this study is well described.

3. How much time do you estimate the authors will need to complete the suggested revisions:

Estimated time to Complete Revisions (Required)

(Decision Recommendation)

Between 3 and 6 months

4. Review Commons values the work of reviewers and encourages them to get credit for their work. Select 'Yes' below to register your reviewing activity at Web of Science Reviewer Recognition Service (formerly Publons); note that the content of your review will not be visible on Web of Science.

Yes

Review #2

1. Evidence, reproducibility and clarity:

Evidence, reproducibility and clarity (Required)

****Summary:****

This work by Parada et al showed that in the oscillatory Min System, MinD gradient was steeper in longer e.coli cells, while period was stable. This behavior was recapitulated in a mathematical model and it also revealed coordinated reaction rates in a wide range of parameter space.

****Major comments:****

1. There were some inconsistencies between experimental and modeling data. Wave slope (λN) plateaued at $\sim 3\mu\text{m}$ in the model but not shown in the experiment (Fig.3B). The period was much less in the model (Fig. S8) than in the experiment (Fig. 1B).
2. Generally, I found that the data of starved condition added little to the major message. Unless the model can recapitulate the even steeper gradient in such condition by tuning starvation-related parameters, it may be removed.
3. The authors need to compare what was different/novel between the model in this study and previous models such as Wu, et al 2015 and highlight the uniqueness of this work.
4. The model explored parameter space of reaction rates and found 60 sets. The KdE, KD, KdD, KADP-ATP ranged 6 orders of magnitude. It is interesting data in itself, but cells were not likely to

vary that much for reaction rates. The relevance should be discussed.

****Minor comments:****

1. Fig.1B colors were conflicting. The legend was different than diagram. Fig.1C no scale for x axis.
2. Fig.S6A How the 638 oscillatory parameter sets were matched with experimental data and screened to 174 sets was not clear. Data of fitting error <0.12 and slope <2 were filtered. Authors should explain the criterion for data filtering.
3. Significant digits were not used properly. For example, the period (table 1) was showed as 46.00 sec, but the imaging interval was 12 sec, the 2 decimal digits were thus meaningless. The same argument goes for length measurement at 2.84 μm , while the optical resolution of the microscope used should be no good than 200nm.
4. For scatter plot like Fig.1D-G, generally smaller dots would show trend more obvious.
5. The molecular mechanism of why MinD gradient increases with length was not the scope of the current study, but better to be discussed.
6. Fig.S8, why sudden jump in period in many of the sets of both groups?

2. Significance:

Significance (Required)

Min system was well-studied oscillation mechanism to restrict FtsZ at cell center. Previous work has shown how the system work molecularly, simulated the behavior and reconstituted many different patterns in vitro. The major new information from this work was: 1. the rigorously measured endogenous level of MinD and MinE; 2. gradient increased with length; 3. a model recapitulated this relationship and explored parameter space of reaction rates.

The paper was well presented, experiments and analysis were rigorous, and the conclusions were not overstated. It should interest specialized cell biologists studying cell size, oscillation pattern.

3. How much time do you estimate the authors will need to complete the suggested revisions:

Estimated time to Complete Revisions (Required)

(Decision Recommendation)

Less than 1 month

4. Review Commons values the work of reviewers and encourages them to get credit for their work. Select 'Yes' below to register your reviewing activity at Web of Science Reviewer Recognition Service (formerly Publons); note that the content of your review will not be visible on Web of Science.

Yes

Review #3

1. Evidence, reproducibility and clarity:

Evidence, reproducibility and clarity (Required)

The manuscript shows that the concentration of MinD does not change during the division cycle of *E. coli*. Due to the oscillation pattern the concentration of MinD decreases at the mid-cell which makes it favorable for the division. The mid-cell decrease in concentration of MinD is majorly length dependent. The oscillation pattern is not due to the change in concentration of MinD, but due to the plasticity arises from the spatial differences in molecular interactions between MinD and MinE. The manuscript is well written, the experiments are performed carefully and the results will be of interest to readers from variety of field. However, there are several concerns need explanation.

****Major concerns:****

One of my major concern is these interactions are not shown experimentally but explained using either the previously published literature or mathematical models. Further, the previous literatures are shown on in vitro models which does not mimic the in vivo system fully.

The concentration of MinD does not change with the increasing length of the cell. Is the MinD concentration (or copy numbers) is different in the case of cells growing in low glucose and when compared to the cells growing at high glucose? As per the current study a particular l -ratio at the mid-cell is required to initiate the cell division. In the case of cells growing at low glucose, how this required l -ratio is achieved at the mid-cell?

There is decrease in the MinD oscillation time observed in low glucose condition. As explained by the authors the MinD oscillation is mainly guided by the FtsE induced removal of MinD from the membrane, how the authors can explain this decrease? Further, it is explained that the concentration of cellular ATP is in much higher concentration compared to the required amount for this oscillation. As the l ratio is majorly dependent on the cell length, what could be the reason for the differential λN in the case of low and high glucose condition?

MinD is a highly insoluble protein. It also has an amphipathic helix and thus most of the time it binds to the membrane. The method used by the author to determine the cellular MinD concentration (mentioned in Fig S1) will only give the concentration of the soluble MinD and not of the total MinD. How the authors justify this as the total concentration. This is also the same in the case of MinE copy number calculation. Authors may need to perform the transcriptome analysis and compare both the data.

One of the main question asked by the authors in the abstract is. "How the intracellular Min protein concentration gradients are coordinated with cell growth to achieve spatiotemporal accuracy of cell division is unknown". Although the authors have shown that there is a change in concentration gradient during cell growth, the mechanism for the same is not very well explained. Authors have not provided any specific explanation for the increase in the velocity of the MinD oscillation and the gradient formation. How the velocity of MinD is increasing although there is no increase in the MinD concentration.

Figure 2B: shows the overall concentration of MinD in a single cell varies between 1180 - 1160 molecules/ μm^2 . In Fig 2C it is mentioned that mid-cell has a MinD concentration of 120-20

molecules/ μm^2 . Further, Fig3C and 3F shows I-ratio values varies between 0.6-0.4. Considering the values given the I-ratio (I min/ I max) should be between 0.1- 0.01. Authors need to explain the same.

Figure 2C: The data in both the Y-axes are not matching and needs more clarification in the legend. Whether the number of molecules were counted only in the marked 200 nm area? If so, why the Y-axis 1 (molecules/ μm^2) is decreasing 7 times, whereas, Y-axis 2 (molecules) is only by 2 times.

****Other comments:****

Line 84: Requires reference for this statement.

Line 96: Can authors provide other evidence or validation for the determination of the copy numbers such as transcriptome analysis.

Fig 1C: what is the units of time in Fig 1C? Is it equal for all the cell lengths?

Page 6, line 136-138: what could be the possible mechanism for change in velocity at different cell cycle time?

Page 7, line 155: Any evidence for claiming the same?

Page 7, line 156: Is there any proof authors can show that burst MinD synthesis occurs during the division? If not in the case of MinD, is it shown in any other protein?

Page 9, line 217: The Fig 4A is not explained clearly and all the terms mentioned needs to be explained. This figure is used to explain the differential concentration of MinD at the poles and the mid-cell, thus needs to be explain more clearly.

Page 12, line 285: What is meaning of default speed of MinD oscillation in new-born cells? Do the authors observed any specific velocity in the new-born cells? What is the explanation for length dependent oscillation velocity for MinD?

2. Significance:

Significance (Required)

General assessment: Major work of the manuscript is relying on the mathematical models, whereas the audience are majorly from the biology fields and thus simplified explanations are required in many places. Many of the legends in the figures require more explanation for better understanding. If possible more experimental data can be added, specifically to explain the model mentioned in figure 4A.

Advance: The study is adding to the existing knowledge and will be helpful to fill the conceptual gaps in understanding the mid-cell MinD concentration and what may favor the initiation of bacterial division.

Audience: Majorly the microbiology community will be interested in the study. This will also be interest to Physicists and mathematical persons working to understand bacterial division.

3. How much time do you estimate the authors will need to complete the suggested revisions:

Estimated time to Complete Revisions (Required)

(Decision Recommendation)

Between 1 and 3 months

4. Review Commons values the work of reviewers and encourages them to get credit for their work. Select 'Yes' below to register your reviewing activity at Web of Science Reviewer Recognition Service (formerly Publons); note that the content of your review will not be visible on Web of Science.

Yes

Review #4

1. Evidence, reproducibility and clarity:

Evidence, reproducibility and clarity (Required)

The study by Parada et al. illuminates the intricate interplay between Min proteins, exemplified by MinD, and cell growth in *E. coli*. Their findings demonstrate that the MinD concentration gradient steepens progressively as cells elongate, potentially influencing FtsZ ring formation via MinC. Moreover, their comprehensive reaction-diffusion model not only corroborates experimental observations of length-dependent concentration gradients but also underscores the critical role of kinetic interactions involving Min proteins, the membrane, and ATP. This elucidation significantly advances our understanding of the oscillatory mechanisms within the Min system. Both the experimental and simulation data are robust, and the manuscript is exceptionally well-written. I express my full support for publication pending the satisfactory resolution of the outlined concerns.

1. Remove the dot in front of "Min" in line 57.
2. In lines 82-84, the statement "...The distribution of the division inhibitor MinC may be synchronized with spatiotemporal differences in MinD concentrations, leading to a stable placement of the FtsZ ring at the midcell..." suggests a potential synchronization between MinC and MinD oscillations. It is crucial to investigate if sfGFP-MinC exhibits similar concentration gradient oscillatory behavior in vivo as observed with MinD.
3. Ensure consistent significant digits throughout the text. For instance, $1.95 \pm 0.16 \mu\text{M}$ in line 97, $1.4 \pm 0.13 \mu\text{M}$ in line 98, and $1.9 \pm 0.2 \mu\text{M}$ in line 100 have varying precision. Consider using integers for molecules.
4. Address the discrepancy in expression levels of MinD and MinE between strain FW1541 and its parental strain W3110. Given the labeling effect, it is possible that MinD expression levels differ. However, MinC's expression level should be approximately the same. Conduct whole-genome

sequencing of both strains to identify any additional mutations.

5. Clarify the apparent discrepancy between lines 112 and 127. Line 112 suggests that the periodic regularity of interpolar oscillations increases with cell length, as demonstrated in Fig 1B-C, 1E, Fig S5. However, in the subsequent section (starting from line 127), the authors state that oscillation periods remain relatively stable across cells of different lengths. Provide clarification on this apparent discrepancy.

6. Specify if the analysis was limited to non-constricted cells. If so, state this explicitly in the text, as it could impact the interpretation of results, especially in relation to the linear dependence of cell length on time before constriction, as shown in Fig S3C.

7. Improve clarity in Fig 2A by using distinct colors (e.g., green and red) for differentiation on the Y-axis.

8. Correct "of" to "from" in line 223 for improved clarity and accuracy.

9. Include the missing "A" in Fig S6A for completeness and accuracy.

10. Ensure consistency in referencing style (full names versus short names) throughout the manuscript.

2. Significance:

Significance (Required)

While numerous commendable in vitro studies have explored the oscillatory behavior of the Min system, this work uniquely delves into the oscillation of MinD within live cells. It unveils the remarkable coordination between intracellular Min protein concentration gradients and cell growth, shedding light on the precise spatiotemporal regulation of cell division.

3. How much time do you estimate the authors will need to complete the suggested revisions:

Estimated time to Complete Revisions (Required)

(Decision Recommendation)

Less than 1 month

4. Review Commons values the work of reviewers and encourages them to get credit for their work. Select 'Yes' below to register your reviewing activity at Web of Science Reviewer Recognition Service (formerly Publons); note that the content of your review will not be visible on Web of Science.

Yes

**Manuscript number:** RC-2023-02110

**Corresponding author(s):** Yu-Ling Shih

1. General Statements [optional]

*This section is optional. Insert here any general statements you wish to make about the goal of*
*the study or about the reviews.*

We sincerely thank the reviewers for their constructive feedback. We have made
significant revisions to where applicable and to the mathematical modelling section of the
manuscript to address your concerns and questions. We summarize the key points in the revised
manuscript as follows.

1. The key finding of our study, involving experimental measurements and mathematical
modelling, is plasticity in the MinD concentration gradient, which results from spatial
differences in molecular interactions and is an intrinsic property of the Min system during cell
growth. This study reveals not only the role of the MinD concentration gradient in modulating
bacterial cell division site placement but also showcasing an example of cellular components in
the form of a concentration gradient in fundamental cellular processes, a concept crucial in cell
biology.

2. The reviewers suggested clarification on the differences between our study and previous
studies involving experimental measurements and mathematical modeling of Min oscillations in
cells. We reply that, while previous works aimed to measure the MinD oscillations as a function
of cell length, they addressed different problems and had different experimental design and
execution methods compared to our current study. This also applies to mathematical modeling,
so direct comparisons are challenging due to the differing mathematical frameworks and
assumptions used. For instance, our modeling was to investigate the MinD concentration
gradient during cell elongation, without addressing the impact of cell shape, confinement, or the
nucleation effect of MinD. Consequently, our model cannot be generalized to other shapes, such
as those studied by Wu et al. (2015) (Wu *et al*, 2015) and Zieske et al. (2016) (Zieske *et al*,
2016). In addition, our work investigates variations in the shape of the concentration gradient as
cells elongate, which differs from the focus on the transition from stochastic to regular pole-to-
pole oscillation at specific cell lengths reported by Fischer-Friedrich et al. (2010). Meanwhile,
our work also differs from Meacci's works (Meacci & Kruse, 2005) (Meacci *et al*, 2006) that as
an impactful work two decades ago, reported general behaviours of Min oscillations in living
cells and used mathematical models. Detailed are added to the revised manuscript where
appropriate.

3. We have re-run the simulation to improve the modelling procedures and results and the
corresponding text are re-written. This operation allowed us to not only probe for the general
behaviours of the system but also the variable length-dependent concentration gradients. As a
result, we were able to obtain a few parameter sets that generate features of the oscillation period,
λ_N and I_{Ratio} , that well mimic MinD oscillation in the cellular context.

We further tested the impact of different kinetic constants, k_{de} , k_{dD} , k_{dE} , k_D , and $k_{ADP \rightarrow ATP}$,
which represent different molecular interactions influencing the oscillation period, λ_N and I_{Ratio} .
Detailed are incorporated into the revised manuscript where appropriate.

5. Regarding the opposite opinions of the inclusion (Reviewer 1) or removal (Reviewer 2) of
results from more culture conditions, we decided to keep only one condition as in the previous
version for the following reasons. To draw convincing conclusions, we consider it more
important to characterize all aspects under the same growth condition and avoid manipulation.
Therefore, the main conclusions are drawn from our experiments characterizing several aspects
of MinD oscillations in cells growing with 0.4% glucose. We maintain only one other condition,
0.1% glucose, in support of these observations. Including analyses of cells growing under other
conditions will not change the main conclusions of this work.

6. Studying the variable concentration gradient underlying the dynamic oscillations of the Min
system may be of broad interest to cell biologists since the concentration gradient plays a
fundamental role in various cellular processes, and the concept of concentration gradients is
crucial in cell biology. Examples of related processes include passive and active transport,
osmosis, cell signalling, and maintenance of cellular homeostasis. These processes allow cells to
respond to their environment, regulate their internal conditions, and perform important functions
required for survival and normal function.

Therefore, the audience of this work can include the broader general audience of cell biology and
physical biology rather than just the immediate specialized audience interested in the Min system.

2. Point-by-point description of the revisions

**Reviewer comments:** Arial

**Author responses:** Times New Roman

Changes are highlighted in green in a copy of the manuscript for your review.

**Reviewer #1** (Evidence, reproducibility and clarity (Required)):

Summary:

Parada et al. studied both experimentally and theoretically the MinD concentration distribution of
Min waves during cell growth. The main finding was that (i) the gradient of MinD is steeper for
longer cells and accordingly the MinD concentration at the middle of cell is lower, (ii) period of
the oscillation is independent to the cell length, and (iii) those features are shared even under
glucose starvation except the MinD gradient is steeper. (iv) Those results are supplemented by
the analyses of the reaction-diffusion equations in which parameters that can reproduce the
MinD concentration distribution are identified.

I think the results are interesting; basically, as the cell grows, the contrast of the wave becomes
clearer, such the MinD concentration at the cell centre decreases. The results may clarify the
mechanism of FtsZ accumulation at the cell centre more quantitatively. The experiments were
performed by measuring the fluorescent intensity of MinD during cell growth and analysing the
intensity distribution along the long axis of the cell. The theoretical results were based on the
analyses of the reaction-diffusion model. Both approaches are already well established and the
results sound.

Nevertheless, I do not think the novelty of this work is not well highlighted in the current
manuscript; I think most of the results, except (iii) and (iv), have already been shown explicitly or
implicitly in the previous studies. Min oscillations in a growing cell have been analysed both
theoretically and experimentally in (Meacci 2005) and [1] (Fischer-Friedrich *et al*, 2010). The
concentration distribution and period of the oscillation were measured. The complete results
were presented in [2] (Meacci *et al.*, 2006), and I am not aware of those results in scientific
journals (the thesis is available online). Nevertheless, I think it is fair to cite those studies and
compare the current results with them. In fact, in [2], it was shown that the concentration of
MinD near the cell centre decreases as the cell grows, the total MinD concentration is
approximately constant during the growth (therefore, the number of the molecules increases),
and that the variance of the period becomes smaller as the cell grows. I do not think those
previous studies spoil this work, and this work deserves publication somewhere. Still, the
authors should highlight the novelty of this study more clearly.

**ANS:** We thank the reviewer for recognizing the soundness of our experimental and theoretical
approaches and results. The key finding of our study, involving experimental measurements and
mathematical modelling, is plasticity in the MinD concentration gradient, which results from
spatial differences in molecular interactions and is an intrinsic property of the Min system during
cell growth. This study reveals not only the role of the MinD concentration gradient in
modulating bacterial cell division site placement but also showcasing an example of cellular
components in the form of a concentration gradient in fundamental cellular processes, a concept
crucial in cell biology.

We believe that the established techniques and methods are integral to a broad range of works
and provide confidence in improving them and using them to test hypotheses and obtain results.

We also appreciate the reviewer for pointing out that Meacci's PhD thesis entitled "Physical
aspects of Min oscillations in *Escherichia coli*" (Meacci & Kruse, 2005) is available online for
public access. This thesis, along with two publications (Meacci & Kruse, 2005) (Meacci *et al.*,
2006), explored Min oscillations in growing cells and used mathematical models. These two
published works are cited in the previous version of the manuscript because we agree that these
earlier works provide valuable context. As recommended, we went through these works again
and the work by Fischer-Friedrich *et al.* (2010) (Fischer-Friedrich *et al.*, 2010) to compare their
wet experiments and mathematical models with ours, which are detailed in the Supplemental
Results and Discussion (Lines 26-147).

Here, we emphasize that although the published works and our work set the goal of measuring
the spatiotemporal distribution of oscillating MinD concentration gradients as a function of cell
length, we conceived the problem differently and therefore used different experimental designs
and analysis approaches, which have led to the key conclusions that differentiate our work from
theirs.

Major comments:

(i) In (Meacci 2005) and [1,2], it was claimed that the standard deviation of the period is
comparable with the mean period, particularly for the shorter cell. Therefore, they did not claim
the period is independent to the cell length. As far as I understood, the variance arises from the
variance of the total protein concentration in the assemble of cells. I am wondering how the
authors are able to conclude the constant period in different cell length.

**ANS:** Thank you for these comments. Because all three works differ in terms of experimental
design, execution, and data analyses, it is not possible to make a fair comparison between them
although similar features can be found at first glance. Therefore, we would like to draw attention
to the specific points and views that altogether contribute to our understanding of the Min system.
Details of oscillation period and its relationships to cell length from different works are
summarised.

1. In our experiments, we found that the oscillation periods ranged from 36.8 to 65.6 sec, as
measured from a population of cells (length of 1.9-4.5 μm ; main text, Fig. 1E). Moreover, the
standard deviations of the period ranged from 5.4% to 34.8% of the period, with larger standard
deviations more common in shorter cells (Fig. 1D), indicating that regular inter-polar oscillations
are more likely to occur in longer cells.

In Fischer-Friedrich *et al.* (2010) (Fischer-Friedrich *et al.*, 2010), the authors investigated the
length-dependent switching from stochastic to regular oscillation states. They reported stochastic
switching MinD oscillation between two cell poles in cells below 2.5 μm . MinD starts to
oscillate regularly from pole-to-pole between 2.5-3 μm with an oscillation period of 80 sec.
Above 3.5 μm , MinD invariably undergoes regular oscillation with an initial period of 87 sec and
then decreases to 70 sec at the end. In addition, their observation of a longer period at the initial
phase and a shorter period after the cells grew beyond 3.5 μm somewhat supported our
simulation results (Fig. 4C-H, left).

In Meacci's work (Thesis: Figure 2.14; Meacci and Kruse (2005) (Meacci & Kruse, 2005):
Figure 5(b)), the temporal oscillation periods were measured from 40 to 120 sec when focusing
on cells with lengths similar to those in our measurements (black dots in Meacci's chart).

Our measurements of oscillation periods clearly show much smaller fluctuations than those in
Meacci's study. Differences can arise across different bacterial strains and culture conditions that
may significantly affect the amount and quality of protein expressed in individual studies.

2. We have changed the wording from 'constant period' to 'quite stable period' throughout the
manuscript. This description is based on our experimental measurements (Fig. 1D, E) and is also
supported by our mathematical modelling (Fig. 4C-H, left).

3. We have added comparison of different models in the Supplemental Information (Lines 26-
147).

I also point out that in the theoretical part of (Meacci 2005), the period is, in fact, increasing as
the cell grows and suddenly decreases at the length in which cell division occurs.

**ANS:** Thank you for this comment about the statement from the theoretical model of Meacci and
Kruse (2005): "the period is increasing as the cell grows and suddenly decreases at the length
in which cell division occurs.". Indeed, discrepancy of this point does exist. Our simulation results
revealed a mild increase in the oscillation period during cell elongation (Fig. 4C). The increase is
adjustable by varying the reaction rate constants in the simulation (Fig. 4D-H). In addition, our
imaging data showed that the oscillation period increased in newly divided cells (Fig. S4).

As mentioned earlier, because these two works differ in terms of experimental design, execution,
and data analyses, it is not possible to make a fair comparison between them.

(ii) I do not think the explanations of the reaction-diffusion model were well described. The
authors mentioned that they studied a one-dimensional model and used the delta function to
describe the membrane reaction. Did the authors study 1D cytosol and 0D membrane? Then,
why the surface diffusion term exists in (4) and (5)? I believe the authors simply assumed that
both the membrane and the cytosol are 1D (with larger diffusion constants for cytosolic Min
concentrations). Then, the delta functions in (1)-(5) are not necessary. In (Wu 2015), the delta
function was used in order to treat a 2D membrane embedded in 3D space.

Besides that, there is no description of the initial conditions for the concentration fields to solve
the reaction-diffusion equations. I think the description of the no-flux boundary condition is better
put in the Methods rather than supplementary materials.

**ANS:** Thank you for your suggestion to improve the description of the numerical model. As
summarized below, we have rewritten this section of 'Simulating the dynamic MinD
concentration gradient in growing cells' in the manuscript (Lines 235-277).

1. We have specified the dimensionality of the rate and diffusion constants of each molecule,
where applicable, in our 1D model from Lines 236-262. Their dimensionality can also be
conceived from their units, as listed in Tables 2 and S4.

2. We have specified the initial 'no-flux' boundary conditions in Lines 265, 652, and 669.

3. We agree that the delta function is not necessary and have removed it from the equations.

(iii) As in the previous comment, the current model did not take into account the geometry of the
system; namely, cytosol is in 3D and membrane is on 2D. Recent theoretical studies can handle
the effect, and also the effect of confinement. I would appreciate it if the authors would make a
comment on whether those issues are relevant or not for the conclusion of this work.

**ANS:** Thank you for pointing out this interesting aspect of cell geometry. Our model is built to
adequately describe changes in the MinD concentration gradient during cell elongation under the
assumption that a 1D description is sufficient. Thus, our model cannot be generalized to other
shapes, such as those observed in Wu et al., 2015 (Wu *et al.*, 2015). This point is now
commented upon in Supplemental Information, lines 120-123.

(iv) I would appreciate it if the authors would describe the screening process more clearly. I did
understand the first screening is a finite imaginary part and a positive real part at the first mode
of spatial inhomogeneity in the eigenvalues. However, I did not understand the other processes
clearly. The second screening is based on λ_N and I_{Ratio} , but its criteria is not clear. I
think both quantities fluctuated in experimental results and I am not sure what to define
numerical results match them.

The third process is based on a fitting error using the fitting function of linear increase plus a
constant. I am not sure why we need to exclude, for example, the bottom right example in
Fig.S6 because it shows no oscillation until the cell length of 3 μ m but then the gradient linearly
increases. Please clarify how to justify the criteria. The same argument applies to the fourth
screening process. It is not clear why the slope should be smaller than 2.

**ANS:** Thank you for your suggestions to improve the description of the screening process. We
have re-run the simulation to improve the screening process and obtained new results, and the
corresponding text and illustration are provided in the main text (Lines 235-277, 654-675) and
Fig. S6.

In brief, we fixed the diffusion coefficients D_D and D_E from Meacci et al. (2006) (Meacci *et al.*,
2006); the dissociation rate constant k_{de} from a previous simulation (Wu *et al.*, 2015); and the
experimentally measured MinD and MinE concentrations in this study. Meanwhile, the diffusion
coefficients D_d and D_{de} were assumed values based on bacterial membrane protein diffusion
(Schavemaker *et al.*, 2018). This operation allowed us to probe for the general behaviours of the
system. As a result, we were able to obtain a few parameter sets, including #2728, that generate
features of the oscillation period, λ_N and I_{Ratio} , that well mimic MinD oscillation in the cellular
context (Figs. 4C, S7-9).

We further tested the impact of different kinetic constants, k_{de} , k_{dD} , k_{dE} , k_D , and $k_{ADP \rightarrow ATP}$,
which represent different molecular interactions influencing the oscillation period, λ_N and I_{Ratio}
(Fig 4D-H). Our findings have provided us with a solid theoretical view of how oscillation
features may be controlled by different molecular interactions.

Furthermore, the modelling results help us understand the possible mechanisms associated with
oscillation cycle maintenance and length-dependent variable concentration gradients.

(v) The authors claimed that the steeper gradient of MinD under glucose starvation results in
cell division for shorter cells. I do not think the claim is convincing. It is necessary to measure

the correlation between the length at the cell division and the gradient. It would also be nicer to
show the correlation under other parameters. I think those studies truly support the authors'
claim and the novelty of this work.

**ANS:** Thank you for the comments. We would like to draw your attention to the right side of the
graph shown in Fig. 3B, E, where measurements were obtained from cells prior to division. Our
claim that “the steeper gradient of MinD under glucose starvation results in cell division for
shorter cells” is also supported by the wave slope (λ_N range): 0.4% glucose of 1.49-2.66 (cell
length range: 1.7-4.5 μm) and glucose starvation of 1.34-3.54 (cell length range: 2.1-3.8 μm).
Therefore, under glucose starvation, λ_N increases more significantly with increasing length,
allowing us to speculate on the contribution of steeper concentration gradient in stressed shorter
cell to division. In the revised manuscript, the statement can be found in the Results section
(Lines 217-218).

About the correlation between the concentration gradient and cell length at division under
different conditions, we consider it more important to characterize all aspects under the same
growth condition and avoid manipulation. In this study, the main conclusions are drawn from our
experiments characterizing several aspects of MinD oscillations in cells growing with 0.4%
glucose. In support of these observations, we decided to maintain only one other condition, 0.1%
glucose. Further analysis of cells growing under other conditions will not change the main
conclusions but will increase the difficulty of determining how the MinD concentration changes
with cell growth.

(vi) The conclusion at Line 346 "This plasticity arises from spatial differences in molecular
interactions between MinD and MinE, as demonstrated..." looks unclear to me. My
understanding is that (i) by screening the randomly sampled parameters in the reaction-diffusion
model, the authors found the parameters that "match" experimental results, and (ii) the
parameters after screening show the correlation between them (k_{dD} - k_{dE} and k_D - k_{ATP} -
$>ADP$). The logic heavily relies on the reaction-diffusion model is quantitatively correct.

First, I think it is better to explain the logic more explicitly, that is, the claim of the molecular
interaction is not based on the experimental facts.

**ANS:** Thank you for your comments. The kinetic parameters of the mathematical model are
discussed in the main text, lines 243-262. Sources of them are now also listed in Table 2- Values
of the simulated parameter set #2827.

Second, I personally think the reaction-diffusion model used in this work does not reproduce
quantitatively the experimental results, as discussed in (iii) and also (iv). Please make some
discussions on how to justify the comparison between the model and experiments.

**ANS:** Thank you for your constructive comments. Using the improved screening process
involving experimental observations (Main text, lines 263-277; Fig. S6), we were able to obtain
a few parameter sets, including #2728, that generate features of the oscillation period, λ_N and
I_{Ratio} , that well mimic MinD oscillation in the cellular context (Figs. 4C, S7-9).

(vii) I did not capture the point why the authors can claim "... further distinguishing in vivo and in

vitro observations. " at Line 350. I did not find the results comparing with vitro studies. I would
appreciate a demonstration of vitro results and/or references.

**ANS:** Sorry for the confusion. This sentence has been removed.

Minor comments:

(1) Line 214: It should be "Fange and Elf".

**ANS:** This has been corrected in the revised manuscript (Lines 236-237).

(2) I think it is better to show sampled points in Fig. 4C and 4D to show how dense the authors
sampled in the parameter space.

**ANS:** Since we have rewritten this part, the suggested revision is no longer applicable.

REFERENCES:

[1] Fischer-Friedrich, Elisabeth / Meacci, Giovanni / Lutkenhaus, Joe / Chaté, Hugues / Kruse,
Karsten, "Intra- and intercellular fluctuations in Min-protein dynamics decrease with cell length",
Proceedings of the National Academy of Sciences, 107, 6134-6139 (2010).

[2] Meacci, Giovanni, "Physical Aspects of Min Oscillations in Escherichia Coli", PhD thesis
(2006) available at

Reviewer #1 (Significance (Required)):

General assessment:

I think the strength of this study is that it potentially shows the quantitative correlation between
the MinD concentration gradient during the oscillation and the cell length when it divides.
However, the current data of glucose starvation is not convincing enough.

**ANS:** Thank you for your comment.

The main finding of our study is plasticity in the MinD concentration gradient, which results
from spatial differences in molecular interactions and is an intrinsic property of the Min system
during cell growth.

We hypothesized that if the plasticity of the MinD concentration gradient is an intrinsic property
of the system, then this property would be robust and show consistent behaviour under different
growth conditions. Therefore, we tested this hypothesis by studying MinD oscillations under a
low-glucose condition, and the results strengthened the main conclusion derived from
experiments under the regular growth condition containing 0.4 % glucose. We believe that
further analysis of cells growing under other conditions will not change the main conclusions.
Therefore, we decide to make this section concise, containing only one additional condition,
even though we have more data than presented here.

The model parts are interesting but their connection to the experiments is not clear in the
current manuscript.

**ANS:** As mentioned earlier in this response letter, we have re-run the simulation to refine and
improve the results, and the corresponding text and illustration are provided in the main text

(Lines 235-277, 638-677, and Fig. S6). This operation allowed us to probe for the general
behaviours of the system. As a result, we were able to obtain a few parameter sets, including
#2728, that generate features of the oscillation period, λ_N and I_{Ratio} , that strongly mimic MinD
oscillation in the cellular context (Figs. 4C, S7-9). We further tested the impact of different
kinetic constants, k_{de} , k_{dD} , k_{dE} , k_D , and $k_{ADP \rightarrow ATP}$, which represent different molecular
interactions influencing the oscillation period, λ_N and I_{Ratio} (Figs. 4D-H). This effort has
provided us with a solid theoretical view of how oscillation features may be controlled by
different molecular interactions.

Advance:

The advance of this study is to measure the MinD concentration gradient under glucose
starvation, and to compare the experimental results with the (simplified) model under a wide
range of parameters. I do not think the advance in the current manuscript looks conceptual level
because the conceptual conclusions are not really convincing from the results. In this respect,
the advance of this work may be technical.

**ANS:** Thank you for this comment, but we disagree about the work is only technical.

In combination with both experimental and theoretical efforts, this work offers a quantitative
understanding of MinD oscillations in the cellular environment.

Specifically, we emphasize the conceptual conclusions of the inherent plasticity and adaptability
of the MinD concentration gradient, which contributes to division site selection. The
mathematical modelling gives us a theoretical view of how variable concentration gradient in
growing cells may be controlled by different molecular interactions. Thus, this work provides
insights into bacterial cell division regulation for further studies in the field.

Audience:

As a theoretician working on biophysics, including the model of the Min system, I think a
specialised audience would be interested in this study. People who are studying the mechanism
of the Min oscillation and resulting cell division, particularly those who are interested in both
experiments and models, would be interested in this work. For the broad audience, I do not
think the novelty of this study is well described.

**ANS:** Thank you for your comment.

We would like to point out that studying the variable concentration gradient underlying the
dynamic oscillations of the Min system may be of broad interest to cell biologists since the
concentration gradient plays a fundamental role in various cellular processes, and the concept of
concentration gradients is crucial in cell biology. Examples include passive and active transport,
osmosis, cell signalling, and maintenance of cellular homeostasis. These processes allow cells to
respond to their environment, regulate their internal conditions, and perform important functions
required for survival and normal function.

Therefore, the audience of this work may include the broader general audience of cell biology
and physical biology rather than just the immediate specialized audience interested in the Min
system. We will also reiterate the importance of specialized research, which often provides the
basis for broader application and understanding.

Reviewer #2 (Evidence, reproducibility and clarity (Required)):

Summary:

This work by Parada et al showed that in the oscillatory Min System, MinD gradient was steeper
in longer e.coli cells, while period was stable. This behavior was recapitulated in a mathematical
model and it also revealed coordinated reaction rates in a wide range of parameter space.

**ANS:** We thank the reviewer for the concise summary of our work.

Major comments:

1. There were some inconsistencies between experimental and modeling data. Wave slope (λ_N)
plateaued at $\sim 3\mu\text{m}$ in the model but not shown in the experiment (Fig.3B). The period was much
less in the model (Fig. S8) than in the experiment (Fig. 1B).

**ANS:** Thank you for pointing out this problem.

We have re-run the simulation to refine and improve the results, and the corresponding text and
illustration are provided in the main text (Lines 235-277, 638-677) and Fig. S6. This operation
allowed us to probe for the general behaviours of the system. As a result, we were able to obtain
a few parameter sets, including #2728, that generate features of the oscillation period, λ_N and
I_{Ratio} , that well mimic MinD oscillation in the cellular context (Figs. 4C, S7-9).

Regarding oscillation period, we implemented a filter of ≥ 15 sec in the early data screening
process and obtained longer period (Fig. S6, S7). Based on the parameters of set #2827 (Fig. 4C),
we further tested the impact of different kinetic constants that represent different molecular
interactions on the oscillation period (Fig. 4D-H). We found that the period can be tuned closer
to the experimental observations by adjusting rate constants k_{de} , representing detachment of the
MinDE complex from the membrane, and $k_{ADP \rightarrow ATP}$, representing recharging of MinD-ADP
with ATP (Fig. 4E,G). Please see Main text, lines 323-349 for details.

In response to the question about why the wave slope (λ_N) plateaus at approximately $3\mu\text{m}$ in the
model (Fig. 3B) but not in the experiment (Fig. 1D), we believe this discrepancy may be due to
the unavoidable limitations of current experimental data. The experiments examined a
heterogeneous population of cells, which introduces variability and noise, whereas the
simulations modeled a growing bacterial cell *in silico*, providing a more controlled and
homogeneous environment. Nonetheless, our mathematical modeling reinforces our conclusions
and hypotheses derived from wet experiments. A statement is added to the main text (Lines 308-
310) to report the discrepancy.

2. Generally, I found that the data of starved condition added little to the major message. Unless
the model can recapitulate the even steeper gradient in such condition by tuning starvation-
related parameters, it may be removed.

**ANS:** We thank the reviewer for this suggestion.

As shown in Fig. 4D-H, varying kinetic rate constants do not lead to significantly steeper
gradients but change the length that the same slope is reached. Therefore, we agree that further
analysis of cells growing under other conditions likely will not change the main conclusions.

Therefore, we decide to make this section concise, containing only one additional condition,
even though we have more data than presented here.

The statement “steeper λ_N may help the cells cope with irregular oscillations” is removed based on
our new simulation results.

3. The authors need to compare what was different/novel between the model in this study and
previous models such as Wu, et al 2015 and highlight the uniqueness of this work.

**ANS:** Thank you for this suggestion.

We now provide a comprehensive comparison between them in the Supplemental Information
(Lines 26-147).

We would like to emphasize that although the goal of the previous works was to measure the
spatiotemporal distribution of oscillating MinD concentration gradients as a function of cell
length, these works conceived the problem differently and therefore used different experimental
designs and execution methods, which differentiates our key conclusions from theirs. This is also
true for mathematical modelling. Although similar observations can be found in some respects,
they are not directly comparable due to the different mathematics and assumptions used in the
simulations. Therefore, we would like to draw attention to the experimental rigor and to the
specific points and views that contribute to our understanding of Min systems.

4. The model explored parameter space of reaction rates and found 60 sets. The KdE, KD, KdD,
KADP-ATP ranged 6 orders of magnitude. It is interesting data in itself, but cells were not likely
to vary that much for reaction rates. The relevance should be discussed.

**ANS:** Thank you for pointing out this problem of the differences between rate constants.

We re-ran the simulation to improve the screening process, allowing us to identify parameter sets
that generate features resembling the experimental measurements. Using set #2728 as an
example, the variations in the five rate constants k_{de} , k_{dD} , k_{dE} , k_D , and $k_{ADP \rightarrow ATP}$ fall within a
small range (Table 2, S4), eliminating the concern that arose from the previous version of the
manuscript.

Minor comments:

1. Fig.1B colors were conflicting. The legend was different than diagram. Fig.1C no scale for x
axis.

**ANS:** We have resolved the colour conflict in Fig. 1B, and a time range has been added to Fig.
1C.

2. Fig.S6A How the 638 oscillatory parameter sets were matched with experimental data and
screened to 174 sets was not clear. Data of fitting error <0.12 and slope <2 were filtered. Authors
should explain the criterion for data filtering.

**ANS:** Thank you for your suggestions to improve the description of the screening process. In this
revision, we have re-run the simulation to refine and improve the results, and the corresponding
text and illustration are provided in the main text (Lines 235-277, 636-675, and Fig. S6). This

operation allowed us to probe for the general behaviours of the system. The mentioned filter no
longer applies.

3. Significant digits were not used properly. For example, the period (table 1) was showed as
46.00 sec, but the imaging interval was 12 sec, the 2 decimal digits were thus meaningless. The
same argument goes for length measurement at 2.84 μm , while the optical resolution of the
microscope used should be no good than 200nm.

**ANS:** We have corrected this significant digit throughout the manuscript.

4. For scatter plot like Fig.1D-G, generally smaller dots would show trend more obvious.

**ANS:** We have modified the plots and used smaller dots in Figs. 1D-G, 3B, C, E, F, S3D, and
S5B, C.

5. The molecular mechanism of why MinD gradient increases with length was not the scope of
the current study, but better to be discussed.

**ANS:** My apology for this confusion. The molecular mechanism is no doubt important in
explaining the length-dependent MinD gradient, although it is not easy to cover all aspects in one
paper. With the improved modelling procedure and simulation results, the possible mechanisms
associated with oscillation cycle maintenance and length-dependent variable concentration
gradients are discussed in the manuscript (Lines 446-471; Fig. 4, 5).

Our simulation results suggest that both the rate constants k_{de} , representing detachment of the
MinDE complex from the membrane, and $k_{ADP \rightarrow ATP}$, representing recharging of MinD-ADP
with ATP can effectively tune oscillation period (Fig. 4E,G). We learn from literature that MinE
shows different structural features and conformations that contribute to its interaction with MinD.
The cytosolic latent form of MinE sequesters the MinD-interacting domain in a β -stranded
conformation in the core-region of the MinE dimer (Cai *et al*, 2019; Ghasriani *et al*, 2010; King
*et al*, 2000). Upon contact with MinD or membranes, the MinD-interacting domain is exposed
and refolded into an α -helix that interacts with the MinD dimer, forming the MinD-MinE
complex on the membrane surface (Park *et al*, 2011; Wu *et al*, 2011). This process also involves
a critical step where the N-terminus of MinE that is unstructured in the cytosol and folds into an
amphipathic helix when in contact with the membrane (Hsieh *et al*, 2010; Shih *et al*, 2011). In
this step, MinD is in its ATP-bound form in the complex. Subsequently, this interaction
stimulates the ATPase activity in MinD, causing ATP hydrolysis and conformational switches on
the membrane surface. This eventually leads to the detachment of MinD and MinE, which then
recycle back into the cytosol (Hu *et al*, 2002). At this step, MinD is in its ADP-bound form,
which can be recharged with ATP for the next interaction cycle with MinE and the membranes.
Exactly how the nucleotide exchange takes place in cytosol is unknow.

To address the reviewer's question regarding how MinE-stimulated detachment of MinD from
the membrane contributes to the oscillation period under low glucose conditions, we speculate
that stress responses associated with low glucose conditions likely signal to the cell division
process, including the Min system, through uncharacterized regulatory pathways. Such
regulation may involve stabilizing and destabilizing factors of the MinD-MinE complex on the
membrane, which changes the kinetic rate constant k_{de} to alter the oscillation period. In other

words, our simulation result demonstrating the effects of varying k_{de} on the oscillation period
(Fig. 4E) may reflect cellular conditions influenced by stabilizing and destabilizing factors of the
MinD-MinE complex on the membrane. This is a direction currently under explored but will be
worth of further investigation after this work.

6. Fig. S8, why sudden jump in period in many of the sets of both groups?

**ANS:** This supplemental figure is now Fig. S7. A slower oscillation at the initiation of oscillation
appears to be a common property of the system.

Reviewer #2 (Significance (Required)):

Min system was well-studied oscillation mechanism to restrict FtsZ at cell center. Previous work
has shown how the system work molecularly, simulated the behavior and reconstituted many
different patterns in vitro. The major new information from this work was: 1. the rigorously
measured endogenous level of MinD and MinE; 2. gradient increased with length; 3. a model
recapitulated this relationship and explored parameter space of reaction rates.

The paper was well presented, experiments and analysis were rigorous, and the conclusions
were not overstated. It should interest specialized cell biologists studying cell size, oscillation
pattern.

**ANS:** Many thanks to Reviewer 2 for recognizing the contributions of our work to the
understanding of the Min system and its role in cell division. We also thank you for identifying
professional cell biologists studying cell size and oscillation patterns as readers of our paper.

We would like to emphasize that cellular concentration gradients play a fundamental role in
various cellular processes and that the concept of concentration gradients is crucial in cell
biology. These concentration gradient-mediated processes allow cells to respond to their
environment, regulate their internal conditions and perform important functions required for
survival. In addition, the variable concentration gradient, characterized by the numerical
descriptor λ_N and reproduced in a simple mathematical model, demonstrates a nonlinear
dynamics behaviour in physical biology.

Therefore, the audience of this work may include a broader audience in the field of cell biology
and physical biology rather than just an immediate specialist audience. We will also reiterate the
importance of specialized research, which often provides the basis for broader application and
understanding.

Reviewer #3 (Evidence, reproducibility and clarity (Required)):

The manuscript shows that the concentration of MinD does not change during the division cycle
of *E. coli*. Due to the oscillation pattern the concentration of MinD decreases at the mid-cell
which makes it favorable for the division. The mid-cell decrease in concentration of MinD is
majorly length dependent. The oscillation pattern is not due to the change in concentration of
MinD, but due to the plasticity arises from the spatial differences in molecular interactions
between MinD and MinE. The manuscript is well written, the experiments are performed

carefully and the results will be of interest to readers from variety of field. However, there are
several concerns need explanation.

**ANS:** We greatly appreciate the positive feedback from the reviewer, and we address the specific
concerns below.

Major concerns:

One of my major concern is these interactions are not shown experimentally but explained using
either the previously published literature or mathematical models.

**ANS:** We thank the reviewer for reminding us that the reaction rates of Min oscillations have not
been experimentally tested in previous studies or in our model. Accurately measuring these
reaction rates requires advanced techniques and methods to handle spatial and temporal
resolution, which are beyond our current capabilities and the scope of this study. However, we
are aware of the lack of experimental measurements and acknowledge the need for such data.

Further, the previous literatures are shown on in vitro models which does not mimic the in vivo
system fully.

**ANS:** We also thank the reviewer for pointing out the gap between observations in mathematical
modelling and *in vivo* systems in previous studies. To address the reviewers' concerns in this
study, we have re-run the simulation to improve the modeling procedures and results as discussed
above (This response letter, lines 371-391). As shown in Fig. S6, we implemented three filters
based on experiments, including period ≥ 15 sec, pole-to-pole, and λ_N values between 1.2 and 3.
As a result, we were able to obtain a few parameter sets, including #2728, that generate features
of the oscillation period, λ_N and I_{Ratio} , that better mimic MinD oscillation in cells (Figs. 4C, S7-
9), although not exactly the same.

We further tested the impact of different kinetic constants, k_{de} , k_{dD} , k_{dE} , k_D , and $k_{ADP \rightarrow ATP}$,
which represent different molecular interactions, on influencing the oscillation period, λ_N and
I_{Ratio} (Fig. 4D-H). These findings have provided us with a theoretical view of how oscillation
features may be controlled by different molecular interactions that influence the gradient shapes.

The concentration of MinD does not change with the increasing length of the cell. Is the MinD
concentration (or copy numbers) is different in the case of cells growing in low glucose and
when compared to the cells growing at high glucose?

**ANS:** Thank you for your question. We did not measure the MinD concentration under low-
glucose condition, although it may be interesting. We will keep this in mind for possible future
investigations to fully study the effects of nutrients.

As per the current study a particular I-ratio at the mid-cell is required to initiate the cell division.
In the case of cells growing at low glucose, how this required I-ratio is achieved at the mid-cell?

**ANS:** Thank you for the excellent question. As described in the main text, lines 197-199, I_{Ratio}
is defined as the ratio of the minimum intensity to the maximum intensity measured from the

experimental data, which gradually decreases as the cell length increases (Fig. 3C). Since the
minimum and maximum intensities were measured from the concentration gradient, which is
characterized by the slope of the concentration gradient (λ_N), there exists a correlation between
I_{Ratio} and λ_N . That is, a larger λ_N will result in a smaller I_{Ratio} .

When comparing measurements made from cells grown with 0.4% and 0.1% glucose (Fig. 3B, C,
E, F), the changes in λ_N are more drastic within a shorter length range under low-glucose
condition, which is accompanied by more drastic changes in I_{Ratio} . Furthermore, when the I_{Ratio}
value was approximately 0.5, the corresponding cell length was significantly shorter under low-
glucose condition. We hypothesize that there may be an effective I_{Ratio} that is low enough for
stable FtsZ ring formation. Based on our results, this effective I_{Ratio} in cells growing under low-
glucose condition can occur at shorter cell lengths, necessitating a faster reduction in I_{Ratio} and
division in shorter cells. Please see the main text, lines 389-406.

There is decrease in the MinD oscillation time observed in low glucose condition. As explained
by the authors the MinD oscillation is mainly guided by the FtsE induced removal of MinD from
the membrane, how the authors can explain this decrease?

**ANS:** Thank you for raising the question of how the MinE-induced detachment of membrane-
bound MinD contributes to the oscillation time of MinD under low-glucose conditions.

Our simulation results suggest that both the rate constants k_{de} , representing detachment of the
MinDE complex from the membrane, and $k_{ADP \rightarrow ATP}$, representing recharging of MinD-ADP
with ATP can effectively tune oscillation period (Fig. 4E,G). We learn from literature that MinE
shows different structural features and conformations that contribute to its interaction with MinD.
The cytosolic latent form of MinE sequesters the MinD-interacting domain in a β -stranded
conformation in the core-region of the MinE dimer (Cai *et al.*, 2019; Ghasriani *et al.*, 2010; King
*et al.*, 2000). Upon contact with MinD or membranes, the MinD-interacting domain is exposed
and refolded into an α -helix that interacts with the MinD dimer, forming the MinD-MinE
complex on the membrane surface (Park *et al.*, 2011; Wu *et al.*, 2011). This process also
involves a critical step where the N-terminus of MinE that is unstructured in the cytosol and
folds into an amphipathic helix when in contact with the membrane (Hsieh *et al.*, 2010; Shih *et*
*al.*, 2011). In this step, MinD is in its ATP-bound form in the complex. Subsequently, this
interaction stimulates the ATPase activity in MinD, causing ATP hydrolysis and conformational
switches on the membrane surface. This eventually leads to the detachment of MinD and MinE,
which then recycle back into the cytosol (Hu *et al.*, 2002). At this step, MinD is in its ADP-
bound form, which can be recharged with ATP for the next interaction cycle with MinE and the
membranes. Exactly how the nucleotide exchange takes place in cytosol is unknown.

To address the reviewer's question regarding how MinE-stimulated detachment of MinD from
the membrane contributes to the oscillation period under low glucose conditions, we speculate
that stress responses associated with low glucose conditions likely signal to the cell division
process, including the Min system, through uncharacterized regulatory pathways. Such
regulation may involve stabilizing and destabilizing factors of the MinD-MinE complex on the
membrane, which changes the kinetic rate constant k_{de} to alter the oscillation period. In other
words, our simulation result demonstrating the effects of varying k_{de} on the oscillation period
(Fig. 4E) may reflect cellular conditions influenced by stabilizing and destabilizing factors of the

MinD-MinE complex on the membrane. This is a direction currently under explored but will be
worth of further investigation after this work.

Please see main text, lines 446–471.

Further, it is explained that the concentration of cellular ATP is in much higher concentration
compared to the required amount for this oscillation. **ANS:** Noted.

As the I ratio is majorly dependent on the cell length, what could be the reason for the differential
λN in the case of low and high glucose condition?

**ANS:** Please refer to the previous answer to the question: ‘As per the current study a particular I -
ratio at the mid-cell ...’. (This response letter, Lines 571-578)

MinD is a highly insoluble protein. It also has an amphipathic helix and thus most of the time it
binds to the membrane. The method used by the author to determine the cellular MinD
concentration (mentioned in Fig S1) will only give the concentration of the soluble MinD and not
of the total MinD. How the authors justify this as the total concentration. This is also the same in
the case of MinE copy number calculation. Authors may need to perform the transcriptome
analysis and compare both the data.

**ANS:** We thank the reviewer for the comments. Since the attachment of MinD and MinE to the
membrane is transient and MinD-membrane interactions require ATP, we expect that most of the
protein would be released from the membrane into the cytoplasm after cell disruption. The MinD
concentration determined from the soluble fraction separated from the total cell lysate is
expected to adequately represent the total MinD concentration. Furthermore, our measurements
of molecule numbers are within the range of previous measurements (Di Ventura & Sourjik,
2011; Juarez & Margolin, 2010; Meacci & Kruse, 2005; Tostevin & Howard, 2006; Touhami *et*
*al*, 2006), supporting our measurements are reliable and sufficient for subsequent interpretation.
Therefore, we think that direct measurement of cellular protein abundance is reliable and
sufficient for our purposes in this study.

Transcriptome analysis has its limitations. RNA abundance measured by transcriptome analysis
does not directly correlate with protein abundance in living cells due to complexities arising from
posttranscriptional processing, translation, posttranslational modifications, and protein stability
issues. Therefore, measuring protein abundance from cell extracts is more straightforward for our
purposes.

One of the main question asked by the authors in the abstract is. "How the intracellular Min
protein concentration gradients are coordinated with cell growth to achieve spatiotemporal
accuracy of cell division is unknown". Although the authors have shown that there is a change in
concentration gradient during cell growth, the mechanism for the same is not very well
explained.

**ANS:** Thank you for pointing out the shortcomings in the writing. In the revised manuscript, we
now provide a discussion of possible mechanisms based on protein structures and protein-protein

and protein-membrane interactions (Main text, lines 446-471). See also this response letter, lines
582–611.

Authors have not provided any specific explanation for the increase in the velocity of the MinD
oscillation and the gradient formation. How the velocity of MinD is increasing although there is
no increase in the MinD concentration.

**ANS:** Since in this manuscript, the velocity is derived from the measured oscillation period and
cell length, we report the velocity only at the beginning of the Results section and do not discuss
velocity further. Instead, we focus on how the oscillation period is maintained during the cell
growth cycle. The kinetic properties and possible mechanism that can critically influence the
oscillation period is discussed as above (this response letter, lines 583–612) and in the main text
(Lines 446–471). To avoid confusion, we have modified the text and tone down the velocity
when mentioned.

Figure 2B: shows the overall concentration of MinD in a single cell varies between 1180 - 1160
molecules/ μm^2 . In Fig 2C it is mentioned that mid-cell has a MinD concentration of 120-20
molecules/ μm^2 . Further, Fig3C and 3F shows I-ratio values varies between 0.6-0.4.
Considering the values given the I-ratio (I min/ I max) should be between 0.1- 0.01. Authors
need to explain the same.

Figure 2C: The data in both the Y-axes are not matching and needs more clarification in the
legend. Whether the number of molecules were counted only in the marked 200 nm area? If so,
why the Y-axis 1 (molecules/ μm^2) is decreasing 7 times, whereas, Y-axis 2 (molecules) is only
by 2 times.

**ANS:** Thank you for these questions. In this work, the fluorescence intensity of sfGFP-MinD
was converted into molecule numbers based on estimates from Western blot analyses (Fig. S1).
The number of molecules for MinD and MinE was assumed to be the mean number and was fit
into the midpoint of the doubling time (Fig. 2B, black dashed line; Main text, lines 162-165). Fig.
2C was obtained by further processing the same dataset to restrict the region of analysis to the
midcell zone (200–nm range). Please refer to the main text, lines 156-176.

On the other hand, the λ_N and I_{Ratio} values were calculated from the processed intensity data
without conversion into molecule numbers (Fig. S2; Main text, lines 188-207, 555-581).

Therefore, due to the conversion from intensity to molecule number in Figs. S2B, C and the
image processing procedure applied to the calculation of λ_N and I_{Ratio} , no reasonable comparison
can be made for the fold change and the upper and lower limits between molecule numbers and
the λ_N and I_{Ratio} values.

Other comments:

Line 84: Requires reference for this statement.

**ANS:** A recent review article has been added in the main text, line 80: '(Cameron & Margolin,
2024)'.

Line 96: Can authors provide other evidence or validation for the determination of the copy
numbers such as transcriptome analysis.

**ANS:** Please refer to this response letter, lines 637-641.

Fig 1C: what is the units of time in Fig 1C? Is it equal for all the cell lengths?

**ANS:** As described in the main text, lines 533-534, ‘Time-lapse images of sfGFP-MinD were
acquired at 12-sec intervals for 10 min or before the fluorescence diminished’. This condition is
applied to all the acquired images in this work.

Page 6, line 136-138: what could be the possible mechanism for change in velocity at different
cell cycle time?

**ANS:** In response to questions about velocity, please refer to this response letter, lines 656-662.

For the kinetic properties and possible mechanism that may influence the oscillation period,
please refer to this response letter, lines 583–612. The discussion is based on protein structures
and protein-protein and protein-membrane interactions and their relationship to kinetic rate
constants in the mathematical model (Main text, lines 446-471).

Page 7, line 155: Any evidence for claiming the same?

**ANS:** To avoid confusion, the sentence has been modified as follows: ‘Thus, the fairly stable
oscillation period and variable velocity did not change the precision of the septum placement.’
(Main text, lines 153-154)

Page 7, line 156: Is there any proof authors can show that burst MinD synthesis occurs during
the division?

**ANS:** The text is now in the main text, lines 166-169: ‘Interestingly, the value after division was
not doubled, which could indicate a balanced outcome between *de novo* synthesis and
degradation or a burst of MinD synthesis at cell division followed by constant synthesis.’, which
is an inference based on our observations.

If not in the case of MinD, is it shown in any other protein?

**ANS:** Thank you for the question. In previous studies by Männik et al. (2018) (Mannik *et al*,
2018) and Vischer et al. (2015) (Vischer *et al*, 2015), the division protein FtsZ increased the
cellular concentration throughout the cell cycle under slow growth conditions and degraded
rapidly at the end of the cell cycle, a process controlled by the ClpXP protease. Because we do
not know the relevance of these observations to our study, which focused on the plasticity of the
MinD concentration gradient, we decided not to discuss them in the manuscript.

Page 9, line 217: The Fig 4A is not explained clearly and all the terms mentioned needs to be
explained. This figure is used to explain the differential concentration of MinD at the poles and
the mid-cell, thus needs to be explain more clearly.

**ANS:** Thank you for your comments. Terms for the rate constants, concentrations, and diffusion
coefficients in the mathematical model are now explained in the main text, lines 239-262. The
relationships between reaction steps and rate constants are also shown in Materials and Methods,
lines 611-626.

Page 12, line 285: What is meaning of default speed of MinD oscillation in new-born cells? Do
the authors observed any specific velocity in the new-born cells?

**ANS:** Sorry for the confusion. The statement is based on our observations and analyses as shown
in Fig. S4, which are discussed in the main text, lines 147-154. Due to the small sample size of
cells right at splitting, it is not possible to conclude a specific period and a specific velocity.
Therefore, the sentence has been removed from the revised manuscript to avoid unnecessary
confusion.

What is the explanation for length dependent oscillation velocity for MinD?

**ANS:** Thank you for the question. In response to questions about velocity, please refer to this
response letter, lines 656-662.

For the kinetic properties and possible mechanism that may influence the oscillation period,
please refer to this response letter, lines 583–612. The discussion is based on protein structures
and protein-protein and protein-membrane interactions and their relationship to kinetic rate
constants in the mathematical model (Main text, lines 441-467).

Reviewer #3 (Significance (Required)):

General assessment: Major work of the manuscript is relying on the mathematical models,
whereas the audience are majorly from the biology fields and thus simplified explanations are
required in many places. Many of the legends in the figures require more explanation for better
understanding. If possible more experimental data can be added, specifically to explain the
model mentioned in figure 4A.

**ANS:** Thank you for your suggestions. We have modified the figure legends to include more
explanations. As mentioned above, terms for the rate constants, concentrations, and diffusion
coefficients in the mathematical model are now explained in the main text, lines 236-262. The
relationships between reaction steps and rate constants are also shown in Materials and Methods,
lines 611-626.

We have also revised Fig. 4 to include improvements in modelling results to better fit the
experimental data and to examine the impacts of the kinetics constants of the reaction steps in the
Min system.

Advance: The study is adding to the existing knowledge and will be helpful to fill the conceptual
gaps in understanding the mid-cell MinD concentration and what may favor the initiation of

bacterial division.

Audience: Majorly the microbiology community will be interested in the study. This will also be
interest to Physicists and mathematical persons working to understand bacterial division.

**ANS:** We thank the reviewer for these positive comments.

Reviewer #4 (Evidence, reproducibility and clarity (Required)):

The study by Parada et al. illuminates the intricate interplay between Min proteins, exemplified
by MinD, and cell growth in *E. coli*. Their findings demonstrate that the MinD concentration
gradient steepens progressively as cells elongate, potentially influencing FtsZ ring formation via
MinC. Moreover, their comprehensive reaction-diffusion model not only corroborates
experimental observations of length-dependent concentration gradients but also underscores
the critical role of kinetic interactions involving Min proteins, the membrane, and ATP. This
elucidation significantly advances our understanding of the oscillatory mechanisms within the
Min system. Both the experimental and simulation data are robust, and the manuscript is
exceptionally well-written. I express my full support for publication pending the satisfactory
resolution of the outlined concerns.

**ANS:** We appreciate the reviewer's positive comments and feedback and have addressed your
questions to the best of our ability.

1. Remove the dot in front of "Min" in line 57.

**ANS:** This has now been removed.

2. In lines 82-84, the statement "...The distribution of the division inhibitor MinC may be
synchronized with spatiotemporal differences in MinD concentrations, leading to a stable
placement of the FtsZ ring at the midcell..." suggests a potential synchronization between MinC
and MinD oscillations. It is crucial to investigate if sfGFP-MinC exhibits similar concentration
gradient oscillatory behavior in vivo as observed with MinD.

**ANS:** Thank you for bringing up the interests in MinC. However, with many investigations
already covered in this manuscript, we prefer to investigate sfGFP-MinC in future studies, which
will have different focuses on how MinC dynamics are linked to the variable MinD
concentration gradient to directly impact FtsZ ring formation. We would like to keep the main
focus of this study as plasticity in the MinD concentration gradient, which results from spatial
differences in molecular interactions and is an intrinsic property of the Min system during cell
growth.

3. Ensure consistent significant digits throughout the text. For instance, $1.95 \pm 0.16 \mu\text{M}$ in line 97,
$1.4 \pm 0.13 \mu\text{M}$ in line 98, and $1.9 \pm 0.2 \mu\text{M}$ in line 100 have varying precision. Consider using
integers for molecules.

**ANS:** We have corrected the significant digits in the main text and supplemental information.

4. Address the discrepancy in expression levels of MinD and MinE between strain FW1541 and

its parental strain W3110. Given the labeling effect, it is possible that MinD expression levels
differ. However, MinC's expression level should be approximately the same. Conduct whole-
genome sequencing of both strains to identify any additional mutations.

**ANS:** Thank you for the comments. As emphasized in the main text (Lines 63-64), the most
important aspect contributing to MinDE oscillations is the concentration ratio between MinD and
MinE. There may not be an absolute number of MinD (and MinE) molecules, but there is a range
within which variation can be tolerated as long as the ratio of MinD to MinE is maintained. In
addition, although the numbers obtained from W3110 and FW1541 are not the same, they are all
comparable to those in previous studies (Hale *et al*, 2001; Li *et al*, 2014; Schmidt *et al*, 2016;
Shih *et al*, 2002) (Main text, lines 109-113).

Furthermore, we have performed whole-genome sequencing of the W3110 and FW1541 strains.
We confirm that sfGFP was correctly inserted. The sequence alignment of the *minCDE* locus is
provided for your reference as a part of this response letter. Although there are some sporadic
mutations, there is no obvious reason to believe that the mutations would impact Min protein
expression.

5. Clarify the apparent discrepancy between lines 112 and 127. Line 112 suggests that the
periodic regularity of interpolar oscillations increases with cell length, as demonstrated in Fig
1B-C, 1E, Fig S5. However, in the subsequent section (starting from line 127), the authors state
that oscillation periods remain relatively stable across cells of different lengths. Provide
clarification on this apparent discrepancy.

**ANS:** Thank you for pointing out this confusion caused by misuse of the term.

In Lines 120-122, the statement has been modified as follows: ‘...the uniformity of the
oscillation intervals appears to increase with length...’

In line 137, ‘*The oscillation period*’ refers to the time required for the oscillation cycle. Since the
correction in line 123 should be enough to clarify, we do not modify the statement in line 139.

6. Specify if the analysis was limited to non-constricted cells. If so, state this explicitly in the text,
as it could impact the interpretation of results, especially in relation to the linear dependence of
cell length on time before constriction, as shown in Fig S3C.

**ANS:** We did not specifically remove those constricted cells, but cells before splitting were
considered one cell. We have added a statement to clarify in Lines 142-143.

7. Improve clarity in Fig 2A by using distinct colors (e.g., green and red) for differentiation on the
Y-axis.

**ANS:** The Y axes of Fig. 2A have been modified.

8. Correct "of" to "from" in line 223 for improved clarity and accuracy.

**ANS:** Corrected.

9. Include the missing "A" in Fig S6A for completeness and accuracy.

**ANS:** This figure has been updated.

10. Ensure consistency in referencing style (full names versus short names) throughout the
manuscript.

**ANS:** This has now been done.

Reviewer #4 (Significance (Required)):

While numerous commendable in vitro studies have explored the oscillatory behavior of the Min
system, this work uniquely delves into the oscillation of MinD within live cells. It unveils the
remarkable coordination between intracellular Min protein concentration gradients and cell
growth, shedding light on the precise spatiotemporal regulation of cell division.

**ANS:** We thank the reviewer for this positive comment.

**References**

Cai M, Huang Y, Shen Y, Li M, Mizuuchi M, Ghirlando R, Mizuuchi K, Clore GM (2019)

Probing transient excited states of the bacterial cell division regulator MinE by relaxation
dispersion NMR spectroscopy. *Proc Natl Acad Sci USA* 116: 25446-25455

Di Ventura B, Sourjik V (2011) Self-organized partitioning of dynamically localized proteins in
bacterial cell division. *Mol Syst Biol* 7: 457

Fischer-Friedrich E, Meacci G, Lutkenhaus J, Chate H, Kruse K (2010) Intra- and intercellular
fluctuations in Min-protein dynamics decrease with cell length. *Proc Natl Acad Sci USA*
107: 6134-6139

Ghasriani H, Ducat T, Hart CT, Hafizi F, Chang N, Al-Baldawi A, Ayed SH, Lundstrom P,
Dillon JA, Goto NK (2010) Appropriation of the MinD protein-interaction motif by the
dimeric interface of the bacterial cell division regulator MinE. *Proc Natl Acad Sci USA* 107:
18416-18421

Hale CA, Meinhardt H, de Boer PA (2001) Dynamic localization cycle of the cell division
regulator MinE in *Escherichia coli*. *EMBO J* 20: 1563-1572

Hsieh CW, Lin TY, Lai HM, Lin CC, Hsieh TS, Shih YL (2010) Direct MinE-membrane
interaction contributes to the proper localization of MinDE in *E. coli*. *Mol Microbiol* 75:
499-512

Hu Z, Gogol E, Lutkenhaus J (2002) Dynamic assembly of MinD on phospholipid vesicles
regulated by ATP and MinE. *Proc Natl Acad Sci USA* 99: 6761-6766

Juarez JR, Margolin W (2010) Changes in the Min oscillation pattern before and after cell birth.
*J Bacteriol* 192: 4134-4142

- King GF, Shih YL, Maciejewski MW, Bains NPS, Pan B, Rowland S, Mullen GP, Rothfield LI
 (2000) Structural basis for the topological specificity function of MinE. *Nat Struct Biol* 7:
 1013-1017
- Li GW, Burkhardt D, Gross C, Weissman JS (2014) Quantifying absolute protein synthesis rates
 reveals principles underlying allocation of cellular resources. *Cell* 157: 624-635
- Mannik J, Walker BE, Mannik J (2018) Cell cycle-dependent regulation of FtsZ in *Escherichia*
 *coli* in slow growth conditions. *Mol Microbiol* 110: 1030-1044
- Meacci G, Kruse K (2005) Min-oscillations in *Escherichia coli* induced by interactions of
 membrane-bound proteins. *Phys Biol* 2: 89-97
- Meacci G, Ries J, Fischer-Friedrich E, Kahya N, Schwille P, Kruse K (2006) Mobility of Min-
 proteins in *Escherichia coli* measured by fluorescence correlation spectroscopy. *Phys Biol* 3:
 255-263
- Park KT, Wu W, Battaile KP, Lovell S, Holyoak T, Lutkenhaus J (2011) The Min oscillator uses
 MinD-dependent conformational changes in MinE to spatially regulate cytokinesis. *Cell* 146:
 396-407
- Schavemaker PE, Boersma AJ, Poolman B (2018) How Important Is Protein Diffusion in
 Prokaryotes? *Front Mol Biosci* 5: 93
- Schmidt A, Kochanowski K, Vedelaar S, Ahrne E, Volkmer B, Callipo L, Knoops K, Bauer M,
 Aebersold R, Heinemann M (2016) The quantitative and condition-dependent *Escherichia*
 *coli* proteome. *Nat Nanotechnol* 34: 104-110
- Shih YL, Fu X, King GF, Le T, Rothfield L (2002) Division site placement in *E. coli*: mutations
 that prevent formation of the MinE ring lead to loss of the normal midcell arrest of growth
 of polar MinD membrane domains. *EMBO J* 21: 3347-3357
- Shih YL, Huang KF, Lai HM, Liao JH, Lee CS, Chang CM, Mak HM, Hsieh CW, Lin CC (2011)
 The N-terminal amphipathic helix of the topological specificity factor MinE is associated
 with shaping membrane curvature. *PLoS One* 6: e21425
- Tostevin F, Howard M (2006) A stochastic model of Min oscillations in *Escherichia coli* and
 Min protein segregation during cell division. *Phys Biol* 3: 1-12
- Touhami A, Jericho M, Rutenberg AD (2006) Temperature dependence of MinD oscillation in
 *Escherichia coli*: running hot and fast. *J Bacteriol* 188: 7661-7667
- Vischer NO, Verheul J, Postma M, van den Berg van Saparoea B, Galli E, Natale P, Gerdes K,
 Luirink J, Vollmer W, Vicente M *et al* (2015) Cell age dependent concentration of
 *Escherichia coli* divisome proteins analyzed with ImageJ and ObjectJ. *Front Microbiol* 6:
 586

- Wu F, van Schie BG, Keymer JE, Dekker C (2015) Symmetry and scale orient Min protein
patterns in shaped bacterial sculptures. *Nat Nanotechnol* 10: 719-726
- Wu W, Park KT, Holyoak T, Lutkenhaus J (2011) Determination of the structure of the MinD-
ATP complex reveals the orientation of MinD on the membrane and the relative location of
the binding sites for MinE and MinC. *Mol Microbiol* 79: 1515-1528
- Zieske K, Chwastek G, Schwille P (2016) Protein patterns and oscillations on lipid monolayers
and in microdroplets. *Angewandte Chemie* 55: 13455-13459

TGATTGACAAGGGTATTTTTTAAGCTATGAATCAGCGCCATTTATCACAGAATAGACTTTTACTCTGAATAAATGGGAGGGTGACTTGCTCAAT
ACTAACTGTTCCCATAAAAAATTCGATACTTAGTCGCGGTAATAGTGTCTTATCTGAAAATGAGACTTATTTACCTCCCACTGAACGGAGTTA

1 -> TGATTGACAAGGGTATTTTTTAAGCTATGAATCAGCGCCATTTATCACAGAATAGACTTTTACTCTGAATAAATGGGAGGGTGACTTGCTCAAT
2 -> TGATTGACAAGGGTATTTTTTAAGCTATGAATCAGCGCCATTTATCACAGAATAGACTTTTACTCTGAATAAATGGGAGGGTGACTTGCTCAAT

ATAATCCAGACTATAACATGCCTTATAGTCTTCGGAACATCATCGCGCGCTGGCGATGATTAATAGCTAATTGAGTAAGGCCAGGATGTCAAACA
TATTAGGTCTGATATTGTACGGAATATCAGAAGCCTTGTAGTAGCGCGGACCGCTACTAATTATCGATTAACCTCATTCCGGTCTACAGTTTGT

1 -> ATAATCCAGACTATAACATGCCTTATAGTCTTCGGAACATCATCGCGCGCTGGCGATGATTAATAGCTAATTGAGTAAGGCCAGGATGTCAAACA
2 -> ATAATCCAGACTATAACATGCCTTATAGTCTTCGGAACATCATCGCGCGCTGGCGATGATTAATAGCTAATTGAGTAAGGCCAGGATGTCAAACA

CGCCAATCGAGCTTAAAGGCAGTAGCTTCACTTTATCTGTGGTTTCATCTGCATGAGGCAGAACCTAAGGTTATCCATCAGGCGCTGGAAGACAAA
GCGGTTAGCTCGAATTTCCGTCATCGAAGTGAAATAGACACCAAGTAGACGTA CTCCGTC TTGGATTCCAATAGGTAGTCCGCGACCTTCTGTTT

1 -> CGCCAATCGAGCTTAAAGGCAGTAGCTTCACTTTATCTGTGGTTTCATCTGCATGAGGCAGAACCTAAGGTTATCCATCAGGCGCTGGAAGACAAA
2 -> CGCCAATCGAGCTTAAAGGCAGTAGCTTCACTTTATCTGTGGTTTCATCTGCATGAGGCAGAACCTAAGGTTATCCATCAGGCGCTGGAAGACAAA

580 590 600 610 620 630 640 650 660
ATCGCTCAGGCCCCCGCATTTTTAAAACATGCCCCCGTTGTAICTCAACGTCAGTGCCTGGAAGACCCGGTAAACTGGTCAGCGATGCATAAGGC
TAGCGAGTCCGGGGGCGTAAAAATTTGTACGGGGCAACATGAGTTGCAGTCACGTGACCTTCTGGGCCATTTGACCAGTCGCTACGTATTCCG
I A Q A P A F L K H A P V V L N V S A L E D P V N W S A M H K A
minC

1 → Bpu10I EcoO109I ApaI NsiI
ATCGCTCAGGCCCCCGCATTTTTAAAACATGCCCCCGTTGTAICTCAACGTCAGTGCCTGGAAGACCCGGTAAACTGGTCAGCGATGCATAAGGC
TAGCGAGTCCGGGGGCGTAAAAATTTGTACGGGGCAACATGAGTTGCAGTCACGTGACCTTCTGGGCCATTTGACCAGTCGCTACGTATTCCG
I A Q A P A F L K H A P V V L N V S A L E D P V N W S A M H K A
minC

2 → Bpu10I EcoO109I ApaI NsiI
ATCGCTCAGGCCCCCGCATTTTTAAAACATGCCCCCGTTGTAICTCAACGTCAGTGCCTGGAAGACCCGGTAAACTGGTCAGCGATGCATAAGGC
TAGCGAGTCCGGGGGCGTAAAAATTTGTACGGGGCAACATGAGTTGCAGTCACGTGACCTTCTGGGCCATTTGACCAGTCGCTACGTATTCCG
I A Q A P A F L K H A P V V L N V S A L E D P V N W S A M H K A
minC

670 680 690 700 710 720 730 740 750 760
GGTTTCGGCAACCGGTTTGGGGTTATTGGCGTAAGTGCTGCAAAGATGCGCAACTTAAAGCCGAAATTGAAAAGATGGGGCTGCCTATCTGA
CCAAAGCCGTTGGCCAAACGCCCAATAACCGCATTACCGACGTTTCTACGCGTTGAATTTGCGCTTTAACTTTTCTACCCCGACGGATAGGACT
V S A T G L R V I G V S G C K D A Q L K A E I E K M G L P I L
minC

1 → GGTTTCGGCAACCGGTTTGGGGTTATTGGCGTAAGTGCTGCAAAGATGCGCAACTTAAAGCCGAAATTGAAAAGATGGGGCTGCCTATCTGA
GGTTTCGGCAACCGGTTTGGGGTTATTGGCGTAAGTGCTGCAAAGATGCGCAACTTAAAGCCGAAATTGAAAAGATGGGGCTGCCTATCTGA
CCAAAGCCGTTGGCCAAACGCCCAATAACCGCATTACCGACGTTTCTACGCGTTGAATTTGCGCTTTAACTTTTCTACCCCGACGGATAGGACT
V S A T G L R V I G V S G C K D A Q L K A E I E K M G L P I L
minC

2 → GGTTTCGGCAACCGGTTTGGGGTTATTGGCGTAAGTGCTGCAAAGATGCGCAACTTAAAGCCGAAATTGAAAAGATGGGGCTGCCTATCTGA
GGTTTCGGCAACCGGTTTGGGGTTATTGGCGTAAGTGCTGCAAAGATGCGCAACTTAAAGCCGAAATTGAAAAGATGGGGCTGCCTATCTGA
CCAAAGCCGTTGGCCAAACGCCCAATAACCGCATTACCGACGTTTCTACGCGTTGAATTTGCGCTTTAACTTTTCTACCCCGACGGATAGGACT
V S A T G L R V I G V S G C K D A Q L K A E I E K M G L P I L
minC

1 → ATGACTTCTTCAAGTCCGCCATGCCGGAAGGCTATGTGCAGGAACGCACGATTTCTTTAAGGATGACGGCACGTACAAAACGGTGCGGAAGTG

1 → AAATTTGAAGGCGATACCCTGGTAAACCGCATTGAGCTGAAAGGCATTGACTTTAAGAAGACGGCAATATCCTGGGCCATAAGCTGGAATACAA

1 → TTTTAAACAGCCACAATGTTTACATCACCGCCGATAAACAAAAAATGGCATTAAAGCGAATTTTAAAATTCGCCACAACGTGGAGGATGGCAGCG

linker

linker

GGCTGGAGCTGCTTCGAAGTTCTATACTTTCTAGAGAATAGGAACTTCGGAATAGGAACTTCAAGATCCCCTTATTAGAAGAAGCTCGTCAAGAA
CCGACCTCGACGAAGCTTCAAGGATATGAAAGATCTCTTATCCTTGAAGCCTTATCCTTGAAGTTCTAGGGGAATAATCTTCTTGAGCAGTTCTT

1 → GGCTGGAGCTGCTTCGAAGTTCTATACTTTCTAGAGAATAGGAACTTCGGAATAGGAACTTCAAGATCCCCTTATTAGAAGAAGCTCGTCAAGAA

2 → GGCTGGAGCTGCTTCGAAGTTCTATACTTTCTAGAGAATAGGAACTTCGGAATAGGAACTTCAAGATCCCCTTATTAGAAGAAGCTCGTCAAGAA
CCGACCTCGACGAAGCTTCAAGGATATGAAAGATCTCTTATCCTTGAAGCCTTATCCTTGAAGTTCTAGGGGAATAATCTTCTTGAGCAGTTCTT

GGCGATAGAAGGCGATGCGCTGCGAATCGGGAGCGGCGATACCGTAAAGCACGAGGAAGCGGTCAGCCCCATTCGCCGCCAAGCTCTTCAGCAATA
CCGCTATCTTCCGCTACGCGACGCTTAGCCCTCGCCGCTATGGCATTTCGTGCTCCTTCGCCAGTCGGGTAAGCGGCGGTTTCGAGAAGTCGTTAT

R Y F A I R Q S D P A A I G Y L V L F R D A W E G G L E E A I
NeoR/KanR

1 → GGCGATAGAAGGCGATGCGCTGCGAATCGGGAGCGGCGATACCGTAAAGCACGAGGAAGCGGTCAGCCCCATTCGCCGCCAAGCTCTTCAGCAATA

2 → GGCGATAGAAGGCGATGCGCTGCGAATCGGGAGCGGCGATACCGTAAAGCACGAGGAAGCGGTCAGCCCCATTCGCCGCCAAGCTCTTCAGCAATA
CCGCTATCTTCCGCTACGCGACGCTTAGCCCTCGCCGCTATGGCATTTCGTGCTCCTTCGCCAGTCGGGTAAGCGGCGGTTTCGAGAAGTCGTTAT

BsrBI BssSaI

R Y F A I R Q S D P A A I G Y L V L F R D A W E G G L E E A I
NeoR/KanR

TCACGGGTAGCCAACGCTATGTCCTGATAGCGGTCCGCCACACCCAGCCGGCCACAGTCGATGAATCCAGAAAAGCGGCCATTTCCACCATGAT
AGTGCCCATCGGTTGCGATACAGGACTATCGCCAGGCGGTGTGGGTCGGCCGGTGTGAGCTACTTAGGCTTTTTCGCCGGTAAAAGGTTGGTACTA

D R T A L A I D Q Y R D A V G L R G C D I F G S F R G N E V M I
NeoR/KanR

1 → TCACGGGTAGCCAACGCTATGTCCTGATAGCGGTCCGCCACACCCAGCCGGCCACAGTCGATGAATCCAGAAAAGCGGCCATTTCCACCATGAT

2 → TCACGGGTAGCCAACGCTATGTCCTGATAGCGGTCCGCCACACCCAGCCGGCCACAGTCGATGAATCCAGAAAAGCGGCCATTTCCACCATGAT
AGTGCCCATCGGTTGCGATACAGGACTATCGCCAGGCGGTGTGGGTCGGCCGGTGTGAGCTACTTAGGCTTTTTCGCCGGTAAAAGGTTGGTACTA

RsrII

D R T A L A I D Q Y R D A V G L R G C D I F G S F R G N E V M I
NeoR/KanR

CTGCCCGGCACTTCGCCCAATAGCAGCCAGTCCCTTCCCGCTTCAAGTACAACTCGAGCACAGCTGCGCAAGGAACGCCCGTCGTGGCCAGCC

1 →

2 →

PfFI
Tth111I

MscI

ACGATAGCCGCGCTGCCTCGTCTCAGTTCATTACAGGGCACCGGACAGGTCGGTCTTGACAAAAGAACCAGGGCGCCCTGCGCTGACAGCCGG

1 →

2 →

DrdI

AACACGGCGGCATCAGAGCAGCCGATTGTCTGTTGTGCCAGTCATAGCCGAATAGCCTCTCCACCCAAGCGGCCGAGAACCTGCGTGCAATCC

1 →

2 →

EagI

3890 3900 3910 3920 3930 3940 3950 3960 3970 3980

ATCTTGTTCAATCATGCGAAACGATCCTCATCCTGTCTCTTGATCAGATCTTGATCCCCTGCGCCATCAGATCCTTGCGGCAAGAAGCCATCC
 TAGAACAAAGTTAGTACGCTTTGCTAGGAGTAGGACAGAGAAGTCTAGAACTAGGGGACGCGGTAGTCTAGGAACCGCGTTCTTTCGGTAGG

5 1
 D Q E I M
 NeoR/KanR

1 → ATCTTGTTCAATCATGCGAAACGATCCTCATCCTGTCTCTTGATCAGATCTTGATCCCCTGCGCCATCAGATCCTTGCGGCAAGAAGCCATCC

2 →

BglII

ATCTTGTTCAATCATGCGAAACGATCCTCATCCTGTCTCTTGATCAGATCTTGATCCCCTGCGCCATCAGATCCTTGCGGCAAGAAGCCATCC
 TAGAACAAAGTTAGTACGCTTTGCTAGGAGTAGGACAGAGAAGTCTAGAACTAGGGGACGCGGTAGTCTAGGAACCGCGTTCTTTCGGTAGG

5 1
 D Q E I M
 NeoR/KanR

3990 4000 4010 4020 4030 4040 4050 4060 4070

AGTTTACTTTGCAGGGCTTCCAACCTTACCAGAGGGCGCCCCAGCTGGCAATTCCGGTTCGCTTGCTGTCCATAAAACCGCCAGTCTAGCTAT
 TCAAATGAAACGTCCTCGAAGGGTTGGAATGGTCTCCCGCGGGGTCGACCGTTAAGGCCAAGCGAACGACAGGTATTTTGGCGGGTCAGATCGATA

1 → AGTTTACTTTGCAGGGCTTCCAACCTTACCAGAGGGCGCCCCAGCTGGCAATTCCGGTTCGCTTGCTGTCCATAAAACCGCCAGTCTAGCTAT

2 →

AGTTTACTTTGCAGGGCTTCCAACCTTACCAGAGGGCGCCCCAGCTGGCAATTCCGGTTCGCTTGCTGTCCATAAAACCGCCAGTCTAGCTAT
 TCAAATGAAACGTCCTCGAAGGGTTGGAATGGTCTCCCGCGGGGTCGACCGTTAAGGCCAAGCGAACGACAGGTATTTTGGCGGGTCAGATCGATA

4080 4090 4100 4110 4120 4130 4140 4150 4160 4170

CGCCATGTAAGCCCACTGCAAGCTACCTGCTTTCTCTTTGCGCTTGCGTTTTCCCTTGTCAGATAGCCCAAGTAGCTGACATTCATCCGGGGTCA
 GCGGTACATTCGGGTGACGTTTCGATGGACGAAAGAGAAACGCGAACGCAAAAGGGAACAGGTCTATCGGGTCATCGACTGTAAGTAGGCCCCAGT

1 → CGCCATGTAAGCCCACTGCAAGCTACCTGCTTTCTCTTTGCGCTTGCGTTTTCCCTTGTCAGATAGCCCAAGTAGCTGACATTCATCCGGGGTCA

2 →

BtsaI

CGCCATGTAAGCCCACTGCAAGCTACCTGCTTTCTCTTTGCGCTTGCGTTTTCCCTTGTCAGATAGCCCAAGTAGCTGACATTCATCCGGGGTCA
 GCGGTACATTCGGGTGACGTTTCGATGGACGAAAGAGAAACGCGAACGCAAAAGGGAACAGGTCTATCGGGTCATCGACTGTAAGTAGGCCCCAGT

GCACCGTTTCTGCGGACTGGCTTTCTACGTGTTCCGCTTCCTTTAGCAGCCCTTGCGCCCTGAGTGCTTGCGGCAGCGTGAGCTTCAAAAAGCGCT
CGTGGCAAAGACGCCTGACCGAAAGATGCACAAGGCGAAGGAAATCGTCGGGAACGCGGGACTCACGAACGCCGTCGCACTCGAAGTTTTCGCGA

1 → GCACCGTTTCTGCGGACTGGCTTTCTACGTGTTCCGCTTCCTTTAGCAGCCCTTGCGCCCTGAGTGCTTGCGGCAGCGTGAGCTTCAAAAAGCGCT

2 → GCACCGTTTCTGCGGACTGGCTTTCTACGTGTTCCGCTTCCTTTAGCAGCCCTTGCGCCCTGAGTGCTTGCGGCAGCGTGAGCTTCAAAAAGCGCT
CGTGGCAAAGACGCCTGACCGAAAGATGCACAAGGCGAAGGAAATCGTCGGGAACGCGGGACTCACGAACGCCGTCGCACTCGAAGTTTTCGCGA

CTGAAGTTCTATACTTTCTAGAGAATAGGAACTTCGAACTGCAGGTCGACGGATCCCCGGAATTGAGCCCGCTGTAAAAGCGCATTTATCTTCA
GACTTCAAGGATATGAAAGATCTCTTATCCTTGAAGCTTGACGTCCAGCTGCCTAGGGGCCTTAACTCGGGCGACATTTTCGCGTAAATAGAAGT

FRT (minimal)

1 → CTGAAGTTCTATACTTTCTAGAGAATAGGAACTTCGAACTGCAGGTCGACGGATCCCCGGAATTGAGCCCGCTGTAAAAGCGCATTTATCTTCA

-----AAAAGCGCATTTATCTTCA
-----TTTTTCGCGTAAATAGAAGT

2 → CTGAAGTTCTATACTTTCTAGAGAATAGGAACTTCGAACTGCAGGTCGACGGATCCCCGGAATTGAGCCCGCTGTAAAAGCGCATTTATCTTCA
GACTTCAAGGATATGAAAGATCTCTTATCCTTGAAGCTTGACGTCCAGCTGCCTAGGGGCCTTAACTCGGGCGACATTTTCGCGTAAATAGAAGT

SalI AccI BamHI

FRT (minimal)

AGGCAGAGTTATCTCTGCCTTGAGTTTTTCATCCCTCTCATCCACGTTGTGGTAAAGCGGGCGAGTATTCTTGCTGATACTCCTCATTGCTATTTTC
TCCGTCTCAATAGAGACGGAACCTCAAAAAGTAGGGAGAGTAGGTGCAACACCATTTGCGCGCTCATAAGAACGACTATGAGGAGTAACGATAAAG

1 → AGGCAGAGTTATCTCTGCCTTGAGTTTTTCATCCCTCTCATCCACGTTGTGGTAAAGCGGGCGAGTATTCTTGCTGATACTCCTCATTGCTATTTTC
TCCGTCTCAATAGAGACGGAACCTCAAAAAGTAGGGAGAGTAGGTGCAACACCATTTGCGCGCTCATAAGAACGACTATGAGGAGTAACGATAAAG

2 → AGGCAGAGTTATCTCTGCCTTGAGTTTTTCATCCCTCTCATCCACGTTGTGGTAAAGCGGGCGAGTATTCTTGCTGATACTCCTCATTGCTATTTTC
TCCGTCTCAATAGAGACGGAACCTCAAAAAGTAGGGAGAGTAGGTGCAACACCATTTGCGCGCTCATAAGAACGACTATGAGGAGTAACGATAAAG

4460 4470 4480 4490 4500 4510 4520 4530 4540 4550

CACCTCCCCCTTTACACCTTAAGGCTGTTCCACGCACAAAACAAAATGTTTATGCCTGGTTGAGTAAATAACCTTATTGTTAGTATGGATATAC
GTGGAGGGGGAAATGTGGAATTCGACAAGGGTGCGTGTGTTTGTGTTTACAAATACGGACCAACTCATTATTGGAATAACAATCATACCTATATG

1 -> ANII
CACCTCCCCCTTTACACCTTAAGGCTGTTCCACGCACAAAACAAAATGTTTATGCCTGGTTGAGTAAATAACCTTATTGTTAGTATGGATATAC
GTGGAGGGGGAAATGTGGAATTCGACAAGGGTGCGTGTGTTTGTGTTTACAAATACGGACCAACTCATTATTGGAATAACAATCATACCTATATG
2 -> ANII
CACCTCCCCCTTTACACCTTAAGGCTGTTCCACGCACAAAACAAAATGTTTATGCCTGGTTGAGTAAATAACCTTATTGTTAGTATGGATATAC
GTGGAGGGGGAAATGTGGAATTCGACAAGGGTGCGTGTGTTTGTGTTTACAAATACGGACCAACTCATTATTGGAATAACAATCATACCTATATG

4560 4570 4580 4590 4600 4610 4620 4630 4640

TGGGAGATGATAACCCCGCAATTTCAATTATTCATTGTTCTCCCATAGGATGAGGTTCCCGCCGTAAC TGCGGGCTTTTTTTGCCCCAGAATTT
ACCCTCTACTATTGGGGCGTTAAAGTAATAAGTAACAAGAGGGTATCCTACTCCAAGGGCGGCATTGACCGCCGAAAAAACGGGGTCTTAAA

1 -> TGGGAGATGATAACCCCGCAATTTCAATTATTCATTGTTCTCCCATAGGATGAGGTTCCCGCCGTAAC TGCGGGCTTTTTTTGCCCCAGAATTT
TGGGAGATGATAACCCCGCAATTTCAATTATTCATTGTTCTCCCATAGGATGAGGTTCCCGCCGTAAC TGCGGGCTTTTTTTGCCCCAGAATTT
ACCCTCTACTATTGGGGCGTTAAAGTAATAAGTAACAAGAGGGTATCCTACTCCAAGGGCGGCATTGACCGCCGAAAAAACGGGGTCTTAAA
2 -> TGGGAGATGATAACCCCGCAATTTCAATTATTCATTGTTCTCCCATAGGATGAGGTTCCCGCCGTAAC TGCGGGCTTTTTTTGCCCCAGAATTT
ACCCTCTACTATTGGGGCGTTAAAGTAATAAGTAACAAGAGGGTATCCTACTCCAAGGGCGGCATTGACCGCCGAAAAAACGGGGTCTTAAA

4650 4660 4670 4680

End (4687)

TCCCTTTCAACATCCTGTA AACGAAAAC TGCGCCGAAGCGCAG 3'
AGGGAAAGTTGTAGGACATTTGCTTTTGACGCGGCTTCGCGTC 5'

1 -> End (2623)
TCCCTTTCAACATCCTGTA AACGAAAAC TGCGCCGAAGCGCAG 3'
AGGGAAAGTTGTAGGACATTTGCTTTTGACGCGGCTTCGCGTC 5'
2 -> End (4687)
TCCCTTTCAACATCCTGTA AACGAAAAC TGCGCCGAAGCGCAG 3'
AGGGAAAGTTGTAGGACATTTGCTTTTGACGCGGCTTCGCGTC 5'

Original Sequence: minCDE W3110-FW1541.dna

- 1: 7776_22 -> 2623 bases 1..2623 (2 gaps)
2: 7776_25 -> 4687 bases 1..4687

W3110-FW1541
4687 bp

August 21, 2024

Re: JCB manuscript #202406107T

Prof. Yu-Ling Shih
Institute of Biological Chemistry, Academia Sinica
128 Sec. 2 Academia Road, Nankang
Taipei 115
Taiwan

Dear Prof. Shih,

Thank you for submitting your revised manuscript entitled "Growth-dependent concentration gradient of the oscillating Min system in *Escherichia coli*." The manuscript has been seen by the original referees from Review Commons whose full comments are appended below.

All reviewers recognize the novelty of your work, particularly your quantitative approach and the insight of your model, which suggests that velocity of Min oscillation, rather than Min oscillation period, is length-dependent. They are supportive of your paper's publication in JCB but have requested improvements in the accuracy of data presentation and interpretation. Most comments do not require new experiments, but we encourage you to carefully address their feedback.

We also agree with Reviewer 1's comment that the conclusion on Line 28 needs to be rephrased for accuracy. The current phrasing, "This study explores the plasticity of these gradients due to spatial variations in molecular interactions, an intrinsic property of the Min system," is misleading. The length-dependence of gradient likely reflects the phase shift between local oscillatory reactions, which manifests as the speed at which waves travel through the excitable medium. The data presently do not address (also do not need to involve) the presence of spatial variations in molecular interactions, which would require additional analysis of oscillation periods at different locations within the same cell. Moreover, attributing this phenomenon to the intrinsic properties of the Min system (as mentioned on Line 82) might be an overstatement. The possibility that the altered global excitability could be regulated by a yet-to-be-discovered mechanism could not be excluded.

Our general policy is that papers are considered through only one revision cycle; however, given that the suggested changes are relatively minor we are open to one additional short round of revision. Please note that we will expect to make a final decision without additional reviewer input upon resubmission.

Please submit the final revision within one month, along with a cover letter that includes a point by point response to the remaining reviewer comments.

Thank you for this interesting contribution to Journal of Cell Biology. You can contact me or the scientific editor listed below at the journal office with any questions at cellbio@rockefeller.edu.

Sincerely,

Min Wu, PhD
Monitoring Editor
Journal of Cell Biology

Dan Simon, PhD
Scientific Editor
Journal of Cell Biology

Reviewer #1 (Comments to the Authors (Required)):

I think the revised manuscript properly addressed the issues that I raised. The novelty of this work is well highlighted. Therefore, I would recommend this work for publication in Journal of Cell Biology. One small comment is that I am still not convinced by the description at L28, "This study explores the plasticity of these gradients due to spatial variations in molecular interactions, an intrinsic property of the Min system." because kinetic parameters in the model are spatially uniform implying that molecular interactions do not vary in space.

Reviewer #2 (Comments to the Authors (Required)):

This revised manuscript has significantly improved modeling, using a more physiological relevant parameter set, and clearer writing and presentation (especially new Table S4 and Fig.S6). Together with the already interesting experimental data, they should warrant publication in JCB, but there are still minor imperfections that better be addressed.

1. The significant digits were mostly corrected throughout the text but still in Table 1 and Fig 1, the period is in two decimal points, beyond the precision (12-sec interval) of their measurement. Same argument for length.
2. In the rebuttal letter, the authors said they only keep data from 0.1% Glu as an additional nutrition condition, but in the figure 1 and 3 it is 0% Glu. Please clarify.
3. Line 892, "P: two-tailed probability" is not in the figure.
4. Line 76, concomitantly.
5. Line 212, "oscillation period slowed". Either "oscillation slowed" or "oscillation period increased".
6. The meaning of each parameter should also be written in the legend of Table. S4, not only in the main text.
7. The argument in line 227-234 is not convincing. Basically it says that the total amount of ATP is enough for minDE consumption, but it is reaction rate that matters and it is also dependent on the ATP concentration. Lower ATP could effectively reduce the conversion from minD_ADG to minD_ATP conversion in reality. I suggest toning down a little, saying it is still possible. Besides, I can't find the source of consumption rate of $\sim 0.9 \mu\text{mole}/\mu\text{mole MinD}/\text{sec}$ from (Shih et al., 2011).
8. Fig S4, how can you get something like 40.599s period at such high precision but only from 15-min imaging interval?

Reviewer #3 (Comments to the Authors (Required)):

The manuscript by Parada et al, determined the MinD number, concentration and oscillation in different cell lengths and in different growth conditions. The authors claim similar concentration and oscillation time period during different growth times and for different cell lengths. They have also described the differentially presence of MinD at poles and mid-cell. Authors further identified Iratio, the ratio of concentration between mid-cell and poles, which plays important roles during the placement of the Z-ring. Finally, authors have presented a mathematical in silico model showing the different possible stages of MinD required for oscillation. Overall, the study is interesting, performed well elaborately and is providing good insight about the Min oscillation process. The study will be of interest to many who are working in the bacterial division. However, the drawbacks of the study is that the in silico data is not supported or verified by experimental data. Further, the study is contradicting several previous studies, for example, which says that MinD oscillation time period are not constant for different cells. A major comment by the reviewer 1 (line 122, reviewer's comment) is not well explained by the authors.

Average cell length with the standard deviation need to be mentioned for with (Fig. 1E) or without (1D) glucose.

Line 128: Whether 0.127 $\mu\text{m}/\text{sec}$ is the average velocity of all cell lengths or a specific cell length? Also provide the SD here with all the average data.

Explain, why the oscillation period is less for cells growing in the presence of 0.4% glucose (Fig. 1E and 1G)?

Line 166: Why the value should be doubled? Authors claim that there was no change in the concentration of MinD. Thus it should also stay same after the division.

Line 171: Please explain how the calculation was performed in the case of mid cell concentration calculation? How the area near the mid-cell was calculated in the Figure 2B and C?

Considering width of bacteria 0.968 μm (Fig. S3E) and 200nm from both the sided of mid-cell, the area becomes $(0.968 \times 0.4) = 0.387 \mu\text{m}^2$.

Number of molecules at mid cell = 220 (Fig 3C)

Thus number of molecules/ μm^2 should be $(220 / 0.387 \mu\text{m}^2) = 568/\mu\text{m}^2$

Please clarify (Fig. 3C)

Line 204: Authors may discuss about the range of "I-ratio" at the mid-cell is required for the initiation of the septum formation.

Line 215: "treated with 0% glucose" should be mention as "in the absence of glucose".

Line 217: Did the authors determine the cell lengths during glucose starvation condition?

The change in the cell length described in Fig 3A/ 3D is only 5% different. This is well within the error level. How the authors will explain this?

Line 299: In silico oscillation resembles oscillation in a cellular context: If the λN is biphasic (Fig. 4C) the I_{ratio} also should be biphasic. You may also check the FigS9, which is showing biphasic I_{ratio} . Please explain the same.

Line 318: λN is not biphasic in the in vitro conditions.

Line 387: Can the authors provide any possible mechanisms for the increase of velocity of Min oscillation in the longer cell length and discuss the same in this section?

Addition of glucose should increase the cell size significantly.

Line 397: All the data shows with 0.4% glucose. Why it is mentioned 0.1% glucose here?

Line 461: Can authors provide any evidence that MinD-ATP has higher affinity for MinE than MinD-ADP? Or this explanation is an assumption.

Bacterial cell number calculation, please explain the same.

Table S3: Please recheck and explain the calculations performed.

In example: for FW1541

Dry weight for 50 ml = 5.6 mg

1 ml = 0.112 mg = 0.112×10^{-3} g

total cell per 1 ml = 7.96×10^8

single cell weight = $[0.112 \times 10^{-3}] / [7.96 \times 10^8]$ g

= 0.014×10^{-11} g

= 1.14×10^{-13} g

Cells / 40 μ g lysate = $[40 \times 10^{-6}] / [1.14 \times 10^{-13}]$

= 35×10^7

Fig 4B, shows much lower oscillation time, i.e., 6 complete oscillations in 100 sec, whereas, the actual time for a complete oscillation is around 50 sec for 4.6 μ m cells. Similarly Fig S7 shows different ranges for oscillations (10-50 sec). Can authors explain this?

Fig. S4 shows that for many cells the oscillation period is very high, which is a contradiction to the author's statement that oscillation period does not vary much for different cell lengths. Can the authors explain these discrepancies?

Fig S5: panels in the figure are not explained in the figure legend. Why the oscillation period shown in panel C is so high?

Figure 5B panel numbering is missing.

In the reviewer's comment (Line 134), Reviewer 1: authors have mentioned that the oscillation period among the cells are varying between 36.8 to 65.6 sec, which is almost double. However, this data is not mentioned and it is not clarified in the main manuscript.

Reviewer 3 (line 621, reviewer's comment): I agree with the reviewer that MinD is an insoluble protein and all the protein will not come to soluble fraction at any condition. This section is not well answered by the authors.

Reviewer #4 (Comments to the Authors (Required)):

The authors have addressed some of my concerns but not others.

1. The significant digits are inconsistent throughout the text. The authors have not adequately addressed the concerns I and Reviewer #2 raised. For instance: in line 103, use 2205 ± 178 molecules instead of $2,205 \pm 178.3$ molecules since molecular numbers should be integers. In line 104, change $1.4 \pm 0.13 \mu\text{M}$ to $1.40 \pm 0.13 \mu\text{M}$. In line 106, change $1.9 \pm 0.2 \mu\text{M}$ to $1.9 \pm 0.2 \mu\text{M}$. Additionally, the comma within the numbers should be removed to maintain consistency throughout the manuscript.

2. In Fig. 2A, the green and red lines appear to be simply overlaid on top of the original figure, which diminishes the quality. Moreover, the text resolution in the figure is too low.

3. Ensure consistency in referencing style (e.g., full names versus short names) throughout the manuscript.

New minor concerns:

1. Could the authors explain the reason for the significant decrease in MinD velocity observed in newborn cells, as mentioned in line 150?

2. In line 254, it should be #2827 instead of #2728.

3. Include the missing "B" in Fig 5 for completeness and accuracy.

Response to reviewers

Reviewer #1 (Comments to the Authors (Required)):

I think the revised manuscript properly addressed the issues that I raised. The novelty of this work is
well highlighted. Therefore, I would recommend this work for publication in Journal of Cell
Biology. One small comment is that I am still not convinced by the description at L28, "This study
explores the plasticity of these gradients due to spatial variations in molecular interactions, an
intrinsic property of the Min system." because kinetic parameters in the model are spatially uniform
implying that molecular interactions do not vary in space.

*Ans: Thank you very much for your positive feedback. Your comments and suggestions have been*
*valuable in improving our work. Regarding the important point raised at line 28, we have revised*
*the sentence in the manuscript as follows.*

*Lines 28-29: 'This study explores the plasticity of the MinD gradients resulting from the*
*interdependent interplay between molecular interactions and diffusion in the system.'*

Reviewer #2 (Comments to the Authors (Required)):

This revised manuscript has significantly improved modeling, using a more physiological relevant
parameter set, and clearer writing and presentation (especially new Table S4 and Fig.S6). Together
with the already interesting experimental data, they should warrant publication in JCB, but there are
still minor imperfections that better be addressed.

*Ans: Thank you very much for your positive feedback. Your comments and suggestions have been*
*valuable in improving our work.*

1. The significant digits were mostly corrected throughout the text but still in Table 1 and Fig 1, the
period is in two decimal points, beyond the precision (12-sec interval) of their measurement. Same
argument for length.

*Ans: We have corrected the significant digits in the manuscript following the rules explained below.*

*Period: There are 2 significant digits, as the images were taken at 12-sec or 15-min intervals.*

*Length and width (μm): Our microscopy system generates digital images with a pixel dimension of*
*$0.0645 \mu\text{m} \times 0.0645 \mu\text{m}$, allowing measurements to go down to four decimal places. However, to*
*keep the table concise, we report measurements to three decimal places.*

*For velocity ($\mu\text{m}/\text{sec}$), area (μm^2), and aspect ratio, we follow the same rules as those of the*
*measured numbers used in their calculations.*

2. In the rebuttal letter, the authors said they only keep data from 0.1% Glu as an additional
nutrition condition, but in the figure 1 and 3 it is 0% Glu. Please clarify.

*Ans: Thank you for pointing out this mistake. We confirm that we used 0% glucose in this work and*
*have made corrections in the manuscript (Lines 218-220).*

3. Line 892, "P: two-tailed probability" is not in the figure.

*Ans: "P: two-tailed probability" is removed (Line 932).*

4. Line 76, concomitantly.

*Ans: Corrected (Line 77).*

5. Line 212, "oscillation period slowed". Either "oscillation slowed" or "oscillation period
increased".

Ans: It is corrected as "oscillation period increased" (Lines 216-217).

6. The meaning of each parameter should also be written in the legend of Table. S4, not only in the
main text.

Ans: The meaning of each parameter is now provided in the footnote of Table S4 (SI lines 391-
400).

7. The argument in line 227-234 is not convincing. Basically it says that the total amount of ATP is
enough for minDE consumption, but it is reaction rate that matters and it is also dependent on the
ATP concentration. Lower ATP could effectively reduce the conversion from minD_ADP to
minD_ATP conversion in reality. I suggest toning down a little, saying it is still possible.

Ans: Thank you for the suggestion. The sentences are re-written with softer tone as below.

Lines 231-233: 'While fluctuations in intracellular ATP concentration caused by glucose downshifts
are not expected to significantly impact the ATP needed for MinD oscillation, we cannot entirely
rule out this possibility.'

Lines 237-239: 'Therefore, even during glucose starvation, the cellular ATP levels appear to be
substantially higher than the amount consumed by MinD oscillations.'

Besides, I can't find the source of consumption rate of $\sim 0.9 \mu\text{mole}/\mu\text{mole MinD}/\text{sec}$ from (Shih et
al., 2011).

Ans: ' $0.9 \mu\text{mole}/\mu\text{mole MinD}/\text{sec}$ ' is a number converted from $30 \text{ nmole}/\text{mg MinD}/\text{min}$, which is
obtained by rounding off from $33.1 \pm 3.8 \text{ nmole}/\text{mg MinD}/\text{min}$ as reported in (Shih et al., 2011).

We made a correction to put the original number $\sim 30 \text{ nmole}/\text{mg MinD}/\text{min}$ into the sentence and 0.9
$\mu\text{mole}/\mu\text{mole MinD}/\text{sec}$ in parentheses.

Lines 236-237: 'approximately $30 \text{ nmole Pi}/\text{mg MinD}/\text{min}$ ($\sim 0.9 \mu\text{mole Pi}/\mu\text{mole MinD}/\text{sec}$)'

8. Fig S4, how can you get something like 40.599s period at such high precision but only from 15-
67 min imaging interval?

Ans: The digits are corrected in Fig. S4.

**Reviewer #3 (Comments to the Authors (Required)):**

The manuscript by Parada et al, determined the MinD number, concentration and oscillation in
different cell lengths and in different growth conditions. The authors claim similar concentration
and oscillation time period during different growth times and for different cell lengths. They have
also described the differentially presence of MinD at poles and mid-cell. Authors further identified
Iratio, the ratio of concentration between mid-cell and poles, which plays important roles during the
placement of the Z-ring. Finally, authors have presented a mathematical in silico model showing the
different possible stages of MinD required for oscillation. Overall, the study is interesting,
performed well elaborately and is providing good insight about the Min oscillation process. The
study will be of interest to many who are working in the bacterial division.

Ans: We sincerely thank you for your critical comments, which have helped to further improve this
manuscript. Your time and effort are greatly appreciated.

However, the drawbacks of the study is that the in silico data is not supported or verified by
experimental data.

Ans: Thank you for reminding us that some parameters are yet to be tested by experiments. Table 2
summarizes the parameters that generated result #2827, as well as their sources. We agree with and
are aware of the general concerns regarding the use of hypothetical values and unconfirmed
parameters in relating simulation studies, including ours. In addition, the missing experimental
evidence for the predicted rate constants in our results will require further investigation using
advanced imaging techniques, ideally at the single-cell level, to achieve in the near future.

Notably, we were able to improve the simulation results by adjusting the kinetic rate constants
closer to the physiological range. This improvement is due to the introduction of an experimentally
measured factor, the wave slope (λ_N), to constrain the simulation process, along with a few other
general features of MinD oscillation.

Further, the study is contradicting several previous studies, for example, which says that MinD
oscillation time period are not constant for different cells.

Ans: Thank you for the discussion.

This study measured MinD oscillation in a population of unsynchronized cells. The experimental
measurements showed heterogeneous distribution in lengths and oscillation periods, etc., among
different cells, similar to previous studies.

After sorting the data by cell length, we observed that the MinD oscillation periods were relatively
close across cells of varying lengths (Fig. 1E). Since cell length is positively correlated with growth
time (Fig. S3C), we transformed the period-length correlation from Fig. 1E into a period-time
correlation, as shown in Fig. S3D. Consequently, we learn that the oscillation period remains
relatively stable throughout the cell cycle. Therefore, oscillation period remains relatively stable
within a growing cell, which exhibits different lengths at different time points. Our findings do not
apply to comparisons of oscillation periods between different cells. The relating context is in the
main text, lines 139-150.

The section title is modified to better describe our observations.

Lines 139-140: ‘The oscillation period is quite stable within a growing cell exhibiting different
lengths at different time points’

A major comment by the reviewer 1 (line 122, reviewer's comment) is not well explained by the
authors.

Ans: Thank you for the kind reminder. We think the reviewer may refer to the novelty for the broad
audience needs to be better explained as previously suggested. Here, we have revised the response
provided in the previous rebuttal letter, Lines 117-124, 383-395, as follows.

The novelty of our study is the discovery of the plasticity of the MinD concentration gradient, a
critical factor in bacterial cell division. By exploring the parameter space of kinetic rate constants
that represent different molecular interactions within the Min system, we reveal how these
interactions lead to variable concentration gradients during cell growth. This concept of variable
cellular concentration gradients not only enhances our understanding of bacterial cell division but
also offers broader insights into essential cellular processes such as transport, signaling, and
homeostasis, which rely on concentration gradients. The significance of this research extends
beyond the specialized audience focused on Min systems, providing relevance to the broader fields
of cell biology and physical biology by demonstrating new examples of nonlinear dynamics of
cellular function.

To better emphasize the significance in the manuscript, we also modified the last paragraph in the
Introduction.

Lines 92-99: ‘Taken together, the findings of this study reveal the inherent plasticity and
adaptability of the MinD concentration gradient, a critical factor in the Min system that orchestrates
division site placement. By exploring the parameter space of kinetic rate constants, that represent

different molecular interactions within the Min system, using a numerical model, we reveal how
these interactions lead to variable concentration gradients during cell growth. This concept of
variable cellular concentration gradients not only enhances our understanding of bacterial cell
division but also offers broader insights into essential cellular processes such as transport, signaling,
and homeostasis, which rely on concentration gradients.'

Average cell length with the standard deviation need to be mentioned for with (Fig. 1E) or without
(1D) glucose.

Ans: In response to the reviewer's suggestion, we have provided both median and average \pm
standard deviation in Table 1.

Line 128: Whether 0.127 $\mu\text{m}/\text{sec}$ is the average velocity of all cell lengths or a specific cell length?
Also provide the SD here with all the average data.

Ans: Please refer to the main text, line 131. We report the median oscillation time that is used to
calculate velocity. In the revised manuscript, we have provided both median and average \pm SD in
Table 1.

Explain, why the oscillation period is less for cells growing in the presence of 0.4% glucose (Fig.
1E and 1G)?

Ans: Thank you for raising the interesting question. The slower MinD oscillation in the absence of
glucose may be part of the cellular response to glucose starvation. The underlying mechanism is of
our interest and will be investigated in future studies.

Line 166: Why the value should be doubled? Authors claim that there was no change in the
concentration of MinD. Thus it should also stay same after the division.

Ans: Thank you for pointing out the issue. The sentence has been corrected in the manuscript.

Lines 170-173: 'Interestingly, the values just before division were not doubled compared to
daughter cells, suggesting a balance between de novo synthesis and degradation, or a burst of MinD
synthesis at cell division followed by a steady rate of synthesis.'

Line 171: Please explain how the calculation was performed in the case of mid cell concentration
calculation? How the area near the mid-cell was calculated in the Figure 2B and C?

Considering width of bacteria 0.968 μm (Fig. S3E) and 200nm from both the sides of mid-cell, the
area becomes $(0.968 \times 0.4) = 0.387 \mu\text{m}^2$.

Number of molecules at mid cell = 220 (Fig 3C)

Thus number of molecules/ μm^2 should be $(220 / 0.387 \mu\text{m}^2) = 568/\mu\text{m}^2$

Please clarify (Fig. 3C)

Ans: Thank you very much for the discussion, which helps us to improve the clarity of our
manuscript.

In Fig. 2C, we calculated the number of protein molecules in the mid-cell zone based on the
fluorescent intensity profile in a gradient shape, rather than assuming a uniform distribution in the
mid-cell zone. The algorithm for calculating the center fraction of MinD molecules is now provided
in the main text, lines 585-598:

*Algorithm for calculating center fraction:* Let $\{\tilde{I}_{ij}(x)\}_{j=1}^n$ be the intensity calculated above after
correcting for photobleaching, associated with the individual intensity data and position $x \in L_j =$
$\{x_1, \dots, x_k\}$, where L_j is the set of observation positions for x and is normalized as before with
$0 = x_1 < x_2 < \dots < x_k = 1$.

We interpolate the intensities $\{\tilde{I}_{ij}(x_a)\}_{j=1}^n$ and $\{\tilde{I}_{ij}(x_b)\}_{j=1}^n$ at $x_a = 0.5 - \frac{100nm}{particle\ length}$ and
$x_b = 0.5 + \frac{100nm}{particle\ length}$ linearly with $x_1 < x_2 < \dots < x_a < \dots < x_b < \dots < x_{k+2}$.

For each j , we calculate the center intensity $I_{j,center}$ by summing the trapezoidal areas between each
interval $[x_l, x_{l+1}]$, where $x_a \leq x_l \leq x_{l+1} \leq x_b$, representing the intensity within the center plus or
minus 100 nm. The total intensity $I_{j,total}$ is calculated by summing the trapezoidal areas between
each interval $[x_l, x_{l+1}]$ for $l = 1, \dots, k + 1$. We then calculate the averages $\overline{I_{center}} = \frac{1}{20} \sum_{j=1}^{20} I_{j,center}$
and $\overline{I_{total}} = \frac{1}{20} \sum_{j=1}^{20} I_{j,total}$ and report the fraction as $\frac{\overline{I_{center}}}{\overline{I_{total}}}$ for each particle.

Then, the number of MinD molecules in the mid-cell zone and their concentration (molecules/ μm^2),
as shown in Fig. 2C, were obtained by multiplying the numbers presented in Fig. 2B by this
fraction.' (Detail in main text, lines 623-631)

As a result, the section of 'Image processing' is re-organized.

The inset illustration in Fig. 2C has also been modified to aid understanding.

A sentence is added to the main text to improve the clarify.

Line 178-179: This was calculated from the fraction of intensity in the MinD gradient profile.

Line 204: Authors may discuss about the range of "I-ratio" at the mid-cell is required for the
initiation of the septum formation.

Ans: Thank you for the suggestion. To avoid confusion, we have modified the sentences.

Lines 221-223: 'However, the I_{Ratio} values at the median cell length showed no clear difference
between nutrient shifts (Figs. 3C, F), suggesting that the I_{Ratio} values at the time of division were
similar regardless of glucose supply.'

Line 215: "treated with 0% glucose" should be mention as "in the absence of glucose".

Ans: Corrected in Lines 218-220.

Line 217: Did the authors determine the cell lengths during glucose starvation condition?

Ans: As shown in Table 1, the cell lengths were determined in both 0.416% and 0% glucose
conditions. The length informaiton of cells grown under glucose starvation was used to plot Fig. 1G
and Figs. 3D,E,F.

The change in the cell length described in Fig 3A/ 3D is only 5% different. This is well within the
error level. How the authors will explain this?

Ans: We can compare the length distributions in Fig. 3B with Fig. 3E, and in Fig. 3C with Fig. 3F.
The length range is narrower for cells grown without glucose (0%), with more cells showing shorter
lengths. As a result, the bell-shaped distribution in Fig. 3D for the 0% glucose sample is narrower
than that in Fig. 3A for the 0.416% glucose sample. The two-tail, unequal variance t-test analysis
yields a P-value of 0.010.

Line 299: In silico oscillation resembles oscillation in a cellular context: If the λN is biphasic (Fig.
4C) the I_{ratio} also should be biphasic. You may also check the Fig S9, which is showing biphasic
I_{ratio} . Please explain the same.

Ans: Thank you for your question that helps us to improve clarity of our presentation.

We have revised Figs. S8 and S9 by removing data that showed no temporal oscillation (only spatial
inhomogeneity) or no spatial oscillation (where only temporal oscillation, with the entire cell
showing uniform behavior). The previous presentation may have been somewhat misleading, as the
apparent biphasic nature of the I_{Ratio} could be due to oscillation occurring only over time, without
spatial variation (resulting in an I_{Ratio} of 1).

After revising the figures, we can observe that many panels in Fig. S9 do not show a clear biphasic
characteristic in I_{Ratio} . In principle, λ_N and I_{Ratio} are closely related, as the MinD intensity at the
mid-cell decays based on the λ_N value. However, due to the presence of basal values, the finite
basal background alters the I_{Ratio} , causing it to deviate from a biphasic pattern.

Line 318: λ_N is not biphasic in the in vitro conditions.

Ans: Thank you for your comment, but this work do not involve *in vitro* conditions.

Further, the previous literature are shown on in vitro models which does not mimic the in vivo
system fully.

Ans: Thank you for the discussion. We do not perform measurements *in vitro* or use data from in
vitro reconstitution experiments from the literature. The parameters used in our simulation are
derived from literature reporting measurements in cellular contexts, from our current study, or are
hypothetical. Their sources are listed in Table 2.

Line 387: Can the authors provide any possible mechanisms for the increase of velocity of Min
oscillation in the longer cell length and discuss the same in this section?

Ans: This study emphasises on understanding how the oscillation period remains fairly stable
during cell growth rather than how the velocity changes, since the mentioned velocity is estimated
from the measured oscillation time and cell length.

We used modelling results to elucidate the possible mechanism related to period maintenance. The
corresponding text and illustration are provided in the main text, lines 339-381, 462-478 and Figs.
4, 5.

In brief, this simulation allowed us to probe for general behaviours of the system, allowing us to
obtain a few parameter sets that generate features of the oscillation period, λ_N and I_{Ratio} , highly
mimicking MinD oscillation in the cellular context (Fig 4C, S7-9). We further tested the impact of
different kinetic constants, k_{de} , k_{dD} , k_{dE} , k_D , and $k_{ADP \rightarrow ATP}$, which represent different molecular
interactions influencing the oscillation period, λ_N and I_{Ratio} (Fig 4D-H). This effort has provided us
with a solid theoretical view of how oscillation features may be controlled by different molecular
interactions.

We also modified the sentence in lines 396-400 to mention velocity: ‘The observed phenomenon of
relatively stable oscillation periods, accompanied by increasing velocity as cells elongate, may be
partly attributed to the physical properties of the Min system, as well as potential unknown
regulatory mechanisms that could adjust the kinetic rate constants of the system.’

Addition of glucose should increase the cell size significantly.

Ans: The cell size does indeed differ with and without glucose, as reported in Table 1.

When comparing the measurements obtained with 0.416% glucose and 0% glucose, the bacteria
grown with glucose increase in width ($P < 0.0001$), length ($P < 0.0001$), and overall area ($P < 0.0001$).
However, the aspect ratio of the bacteria remains largely unchanged under these two conditions
($P = 0.382$).

Line 397: All the data shows with 0.4% glucose. Why it is mentioned 0.1% glucose here?

Ans: ‘0.1%’ has been correct to ‘0%’ or changed to ‘the absence of glucose’ in the sentence (Line
407).

Line 461: Can authors provide any evidence that MinD-ATP has higher affinity for MinE than
MinD-ADP? Or this explanation is an assumption.

Ans: Thank you for your comment. The differences in MinE binding affinity between MinD-ATP
and MinD-ADP are not assumed in our model.

For the reaction steps included in our mathematical model, please refer to the main text, lines 652-
669. The parameter values and their sources for sample #2827 are provided in Table 2.

Bacterial cell number calculation, please explain the same. Table S3: Please recheck and explain the
calculations performed.

In example: for FW1541

Dry weight for 50 ml = 5.6 mg

1 ml = 0.112 mg = 0.112×10^{-3} g

total cell per 1 ml = 7.96×10^8

single cell weight = $[0.112 \times 10^{-3}] / [7.96 \times 10^8]$ g

= 0.014×10^{-11} g

= 1.44×10^{-13} g

Cells / 40 ug lysate = $[40 \times 10^{-6}] / [1.14 \times 10^{-13}]$

= 35×10^7

Ans: We have checked the calculation and confirmed the accuracy, rewrote the paragraph in
Supplemental Information lines 276-286, and re-organized Table S3 to improve the clarify.

SI lines 276-286: ‘The single-cell weight was determined from three independent exponentially
growing cultures of FW1541 and W3110 (Table S3). Cells were collected from these cultures,
freeze-dried overnight at 10 mTorr (using an UNISS Freeze Dryer FDM-20, Taiwan Green Version
Technology Ltd., Taiwan), and then weighed. The dry weights were 5.6 ± 0.7 mg for FW1541 and
5.2 ± 0.6 mg for W3110, equivalent to 0.112 mg/mL and 0.104 mg/mL of culture, respectively.
Assuming water constitutes ~75% of cell weight (Bionumbers ID 105482;
<https://bionumbers.hms.harvard.edu/bionumber.aspx?id=105482>), the wet weight per mL of culture
was estimated to be 0.149 mg for FW1541 and 0.139 mg for W3110. Cell counts under the same
conditions were 7.96×10^8 CFU/mL for FW1541 and 7.71×10^8 CFU/mL for W3110. Consequently,
the single-cell weights were calculated to be 5.63×10^{-13} g for FW1541 and 5.40×10^{-13} g for W3110.’

Fig 4B, shows much lower oscillation time, i.e., 6 complete oscillations in 100 sec, whereas, the
actual time for a complete oscillation is around 50 sec for 4.6 um cells. Similarly Fig S7 shows
different ranges for oscillations (10-50 sec). Can authors explain this?

Ans: Thank you for the discussion regarding the shorter oscillation periods observed in the
simulation results compared to the experimental measurements.

Focusing on the gradient shape during oscillation in this work, we applied a screening criterion
based on the experimental λ_N values (ranging from 1.2 to 3), along with other general
characteristics, to filter the parameter sets (Main text, lines 332-338, Fig. S6A). This approach
allowed us to isolate a few datasets, including sample #2827, which helped us to identify a
combination of rate constants that closely replicate the experimental features.

Nonetheless, as the reviewer noted, the results are not entirely satisfactory, regarding the issue of
generally shorter oscillation periods as shown in Fig. S7.

Therefore, we further explored the parameter space of kinetic rate constants using the dataset from
sample #2827 (Fig. 4D-H). Our analysis revealed that the rate constants k_{de} and $k_{ADP \rightarrow ATP}$, which
regulate MinD and MinE detachment from the membrane and nucleotide exchange, respectively,
have a significant impact on the oscillation period and the gradient slope λ_N (Main text, lines 447-
457; Figs. 4E, G). By decreasing k_{de} or increasing $k_{ADP \rightarrow ATP}$, the oscillation period can be extended
up to 30 seconds.

While the current simulation used a minimal model incorporating the essential steps to produce
oscillations, we plan to explore more complex models in future work, which may further improve
the model to more closely align with the actual measurements. For instance, the new models will
include additional steps in the oscillation reactions, representing potential modulators of k_{de} and
$k_{ADP \rightarrow ATP}$, and introduce stochasticity in reaction and diffusion. Regarding this point, we have
added statements in the main text, lines 422-427.

In addition, we modified Figs. S7-9 by removing non-oscillatory data points to be consistent with
Figs. 4C-H. The explanation of data omission is provided in the caption of Fig. S8.

SI lines 495-499: 'Data points for non-oscillatory cases (Period=0) were omitted for both λ_N (Fig.
S8) and I_{Ratio} (Fig. S9). Non-oscillatory cases are typically homogeneous, with trivial values of λ_N
and I_{Ratio} . Occasional non-oscillatory inhomogeneous cases occurred between two poles, resulting
in non-trivial λ_N and I_{Ratio} . However, these were unrelated to oscillation and have been removed.'

We also removed a parameter set #2775, that shows only one length with a period, from Figs. S7-9.
The corresponding changes are made in Main text, lines 281, 334 and SI lines 471-472, Fig. S6A.

Fig. S4 shows that for many cells the oscillation period is very high, which is a contradiction to the
author's statement that oscillation period does not vary much for different cell lengths. Can the
authors explain these discrepancies?

Ans: Thank you for the discussion on the experiment measuring MinD oscillations before and after
cell division.

The specific method for this data collection is now added to the figure caption of Fig. S4 (SI, lines
452-455): 'To image MinD oscillations before and after cell division, cells mounted on an agarose
pad were initially imaged for 10 minutes, then incubated on the heating stage of the microscope at
30°C for 2 hours before being imaged again for 10 minutes. Divided cells were selected for
analysis.'

In response to the reviewer's question, we suspect that the increase in the oscillation period is due to
the 2-hour incubation on the slide without aeration and nutrient depletion in the agarose pad, which
are limitations of this experimental setup. These limitations could be addressed by using
microfluidic devices in future experiments.

Fig S5: panels in the figure are not explained in the figure legend.

Ans: Thank you for pointing out this missing information. We have re-wrote the caption of Fig. S5
(SI, lines 457-464).

Why the oscillation period shown in panel C is so high?

Ans: The slower oscillation observed in this experiment was likely due to the expression of
additional copies of FtsA-mScarlet-I from a plasmid in FW1541, as well as addition of antibiotics
for plasmid selection. (SI, lines 182-193)

Figure 5B panel numbering is missing.

Ans: The missing label 'B' has been added back.

In the reviewer's comment (Line 134), Reviewer 1: authors have mentioned that the oscillation
period among the cells are varying between 36.8 to 65.6 sec, which is almost double. However, this
data is not mentioned and it is not clarified in the main manuscript.

Ans: In response to the question, we believe the reviewer is referring to our previous response letter
at Line 152, where these numbers were mentioned. We did not emphasize this because the variation
in oscillation time, ranging from 36.8 to 65.6 seconds, falls within a reasonable range.

Nonetheless, we have included this information in the legend of Fig. 1E (Main text, lines 904-905).

Reviewer 3 (line 621, reviewer's comment): I agree with the reviewer that MinD is an insoluble
protein and all the protein will not come to soluble fraction at any condition. This section is not well
answered by the authors.

Ans: Thank you for the discussion.

Here, we provide further support from our own experiments. First, *minD*, *sfGFP-minD*, and *minE*
were expressed from the endogenous promoter on the chromosome chromosome to avoid the
adverse effects of overexpression. Second, during fluorescence microscopy imaging, no bright spots
or aggregates of sfGFP-MinD were observed. Third, cells were grown and collected under the same
conditions as those used for the imaging experiments, although a larger culture volume was used for
experiments of quantifying protein molecules. Our operation effectively avoided MinD aggregation.

**Reviewer #4 (Comments to the Authors (Required)):**

The authors have addressed some of my concerns but not others.

Ans: Thank you very much for your time and effort in helping us improve this manuscript.

1. The significant digits are inconsistent throughout the text. The authors have not adequately
addressed the concerns I and Reviewer #2 raised. For instance: in line 103, use 2205 {plus
minus} 178 molecules instead of 2,205 {plus minus} 178.3 molecules since molecular numbers
should be integers.

In line 104, change 1.4 {plus minus} 0.13 μM to 1.40 {plus minus} 0.13 μM .

In line 106, change 1.9 {plus minus} 0.2 μM to 1.9x {plus minus} 0.2x μM . Additionally, the comma
within the numbers should be removed to maintain consistency throughout the manuscript.

Ans: Thank you for helping us correct these details. The following changes to significant digits and
removal of commas have been made in the revised manuscript. The rules are described in this
response letter, lines 28-33.

Line 106: 2205 \pm 178 molecules/cell

Line 107: 1.40 \pm 0.13 μM

Line 108: 3532 \pm 61 molecules/cell

Line 109: 2150 \pm 228 molecules/cell; 1.90 \pm 0.20 μM

2. In Fig. 2A, the green and red lines appear to be simply overlaid on top of the original figure,
which diminishes the quality. Moreover, the text resolution in the figure is too low.

Ans: Thank you for pointing out the production issue with Fig. 2A. We have re-exported all three
plots in Fig. 2 using MATLAB to ensure consistent formatting and resolution.

3. Ensure consistency in referencing style (e.g., full names versus short names) throughout the
manuscript.

Ans: We have corrected two journal abbreviations.

Line 800-801: *Mol Microbiol.*

Line 809: *Mol Cell.*

Line 833: *Biochim. Biophys. Acta, Rev. Biomembr.*

New minor concerns:

1. Could the authors explain the reason for the significant decrease in MinD velocity observed in
newborn cells, as mentioned in line 150?

Ans: Thank you for pointing out that an explanation is missing.

We performed this experiment to test whether slower velocity would occur in shorter newborn
cells, an assumption based on our observations that the oscillation period remained relatively stable
throughout the cell cycle (Fig. S3D) and velocity increased with time (Fig. 1F). In response to the
reviewer's question, we propose that while this phenomenon may be attributed to a physical
property of the Min oscillation, there is also a possibility that unidentified regulatory mechanisms
could be involved.

The following sentences are added to the main text, lines 396-400:

'The observed phenomenon of relatively stable oscillation periods, accompanied by increasing
velocity as cells elongate, may be partly attributed to the physical properties of the Min system, as
well as potential unknown regulatory mechanisms that could adjust the kinetic rate constants of the
system.'

Please also refer to this letter lines 230-246.

2. In line 254, it should be #2827 instead of #2728.

Ans: Corrected (Line 361).

3. Include the missing "B" in Fig 5 for completeness and accuracy.

Ans: The lable has been added.

October 2, 2024

RE: JCB Manuscript #202406107R

Prof. Yu-Ling Shih
Institute of Biological Chemistry, Academia Sinica
128 Sec. 2 Academia Road, Nankang
Taipei 115
Taiwan

Dear Prof. Shih,

Thank you for submitting your revised manuscript entitled "Growth-dependent concentration gradient of the oscillating Min system in *Escherichia coli*." We would be happy to publish your paper in JCB pending final revisions necessary to meet our formatting guidelines (see details below).

A. MANUSCRIPT ORGANIZATION AND FORMATTING:

1) Text limits: Character count for Articles is < 40,000, not including spaces. Count includes title page, abstract, introduction, results, discussion, and acknowledgments. Count does not include materials and methods, figure legends, references, tables, or supplemental legends. JCB formatting does not allow for supplemental results, discussion, and methods. Please incorporate these sections into the main text.

2) Figure formatting: Articles may have up to 10 main text figures. Scale bars must be present on all microscopy images, including inset magnifications. Molecular weight or nucleic acid size markers must be included on all gel electrophoresis. Please add a scale bar to Figure S4.

Also, please avoid pairing red and green for images and graphs to ensure legibility for color-blind readers. If red and green are paired for images, please ensure that the particular red and green hues used in micrographs are distinctive with any of the colorblind types. If not, please modify colors accordingly or provide separate images of the individual channels.

3) Statistical analysis: Error bars on graphic representations of numerical data must be clearly described in the figure legend. The number of independent data points (n) represented in a graph must be indicated in the legend. Please, indicate whether 'n' refers to technical or biological replicates (i.e. number of analyzed cells, samples or animals, number of independent experiments). If independent experiments with multiple biological replicates have been performed, we recommend using distribution-reproducibility SuperPlots (please see Lord et al., JCB 2020) to better display the distribution of the entire dataset, and report statistics (such as means, error bars, and P values) that address the reproducibility of the findings.

Statistical methods should be explained in full in the materials and methods. For figures presenting pooled data the statistical measure should be defined in the figure legends. Please also be sure to indicate the statistical tests used in each of your experiments (both in the figure legend itself and in a separate methods section) as well as the parameters of the test (for example, if you ran a t-test, please indicate if it was one- or two-sided, etc.). Also, if you used parametric tests, please indicate if the data distribution was tested for normality (and if so, how). If not, you must state something to the effect that "Data distribution was assumed to be normal but this was not formally tested."

4) Abstract: Please change "research" in the second to last sentence to "study" or "findings".

5) Materials and methods: Should be comprehensive and not simply reference a previous publication for details on how an experiment was performed. Please provide full descriptions (at least in brief) in the text for readers who may not have access to referenced manuscripts. The text should not refer to methods "...as previously described." Please also indicate the acquisition and quantification methods for immunoblotting/western blots.

6) For all cell lines, vectors, constructs/cDNAs, etc. - all genetic material: please include database / vendor ID (e.g., Addgene, ATCC, etc.) or if unavailable, please briefly describe their basic genetic features, even if described in other published work or gifted to you by other investigators (and provide references where appropriate). Please be sure to provide the sequences for all of your oligos: primers, si/shRNA, RNAi, gRNAs, etc. in the materials and methods. You must also indicate in the methods the

source, species, and catalog numbers/vendor identifiers (where appropriate) for all of your antibodies, including secondary. If antibodies are not commercial, please add a reference citation if possible.

7) Microscope image acquisition: The following information must be provided about the acquisition and processing of images:

- a. Make and model of microscope
- b. Type, magnification, and numerical aperture of the objective lenses
- c. Temperature
- d. Imaging medium
- e. Fluorochromes
- f. Camera make and model
- g. Acquisition software
- h. Any software used for image processing subsequent to data acquisition. Please include details and types of operations involved (e.g., type of deconvolution, 3D reconstitutions, surface or volume rendering, gamma adjustments, etc.).

8) References: There is no limit to the number of references cited in a manuscript. References should be cited parenthetically in the text by author and year of publication. Abbreviate the names of journals according to PubMed. JCB formatting does not allow for supplemental references. Please remove these and add any non-duplicate references to the main reference list.

9) Supplemental materials: Articles may generally have up to 5 supplemental figures and 10 videos. You currently exceed this limit but, in this case, we will be able to give you the extra space but please try to consolidate these if possible. Please also note that tables, like figures, should be provided as individual, editable files. A summary of all supplemental material should appear at the end of the Materials and methods section. Please include one brief sentence per item.

10) Video legends: Should describe what is being shown, the cell type or tissue being viewed (including relevant cell treatments, concentration and duration, or transfection), the imaging method (e.g., time-lapse epifluorescence microscopy), what each color represents, how often frames were collected, the frames/second display rate, and the number of any figure that has related video stills or images.

11) eTOC summary: A ~40-50 word summary that describes the context and significance of the findings for a general readership should be included on the title page. The statement should be written in the present tense and refer to the work in the third person. It should begin with "First author name(s) et al..." to match our preferred style.

13) A separate author contribution section is required following the Acknowledgments in all research manuscripts. All authors should be mentioned and designated by their first and middle initials and full surnames. We encourage use of the CRediT nomenclature (<https://casrai.org/credit/>).

14) ORCID IDs: ORCID IDs are unique identifiers allowing researchers to create a record of their various scholarly contributions in a single place. Please note that ORCID IDs are required for all authors. At resubmission of your final files, please be sure to provide your ORCID ID and those of all co-authors.

15) JCB requires authors to submit Source Data used to generate figures containing gels and Western blots with all revised manuscripts. This Source Data consists of fully uncropped and unprocessed images for each gel/blot displayed in the main and supplemental figures. Since your paper includes cropped gel and/or blot images, please be sure to provide one Source Data file for each figure that contains gels and/or blots along with your revised manuscript files. File names for Source Data figures should be alphanumeric without any spaces or special characters (i.e., SourceDataF#, where F# refers to the associated main figure number or SourceDataFS# for those associated with Supplementary figures). The lanes of the gels/blots should be labeled as they are in the associated figure, the place where cropping was applied should be marked (with a box), and molecular weight/size standards should be labeled wherever possible. Source Data files will be directly linked to specific figures in the published article.

** Please add Source Data for all blots in Figure S1. **

16) Journal of Cell Biology now requires a data availability statement for all research article submissions. These statements will be published in the article directly above the Acknowledgments. The statement should address all data underlying the research presented in the manuscript. Please visit the JCB instructions for authors for guidelines and examples of statements at

(<https://rupress.org/jcb/pages/editorial-policies#data-availability-statement>).

B. FINAL FILES:

Thank you for your attention to these final processing requirements. Please revise and format the manuscript and upload materials within 7 days. If you need an extension for whatever reason, please let us know and we can work with you to determine a suitable revision period.

Thank you for this interesting contribution, we look forward to publishing your paper in Journal of Cell Biology.

Sincerely,

Min Wu, PhD
Monitoring Editor
Journal of Cell Biology

Dan Simon, PhD
Scientific Editor
Journal of Cell Biology